# Towards Multimodal Time Series Anomaly Detection with Semantic Alignment and Condensed Interaction

**Shiyan Hu[1], Jianxin Jin[1], Yang Shu[1], Peng Chen[1], Bin Yang[1], Chenjuan Guo[1]***

[1]East China Normal University
{syhu,jxjin,pchen}@stu.ecnu.edu.cn
{yshu,byang,cjguo}@dase.ecnu.edu.cn

## ABSTRACT

Time series anomaly detection plays a critical role in many dynamic systems. Despite its importance, previous approaches have primarily relied on unimodal numerical data, overlooking the importance of complementary information from other modalities. In this paper, we propose a novel multimodal time series anomaly detection model (MindTS) that focuses on addressing two key challenges: (1) how to achieve semantically consistent alignment across heterogeneous multimodal data, and (2) how to filter out redundant modality information to enhance cross-modal interaction effectively. To address the first challenge, we propose Fine-grained Time-text Semantic Alignment. It integrates exogenous and endogenous text information through cross-view text fusion and a multimodal alignment mechanism, achieving semantically consistent alignment between time and text modalities. For the second challenge, we introduce Content Condenser Reconstruction, which filters redundant information within the aligned text modality and performs cross-modal reconstruction to enable interaction. Extensive experiments on six real-world multimodal datasets demonstrate that the proposed MindTS achieves competitive or superior results compared to existing methods. The code is available at: https://github.com/decisionintelligence/MindTS.

## 1 INTRODUCTION

Time series anomaly detection identifies anomalous events that significantly deviate from the majority within time series data. It has been widely applied in various domains, including healthcare monitoring, financial fraud detection, and network intrusion detection (Li et al., 2021; Yang et al., 2023b; Boniol et al., 2022; 2024; Sylligardos et al., 2023).

In various real-world scenarios, data often exists in a multimodal form, such as time series (Liu & Paparrizos, 2024; Dai et al., 2024), text (Enevoldsen et al., 2024; Chen et al., 2024b), images (Costanzino et al., 2024; Zhou et al., 2024; Bhunia et al., 2024), and videos (Li et al., 2024; Chen et al., 2024a; He et al., 2024), which collectively serve as complementary heterogeneous information sources. Among these, the text modality, which contains contextual descriptions and provides rich background information for time series, is easy to obtain due to its wide availability. For instance, financial experts combine transaction data on stocks with reports and policies to detect market anomalies. Despite this, time series most existing anomaly detection models remain confined to unimodal numerical frameworks (Yang et al., 2023c; Wang et al., 2023; Song et al., 2023; Shentu et al., 2025; Wu et al., 2025), overlooking the potential of utilizing multimodal data. Therefore, building multimodal time series anomaly detection models becomes a natural and necessary step forward. In this work, we focus on the time series and text modalities, rather than aiming to build a universal multimodal framework that also handles image or video modalities. This focus raises a key research question: **how can we effectively integrate text information and time series?**

Since different modalities reside in distinct semantic spaces, achieving precise alignment between text and time series is crucial for leveraging textual information effectively. A straightforward ap-

---
*Corresponding authors

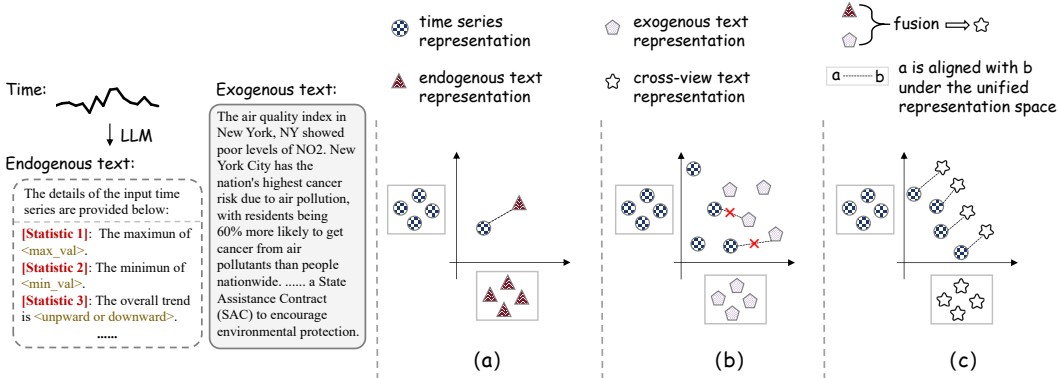

Figure 1: (a) LLM-based methods generate endogenous text from time series without incorporating exogenous information. (b) Exogenous-based methods incorporate text information by retrieving background knowledge from the web. The absence of connecting lines indicates that the two modalities are not aligned. (c) MindTS employs cross-view fusion to ensure semantic consistency between the exogenous text and the time series, enabling more precise alignment across modalities.

proach (Jin et al., 2023; Zhou et al., 2023; Gruver et al., 2023; Cao et al., 2023; Kowsher et al., 2024) is to generate *endogenous text* from the time series itself using large language models (LLMs), which naturally ensures modality alignment (Figure 1a). However, such text typically offers limited semantic richness, as it can only capture intrinsic patterns within the time series. To address this limitation, recent studies (Liu et al., 2024b; Li et al., 2025b) have explored incorporating *exogenous text*, such as news reports or documents, as external contextual information. However, the effectiveness of such methods heavily depends on the quality of exogenous text. The external information sources are often scattered, making semantic alignment with the time series inherently difficult (Figure 1b). Therefore, **a key challenge** is how to incorporate informative exogenous text while ensuring semantic consistency and alignment with the time series.

Moreover, while text provides complementary information to time series, it may also introduce redundant content that hinders anomaly detection. Existing multimodal time series methods perform direct fusion strategies (Liu et al., 2024b; Jin et al., 2023), assuming that all text information is equally useful. This overlooks the lengthy details or irrelevant descriptions within the text modality, which may dilute the contribution of genuinely informative content. In natural language processing, recent approaches (Cha et al., 2023; Liang et al., 2023) filter text representations using techniques such as random masking or by applying random semantic parsing functions, such as paraphrasing, summarization, or translation, to perturb the text for filtering (Ji et al., 2024). However, when applied to multimodal time series, such strategies fail to consider the relevance of text content to the time series, which may result in high-value text information being randomly masked while low-value text information is retained. Therefore, **another key challenge** is how to mitigate the impact of redundant content on cross-modal interaction through an effective filtering mechanism.

To address these challenges, we propose **MindTS**, a Multimodal Time Series Anomaly Detection with Semantic Alignment and Condensed Interaction. Specifically, we propose a fine-grained time-text semantic alignment module that divides the text into two complementary views: *exogenous text and endogenous text*. The exogenous text contains background information from external sources, suitable for sharing across different time steps. In contrast, the endogenous text is derived directly from the time series, exhibiting time-specific characteristics correlated with temporal patterns. To achieve semantic consistency alignment between time-text pairs, we apply cross-view fusion to integrate the complementary strengths of the two text views. The resulting fused text is further aligned with the time series (Figure 1c). Furthermore, we propose a content condenser reconstruction mechanism to filter redundant text information and enhance the effectiveness of cross-modal interaction. Specifically, given aligned text representations as input, the content condenser filters out redundant information from the text by minimizing mutual information, resulting in condensed text representations. The condensed text representations are then used to reconstruct the masked time series, which strengthens cross-modal interaction. The contributions are summarized as follows:

- We propose a novel fine-grained time–text semantic alignment method that jointly exploits both exogenous and endogenous text representations of time patches. The exogenous text introduces external background knowledge, while the endogenous text captures specific characteristics directly derived from time series. By integrating these two complementary text views, our approach ensures more precise semantic alignment with text and time series.

- We propose a novel method, content condenser reconstruction, to filter redundant textual information. By performing cross-modal reconstruction of time series from condensed text, the content condenser reconstruction enhances interaction between modalities.

- Our proposed multimodal anomaly detection model, MindTS, has been extensively evaluated on multimodal datasets. Compared with existing unimodal baselines and multimodal time series frameworks, MindTS achieves competitive or superior performance.

## 2 RELATED WORK

### 2.1 TIME SERIES ANOMALY DETECTION

Time series anomaly detection has been extensively studied, and existing methods can be broadly categorized into non-learning (Breunig et al., 2000; Goldstein & Dengel, 2012; Yeh et al., 2016), classical machine learning (Liu et al., 2008; Ramaswamy et al., 2000; Shyu et al., 2003; Yairi et al., 2001), and deep learning (Xu et al., 2021; Deng & Hooi, 2021; Yang et al., 2023c; Shentu et al., 2025; Hu et al., 2024; Qiu et al., 2025a; Xu et al., 2024; Miao et al., 2025). Deep learning methods can be further divided into reconstruction-based, prediction-based, and contrastive learning-based. DADA (Shentu et al., 2025) adopts a dual-adversarial decoder framework to reconstruct both normal and abnormal series, where abnormal samples are expected to yield high reconstruction errors. GDN (Deng & Hooi, 2021) couples structure learning with graph neural networks by using attention over neighboring sensors to forecast values, and derives anomaly scores from prediction errors. DCdetector (Yang et al., 2023c) is the first to introduce contrastive learning into time series anomaly detection. It maps samples into a shared embedding space, where normal points exhibit strong correlation with others, while anomalous points show weak correlations (Yang et al., 2023a). Although these methods have achieved impressive performance in unimodal time series anomaly detection, they often overlook the rich semantic information available in other modalities, which limits their robustness in complex real-world scenarios.

### 2.2 MULTIMODAL TIME SERIES ANALYSIS

Multimodal approaches mainly exploit time series and textual information to enhance the robustness and effectiveness of time series analysis. Unlike traditional unimodal time series methods, multimodal time series analysis (MMTSA) presents greater challenges due to the complexity of cross-modal interaction and heterogeneous data integration. With the recent advances in LLMs, mainstream research in MMTSA (Liu et al., 2024b; Pan et al., 2024; Liu et al., 2024c; Kowsher et al., 2024; Wang et al., 2025) has focused on transforming time series data into text or image formats and feeding them into LLMs or vision models, respectively. These approaches typically employ a multimodal fusion network to integrate information across modalities and boost overall model performance. For instance, LLM-Mixer (Kowsher et al., 2024) decomposes time series into seasonal and trend components, and feeds them along with textual prompts into a frozen pre-trained LLM. The LLM then generates predictions by leveraging both semantic knowledge and temporal structure. Time-MMD (Liu et al., 2024b) attempts to incorporate exogenous text to improve time series analysis tasks. However, exogenous textual sources are often scattered and weakly correlated with the semantics of specific time segments. Relying solely on hard alignment through temporal step synchronization overlooks deeper semantic associations between time series and text. Furthermore, text data often contains much redundant content. Without appropriate selection mechanisms, cross-modal interaction may introduce semantic redundancy, which hinders the identification of anomalous patterns. To address these issues, our proposed method achieves precise alignment between semantically consistent time-text representations by integrating exogenous and endogenous text information. Moreover, we introduce a mutual information minimization mechanism and a cross-modal reconstruction strategy to achieve text compression and modality-level time series reconstruction. These strategies improve the model's ability to identify anomalous patterns.

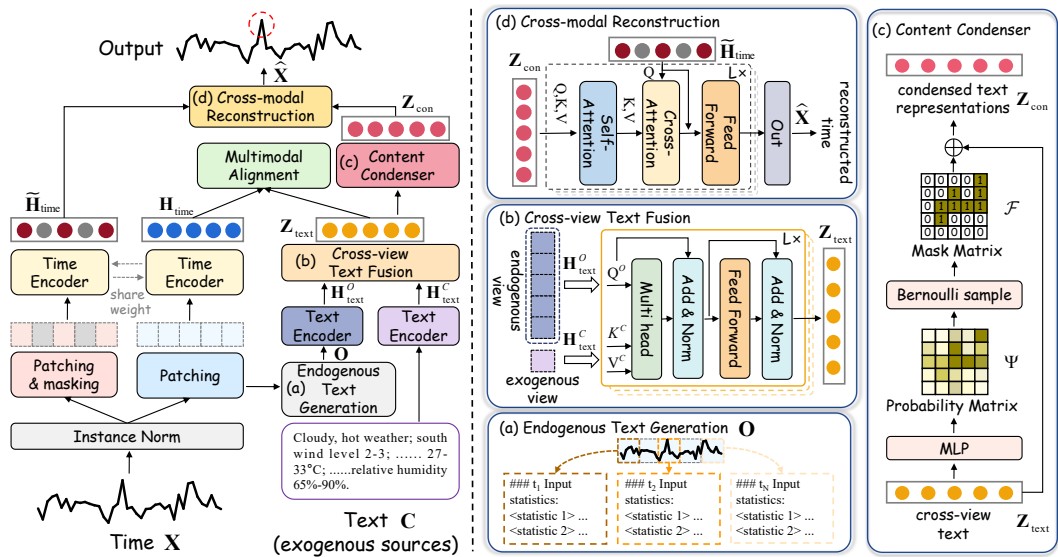

Figure 2: MindTS overview. Given an input time series $\mathbf{X}$, we first apply instance normalization and patching, then encode the patches using a time encoder. (a) Each patch generates its corresponding endogenous text $\mathbf{O}$. Along with the input exogenous text $\mathbf{C}$, both views are encoded and (b) fused via cross-view fusion to obtain fused text representations $\mathbf{Z}_{\text{text}}$. Time and text representations are then semantically aligned via a multimodal alignment layer. (c) To mitigate textual redundancy, the aligned text is compressed using a content condenser. Finally, (d) the condensed text $\mathbf{Z}_{\text{con}}$ is used to reconstruct the masked time series, enhancing cross-modal interaction.

## 3 METHODOLOGY

Given input time series of length $T$ as $\mathbf{X} = (x_1, \ldots, x_T) \in \mathbb{R}^{T \times D}$, where $D$ is the number of features. Traditional unimodal time series anomaly detection outputs $\mathbf{Y} = (y_1, \ldots, y_T) \in \{0, 1\}^T$, where $y_t = 1, t \in \{1, 2, ..., T\}$, indicates that timestamp $t$ is identified as an anomaly. In the task of multimodal time series anomaly detection, we consider the time series data with an additional text modalities. Here we specifically focus on fusing a time series modality $\mathbf{X}$ with a text modality, where the exogenous text modality is represented as a sequence of length $S$, given by $\mathbf{C} = (c_1, \ldots, c_S) \in \mathbb{R}^S$. Similar to the unimodal time series anomaly detection, the problem of multimodal time series anomaly detection also determines whether $y_t$ is an anomaly or not.

### 3.1 OVERALL FRAMEWORK

In order to resolve the problem of multimodal time series anomaly detection, we propose the model MindTS, as illustrated in Figure 2. This model provides an anomaly identification mechanism from a cross-modal perspective based on the input time series and text.

The time series is first input to *instance norm layer* to perform instance normalization and channel-independent processing (Ulyanov et al., 2017; Kim et al., 2021), then the result is output to *patching & time encoder layer* for the fine-grained modeling of patches. By the widely used time transformer (Cirstea et al., 2022; Nie et al., 2022), the following results are derived through the time encoder:

$$\mathbf{P} = \left\{ \mathbf{P}_{\text{time}}^1, \mathbf{P}_{\text{time}}^2, \ldots, \mathbf{P}_{\text{time}}^N \right\} = \text{Patching}(\mathbf{X}), \quad \mathbf{H}_{\text{time}} = \text{TimeEncoder}(\mathbf{P}), \tag{1}$$

where $\mathbf{P}_{\text{time}}^i \in \mathbb{R}^{p \times D}$ denotes the $i$-th patch of $\mathbf{X}$, $\mathbf{H}_{\text{time}} \in \mathbb{R}^{N \times d_{model}}$, $N = [T - p]/l + 1$ is the total patch number, $p$ is the patch size and $l$ is the horizontal sliding stride. Furthermore, endogenous text $\mathbf{O} = \left\{ o^1, o^2, ..., o^N \right\}$ are generated for each patch, where $o^i$ denotes the text generated by LLMs using specifically designed prompts based on the $i$-th patch. In addition, $\mathbf{X}$ is processed through *patching & masking* and a shared-weight time encoder to obtain the masked time representation, denoted as $\widetilde{\mathbf{H}}_{\text{time}}$.

Next, endogenous text $\mathbf{O}$ and exogenous text $\mathbf{C}$ are fed into *text encoder* to model their text representations $\mathbf{H}_{\text{text}}^O$ and $\mathbf{H}_{\text{text}}^C$, respectively. Based on them, the *cross-view text fusion layer* is employed

to capture semantic dependencies, resulting in the text representation as $\mathbf{Z}_{\text{text}}$, which is semantically correlated with the corresponding patch representation $\mathbf{H}_{\text{time}}$. Subsequently, a *multimodal alignment layer* is employed to align semantically consistent time-text pairs.

Finally, based on the aligned text representations $\mathbf{Z}_{\text{text}}$, the *content condenser* filters out redundant text, giving rise to condensed text $\mathbf{Z}_{\text{con}}$. It is then utilized together with the masked time representation $\tilde{\mathbf{H}}_{\text{time}}$ by *cross-modal reconstruction*, which produces the final output $\hat{\mathbf{X}}$ and facilitates modality interaction. Reconstruction error is the anomaly score.

## 3.2 FINE-GRAINED TIME-TEXT SEMANTIC ALIGNMENT

The fine-grained time-text semantic alignment consists of three components: 1) the *endogenous text generation*, 2) the *cross-view text fusion* and 3) the *multimodal alignment* strategy. Specifically, *cross-view text fusion* is designed to integrate text from different views (endogenous and exogenous text), helping enhance semantic consistency between text and time series. The *multimodal alignment* aims to guide the alignment between time representations $\mathbf{H}_{\text{time}}$ and text representations $\mathbf{Z}_{\text{text}}$ from *cross-view text fusion* within a unified space.

**Endogenous text generation.** To mitigate semantic drift and output uncertainty in directly converting time series into natural language (Kowsher et al., 2024; Jin et al., 2023), we design unified endogenous text prompt templates (e.g., mean, extrema, trend) and apply them to each patch to generate corresponding endogenous texts. In this case, the limitations caused by generating a single global prompt can be avoided, and the dynamic property of the time series is matched.

The text encoder leverages open-source LLMs (Liu et al., 2024a; Radford et al., 2019) to encode the endogenous text, resulting in the time-specific text representation $\mathbf{H}_{\text{text}}^O \in \mathbb{R}^{N \times d_{model}}$ as follows:

$$\mathbf{H}_{\text{text}}^O = \text{TextEncoder}(\{o^1, o^2, ..., o^N\}), \tag{2}$$

where $o^i \in \mathbb{R}^{e \times d_{model}}$, $e$ is the LLM's vocabulary size. This prompt leverages the semantic knowledge of LLM to enhance semantic consistency with the time series. On the other hand, to fully exploit the exogenous text $\mathbf{C}$ information, we treat it as a shared text across all patches. This approach ensures that the model does not lose background context due to the limited scope of individual patches. Similarly, the exogenous text encoded by a text encoder, resulting in $\mathbf{H}_{\text{text}}^C \in \mathbb{R}^{1 \times d_{model}}$.

**Cross-view text fusion.** To leverage the rich background knowledge in exogenous text and the strong semantic relevance of endogenous text to time series, MindTS integrates text information from endogenous and exogenous text views, introducing background knowledge while enabling precise mapping to specific patches. Specifically, we adopt a cross-view attention mechanism that can selectively extract complementary information from two text views. To enhance semantic consistency with the time series and extract the most relevant background information, we use the endogenous text $\mathbf{H}_{\text{text}}^O$ as the query and the exogenous text $\mathbf{H}_{\text{text}}^C$ as the key and value to obtain the fused text representation $\mathbf{Z}_{\text{text}}$. This process is expressed as:

$$\mathbf{Z}_{\text{text}} = \text{LayerNorm}\left(\hat{\mathbf{Z}}_{\text{text}} + \text{FeedForward}\left(\hat{\mathbf{Z}}_{\text{text}}\right)\right), \tag{3}$$

$$\hat{\mathbf{Z}}_{\text{text}} = \text{LayerNorm}\left(\mathbf{H}_{\text{text}}^O + \text{CrossAttn}(\mathbf{H}_{\text{text}}^O, \mathbf{H}_{\text{text}}^C, \mathbf{H}_{\text{text}}^C)\right), \tag{4}$$

where $\hat{\mathbf{Z}}_{\text{text}}$ is intermediate variable. LayerNorm$(\cdot)$ denotes layer normalization as widely adopted in (Vaswani et al., 2017; Qiu et al., 2025c; Chen et al., 2024c), FeedForward$(\cdot)$ denotes a multi-layer feedforward network, and CrossAttn(Q, K, V) represents the cross-attention layer.

**Multimodal alignment strategy.** Time series manifest as continuous signals with strong temporal dependencies, while text is discrete, making semantic alignment between the two modalities difficult. Traditional methods (e.g., add or concatenation) fail to capture the semantic alignment. To address this, we employ contrastive learning to explicitly align the two modalities, enhancing semantically consistent alignment by pulling positive pairs (aligned time-text) closer and pushing negative (unrelated ones) farther. Specifically, the similarity matrix between the two representations, $\mathbf{H}_{\text{time}}$ and $\mathbf{Z}_{\text{text}}$, is indicated as follows:

$$\mathbf{K}_{\text{TT}} = \begin{bmatrix} k(\mathbf{h}_{\text{time}}^1, \mathbf{z}_{\text{text}}^1) & \cdots & k(\mathbf{h}_{\text{time}}^1, \mathbf{z}_{\text{text}}^N) \\ \vdots & k(\mathbf{h}_{\text{time}}^j, \mathbf{z}_{\text{text}}^g) & \vdots \\ k(\mathbf{h}_{\text{time}}^N, \mathbf{z}_{\text{text}}^1) & \cdots & k(\mathbf{h}_{\text{time}}^N, \mathbf{z}_{\text{text}}^N) \end{bmatrix} \in \mathbb{R}^{N \times N}, \tag{5}$$

where $k(\cdot, \cdot)$ denotes the similarity between time and text representations. If $j = g$, the $k(\mathbf{h}_{\text{time}}^j, \mathbf{z}_{\text{text}}^g)$ is identified as a positive pair. Therefore, multimodal alignment loss $\mathcal{L}_{MA}$ is defined as:

$$\mathcal{L}_{MA} = -\frac{1}{2N} \left[ \sum_{j=1}^{N} \log \frac{\exp\left( k(\mathbf{h}_{\text{time}}^j, \mathbf{z}_{\text{text}}^j)/\tau \right)}{\sum_{g=1}^{N} \exp\left( k(\mathbf{h}_{\text{time}}^j, \mathbf{z}_{\text{text}}^g)/\tau \right)} + \sum_{g=1}^{N} \log \frac{\exp\left( k(\mathbf{h}_{\text{time}}^g, \mathbf{z}_{\text{text}}^g)/\tau \right)}{\sum_{j=1}^{N} \exp\left( k(\mathbf{h}_{\text{time}}^j, \mathbf{z}_{\text{text}}^g)/\tau \right)} \right], \tag{6}$$

where $\tau$ denotes the temperature.

### 3.3 CONTENT CONDENSER RECONSTRUCTION

As shown in Figure 2, the content condenser reconstruction includes: 1) the *content condenser*, and 2) the *cross-modal reconstruction*. Specifically, after aligning fine-grained representations that are semantically consistent across modalities, the *content condenser* filters redundant text information via masking. It utilizes the condensed text representation to reconstruct the time series.

**Content condenser.** Inspired by the Information Bottleneck (IB) principle (Tishby et al., 2000; Tishby & Zaslavsky, 2015), we propose the *content condenser* to filter redundant representations based on the aligned text representation $\mathbf{Z}_{\text{text}}$. This process produces a condensed text representation $\mathbf{Z}_{\text{con}}$ while preserving essential information for time series. Formally, the objective of finding the optimal condensed representation $\mathbf{Z}_{\text{con}}$ is defined as:

$$\mathbf{Z}_{\text{con}}^* = \arg \min_{\mathbb{P}(\mathbf{Z}_{\text{con}}|\mathbf{Z}_{\text{text}})} I(\mathbf{Z}_{\text{text}}; \mathbf{Z}_{\text{con}}) + R(\hat{\mathbf{X}}, \mathbf{Z}_{\text{con}}), \tag{7}$$

where $I(\cdot; \cdot)$ denotes the mutual information between aligned and condensed text representations. Minimizing it encourages the model to learn more compact representations. $R(\cdot, \cdot)$ denotes the reconstruction objective. Based on this, we introduce cross-modal reconstruction to ensure the condensed text retains sufficient information to recover the time series $\hat{\mathbf{X}}$.

Specifically, given the aligned text representations $\mathbf{Z}_{\text{text}}$, we use an MLP to compute a probability matrix $\Psi = [\psi_i]_{i=1}^N$. A binary mask $\mathcal{F} \sim \text{Bernoulli}(\Psi) \in \{0, 1\}^N$ is then sampled, where the higher the value of $\psi_i$, the more likely it is to sample $\mathcal{F}_i = 1$. The condensed representation is obtained as $\mathbf{Z}_{\text{con}} = \mathbf{Z}_{\text{text}} \odot \mathcal{F}$, where $\odot$ is the element-wise multiplication. To enable gradient propagation during sampling, we adopt the straight-through estimator trick (Jang et al., 2016).

In order to control the marginal distribution of condensed text, thus regulating the condensing level, from the idea of latent distribution with variational auto-encoders, we introduce the distribution $\mathbb{G}(\mathbf{Z}_{\text{con}}) \sim \prod_{i=1}^{N} \text{Bernoulli}(r)$ subject to a hyperparameter $\mu \in (0, 1)$. By adjusting the value of $\mu$, we can restrain the condensing degree of the proposed model. To quantify the mutual information, the following lemma is proposed to get the upper bound before building the loss function.

**Lemma 1.** *For the mutual information $I(\mathbf{Z}_{\text{text}}; \mathbf{Z}_{\text{con}})$, there exists the following tight upper bound that can approximate its value:*

$$I(\mathbf{Z}_{\text{text}}; \mathbf{Z}_{\text{con}}) \leq \mathbb{E}_{\mathbf{Z}_{\text{text}}}[\text{KL}(\mathbb{P}(\mathbf{Z}_{\text{con}}|\mathbf{Z}_{\text{text}})||\mathbb{G}(\mathbf{Z}_{\text{con}}))], \tag{8}$$

*where $\text{KL}(\cdot)$ denotes the Kullback–Leibler (KL) divergence, defined as $\text{KL}(\mathbb{P}(x)||\mathbb{G}(x)) = \sum_{\text{x}} \mathbb{P}(\text{x}) \log \frac{\mathbb{P}(\text{x})}{\mathbb{G}(\text{x})}$, $\mathbb{P}(\cdot)$ is the probability distribution and $\mathbb{G}(\cdot)$ is a variational approximation. The proof is given in Appendix B.*

Utilizing the upper bound in Lemma 1, we can compute the KL divergence to obtain the loss function $\mathcal{L}_{CC}$ as follows:

$$\mathcal{L}_{CC} = \sum_{i=1}^{N} \psi_i \log \frac{\psi_i}{\mu} + (1 - \psi_i) \log \frac{1 - \psi_i}{1 - \mu}. \tag{9}$$

Another issue is that the condensed text might possess a large difference between the $i$-th patch and the $(i+1)$-th patch, which results in the discontinuity and instability of the condenser reconstruction. To avoid this problem, we introduce $\phi_i = \sqrt{(\psi_{i+1} - \psi_i)^2}$ to compute the mask score difference of two Bernoulli samplings. Then, $\mathcal{L}_{SM} = \frac{1}{N} \sum_{i=1}^{N-1} \phi_i$ is proposed to guarantee the smoothness of condensed text representation, ensuring stability in the learned features. In summary, the loss of the content condenser module is defined as $\mathcal{L}_{CL} = \mathcal{L}_{CC} + \mathcal{L}_{SM}$.

**Cross-modal reconstruction.** To enhance cross-modal interaction, a straightforward approach would be to perform time series reconstruction directly from the entire time series and the condensed text. However, as the time series itself contains abundant information for reconstruction, this process cannot fully encourage the model to capture deeper cross-modal dependencies. To address this, we design a more challenging objective: time series reconstruction with the random masked time series $\tilde{\mathbf{X}}$ and condensed texts $\mathbf{Z}_{\text{con}}$. This design strengthens cross-modal dependency and encourages the condensed representation to preserve richer time series–related information. As shown in Figure 2 (d), $\mathbf{X}$ is processed by *patching & masking* to obtain $\tilde{\mathbf{X}}$, which is then encoded by a time encoder to produce $\tilde{\mathbf{H}}_{\text{time}}$. The encoder shares weights with another time encoder. Given the inputs $\tilde{\mathbf{H}}_{\text{time}}$ and $\mathbf{Z}_{\text{con}}$, the reconstructed output $\hat{\mathbf{X}} \in \mathbb{R}^{T \times D}$ is obtained $\hat{\mathbf{X}} = \text{Projection}(\mathbf{U}_{\text{TT}})$, and $\mathbf{U}_{\text{TT}}$ is denoted as:

$$\mathbf{U}_{\text{TT}} = \text{FeedForward}\left(\tilde{\mathbf{H}}_{\text{time}} + \text{CrossAttn}\left(\tilde{\mathbf{H}}_{\text{time}}, \mathbf{Z}'_{\text{con}}, \mathbf{Z}'_{\text{con}}\right)\right), \tag{10}$$

where $\mathbf{Z}'_{\text{con}} = \text{MSA}(\mathbf{Z}_{\text{con}}, \mathbf{Z}_{\text{con}}, \mathbf{Z}_{\text{con}})$ denotes the *self-attention layer*. The cross-modal reconstruction function is formalized as:

$$\mathcal{L}_{\text{Rec}} = \left\|\mathbf{X} - \hat{\mathbf{X}}\right\|_F^2. \tag{11}$$

### 3.4 Joint optimization and inference

Our total loss $\mathcal{L}$ primarily consists of three components: the multimodal alignment loss $\mathcal{L}_{MA}$, the condenser loss based on condensed text $\mathcal{L}_{CL}$, and the cross-modal reconstruction loss $\mathcal{L}_{Rec}$. Therefore, the proposed total loss function is written as:

$$\mathcal{L} = \mathcal{L}_{MA} + \mathcal{L}_{CL} + \mathcal{L}_{\text{Rec}}, \tag{12}$$

During the inference stage, the anomaly score at the current timestamp is computed based on the mean squared error between the time input $\mathbf{X}$ and its reconstructed output.

## 4 Experiments

### 4.1 Experimental settings

**Datasets.** We conduct experiments using 6 real-world datasets (Weather, Energy, Environment, KR, EWJ, and MDT) to assess the performance of MindTS. Each dataset contains both numerical time series and corresponding exogenous text. More details of the datasets are included in Appendix A.1.

**Baselines.** We extensively compare MindTS against 17 baselines, including (1) LLM-based methods: LLMMixer (LMixer) (Kowsher et al., 2024), UniTime (UTime) (Liu et al., 2024c), GPT4TS (G4TS) (Zhou et al., 2023); (2) Pre-trained methods: DADA (Shentu et al., 2025), Timer (Liu et al., 2024d), UniTS (Gao et al., 2024); (3) Deep learning-based methods: ModernTCN (Modern) (Luo & Wang, 2024), TimesNet (TsNet) (Wu et al., 2023), DCdetector (DC) (Yang et al., 2023c), Anomaly Transformer (A.T.) (Xu et al., 2021), PatchTST (Patch) (Nie et al., 2022), TranAD (Tuli et al., 2022), iTransformer (iTrans) (Liu et al., 2023); (4) Non-learning methods: PCA (Shyu et al., 2003), IForest (IF) (Liu et al., 2008), LODA (Pevnỳ, 2016), HBOS (Goldstein & Dengel, 2012). Further details concerning baselines are available in Appendix A.2.

**Metrics.** We adopt Label-based metric: Affiliated-F1-score (Aff-F) (Huet et al., 2022) and Score-based metric: VUS-PR (V-PR) (Paparrizos et al., 2022), VUS-ROC (V-ROC) (Paparrizos et al., 2022) as evaluation metrics. We report the algorithm performance under a total of 16 evaluation metrics in the Appendix A.3. More implementation details are presented in the Appendix A.4.

### 4.2 Main results

We evaluate MindTS with 17 competitive baselines on 6 real-world datasets, as shown in Table 1. MindTS achieves state-of-the-art (SOTA) performance across all datasets under the Aff-F, V-PR, and V-ROC metrics, which demonstrates that MindTS effectively combines multimodal data to detect anomalies. We further incorporate the 11 recent methods that perform well, as shown in Table 1, into the multimodal time series framework MM-TSFLib (Liu et al., 2024b), as reported in Table 2. MM-TSFLib integrates textual information by performing linear interpolation between the output of time series models and bag-of-words-based text embeddings. Although this framework provides a simple yet effective way to incorporate text, MindTS achieves the best or most competitive results

Table 1: Results of MindTS compared with unimodal and LLM-based methods on six real-world datasets. These methods only use the time series in the dataset. MindTS leverages both endogenous and exogenous text together with time series. The best results are highlighted in bold, and the second-best results are underlined.

| Datasets | Metric | MindTS | DADA | LMixer | UTime | Timer | UniTS | G4TS | Modern | TsNet | DC | A.T. | Patch | TranAD | iTrans | PCA | IF | LODA | HBOS |
|---|---|---|---|---|---|---|---|---|---|---|---|---|---|---|---|---|---|---|---|
| Weather | Aff-F | **82.66** | 69.01 | 73.68 | 76.46 | 75.46 | 76.17 | 72.56 | 81.06 | 80.58 | 42.80 | 49.22 | 77.17 | 77.81 | 75.37 | 64.91 | 54.06 | 52.55 | 47.70 |
|  | V-PR | **57.48** | 30.00 | 43.47 | 51.90 | 43.21 | 44.35 | 41.30 | 52.13 | 50.09 | 18.33 | 19.17 | 50.03 | 52.08 | 42.56 | 47.13 | 49.66 | 55.03 | 46.58 |
|  | V-ROC | **82.64** | 61.03 | 71.71 | 78.45 | 73.22 | 75.08 | 70.03 | 81.91 |  | 45.56 | 43.32 | 79.97 | 78.75 | 73.37 | 57.38 | 56.45 | 57.00 | 54.16 |
| Energy | Aff-F | **74.37** | 64.38 | 65.85 | 61.98 | 60.20 | 63.84 | 66.37 | 70.76 | 66.00 | 47.07 | 43.39 | 66.85 | 49.69 | 70.81 | 57.65 | 62.03 | 63.45 | 55.85 |
|  | V-PR | **50.36** | 34.18 | 30.35 | 32.88 | 29.46 | 31.04 | 31.68 | 36.60 | 38.61 | 22.57 | 19.69 | 34.41 | 33.80 | 35.82 | 44.30 | 46.03 | 48.63 | 42.57 |
|  | V-ROC | **74.44** | 54.37 | 53.04 | 49.97 | 46.03 | 51.15 | 53.10 | 65.05 | 59.47 | 45.93 | 31.56 | 58.31 | 56.37 | 63.06 | 53.07 | 53.61 | 55.90 | 51.50 |
| Environment | Aff-F | **85.29** | 84.11 | 84.36 | 81.71 | 84.19 | 83.06 | 72.26 | 81.07 | 80.41 | 62.24 | 59.75 | 81.17 | 61.41 | 74.43 | 55.63 | 46.50 | 46.45 | 22.25 |
|  | V-PR | **56.79** | 54.20 | 52.94 | 48.87 | 51.42 | 50.24 | 23.94 | 42.26 | 50.64 | 7.69 | 18.14 | 45.78 | 4.91 | 24.87 | 17.87 | 8.94 | 18.66 | 52.15 |
|  | V-ROC | **93.78** | 87.69 | 89.75 | 91.77 | 92.10 | 92.03 | 66.79 | 89.78 | 87.97 | 41.28 | 51.98 | 90.86 | 14.20 | 73.81 | 37.08 | 46.20 | 50.69 | 51.03 |
| KR | Aff-F | **90.28** | 84.22 | 71.80 | 88.58 | 89.55 | 82.24 | 79.56 | 84.42 | 85.47 | 61.94 | 70.99 | 79.52 | 73.26 | 79.49 | 58.11 | 69.38 | 60.96 | 64.78 |
|  | V-PR | **53.15** | 45.90 | 15.13 | 46.87 | 51.41 | 43.32 | 38.23 | 39.95 | 51.60 | 8.48 | 7.94 | 36.18 | 28.42 | 27.37 | 24.19 | 43.31 | 51.82 | 52.06 |
|  | V-ROC | **89.86** | 70.82 | 52.79 | 73.55 | 75.99 | 73.93 | 67.81 | 88.87 | 79.00 | 43.04 | 41.97 | 74.65 | 41.05 | 76.12 | 47.51 | 60.70 | 59.99 | 61.41 |
| EWJ | Aff-F | **83.89** | 81.26 | 66.86 | 78.22 | 78.06 | 77.61 | 76.65 | 81.57 | 81.82 | 48.10 | 59.03 | 75.82 | 69.22 | 78.27 | 51.06 | 67.55 | 72.06 | 71.03 |
|  | V-PR | **50.42** | 43.36 | 15.21 | 32.39 | 33.17 | 39.32 | 35.63 | 44.75 | 43.15 | 15.37 | 10.85 | 36.08 | 17.80 | 28.98 | 19.38 | 37.81 | 40.08 | 41.19 |
|  | V-ROC | **84.12** | 71.79 | 46.80 | 64.49 | 67.72 | 73.91 | 67.95 | 83.88 | 75.76 | 47.10 | 31.75 | 71.56 | 49.60 | 72.16 | 45.26 | 59.24 | 61.65 | 62.07 |
| MDT | Aff-F | **89.19** | 77.99 | 67.65 | 76.28 | 78.51 | 75.57 | 80.81 | 80.81 | 80.08 | 47.33 | 66.12 | 79.47 | 63.93 | 78.66 | 54.66 | 53.74 | 55.06 | 52.33 |
|  | V-PR | **65.44** | 46.81 | 19.10 | 38.94 | 38.38 | 37.61 | 44.81 | 52.18 | 50.53 | 15.72 | 15.93 | 41.67 | 14.34 | 36.36 | 22.93 | 35.32 | 44.63 | 44.77 |
|  | V-ROC | **83.02** | 66.76 | 47.06 | 61.00 | 60.28 | 58.67 | 62.30 | 82.30 | 79.56 | 45.02 | 44.53 | 77.69 | 28.55 | 71.87 | 44.09 | 54.02 | 55.98 | 55.30 |

Table 2: The notation with ∗ indicates the results of extending the baselines to their multimodal versions using the recent time series multimodal framework MM-TSFLib, where both time series and text data from datasets are utilized.

| Datasets | Metric | MindTS | DADA* | LMixer* | UTime* | Timer* | UniTS* | G4TS* | Modern* | TsNet* | Patch* | TranAD* | iTrans* |
|---|---|---|---|---|---|---|---|---|---|---|---|---|---|
| Weather | Aff-F | **82.66** | 69.73 | 76.30 | 73.93 | 75.37 | 75.37 | 76.82 | 81.50 | 80.09 | 77.05 | 77.73 | 75.36 |
|  | V-PR | **57.48** | 30.42 | 45.94 | 43.19 | 43.36 | 44.58 | 45.83 | 53.42 | 50.53 | 50.17 | 52.09 | 40.30 |
|  | V-ROC | **82.64** | 61.51 | 75.29 | 73.71 | 73.26 | 75.21 | 74.61 | 81.67 | 82.06 | 80.08 | 78.72 | 70.11 |
| Energy | Aff-F | **74.37** | 64.80 | 61.46 | 65.38 | 60.36 | 65.42 | 67.38 | 72.13 | 66.71 | 66.28 | 50.53 | 72.49 |
|  | V-PR | **50.36** | 34.38 | 30.91 | 30.44 | 29.57 | 31.34 | 31.83 | 37.44 | 38.88 | 34.66 | 33.74 | 36.21 |
|  | V-ROC | **74.44** | 55.63 | 49.06 | 49.85 | 46.39 | 51.89 | 53.52 | 66.37 | 59.80 | 58.47 | 56.38 | 65.62 |
| Environment | Aff-F | **85.29** | 83.84 | 83.76 | 76.73 | 84.52 | 83.43 | 84.44 | 81.36 | 80.21 | 81.71 | 61.38 | 76.02 |
|  | V-PR | **56.79** | 54.20 | 51.22 | 35.64 | 51.20 | 50.06 | 56.65 | 41.36 | 50.39 | 45.52 | 4.93 | 25.85 |
|  | V-ROC | **93.78** | 88.02 | 91.74 | 84.20 | 92.02 | 91.98 | 90.22 | 89.14 | 88.14 | 90.87 | 14.29 | 73.66 |
| KR | Aff-F | **90.28** | 84.22 | 90.03 | 77.38 | 89.61 | 83.06 | 88.29 | 84.87 | 85.84 | 79.52 | 72.50 | 78.39 |
|  | V-PR | **53.15** | 45.68 | 52.98 | 37.12 | 51.56 | 44.27 | **57.93** | 40.86 | 51.73 | 36.37 | 28.47 | 28.12 |
|  | V-ROC | **89.86** | 72.08 | 75.21 | 66.96 | 75.92 | 74.25 | 80.43 | 89.22 | 78.94 | 74.80 | 41.24 | 77.17 |
| EWJ | Aff-F | **83.89** | 81.41 | 78.23 | 73.20 | 79.05 | 78.04 | 81.37 | 81.88 | 81.92 | 76.49 | 69.03 | 78.23 |
|  | V-PR | **50.42** | 43.18 | 34.06 | 25.71 | 33.36 | 39.99 | 43.93 | 45.41 | 43.28 | 36.22 | 17.87 | 29.79 |
|  | V-ROC | **84.12** | 71.06 | 69.37 | 67.57 | 68.19 | 74.39 | 76.93 | 83.98 | 75.91 | 72.21 | 49.85 | 74.11 |
| MDT | Aff-F | **89.19** | 77.89 | 78.31 | 72.33 | 77.93 | 76.70 | 81.86 | 81.68 | 80.62 | 78.87 | 63.60 | 77.47 |
|  | V-PR | **65.44** | 47.22 | 42.61 | 25.31 | 38.23 | 37.78 | 52.65 | 45.88 | 52.30 | 41.85 | 14.55 | 33.88 |
|  | V-ROC | **83.02** | 68.06 | 61.72 | 49.16 | 60.21 | 58.82 | 73.39 | 82.66 | 73.57 | 77.72 | 28.88 | 69.95 |

on all datasets, demonstrating the superior ability of MindTS to capture and integrate multimodal semantics. More baselines and metrics evaluation results can be found in Appendix C. Additional forecasting extension results are provided in Appendix H.

## 4.3 MODEL ANALYSIS

We analyze the effectiveness of fine-grained time-text semantic alignment and content condenser reconstruction, and visualize the anomaly scores. We conducted additional analytical experiments, which are presented in Appendix E, G, I, J, K.

**Ablation study.** To ascertain the impact of different modules within MindTS, we conduct ablation studies on: (a) remove the exogenous text representation; (b) remove the endogenous text representation; (c) delete the time-text semantic alignment; (d) remove the content condenser mechanism; (e) remove the cross-modal reconstruction module; (f) reverse the order of the alignment and content condenser mechanism. Figure 3 illustrates the distinct contribution of each component.

We make the following observations: Ours denotes the complete model. (a) and (b) removing either of the text representations from the two views leads to a notable performance decline. This indicates that integrating the complementary information from endogenous and exogenous texts helps improve the model performance; (c) removing the time-text semantic alignment module leads to a drop, indicating that effective modality alignment is essential for ensuring reliable anomaly detection; (d) removing the content condenser leads to significant performance degradation, likely due to redundant information from text negatively impacting the model; (e) removing cross-modal reconstruction also leads to performance degradation, suggesting that it enhances cross-modal interaction and helps extract time-relevant discriminative features from the text; (f) when the alignment and content condenser order is reversed, the model performance degrades. This may be because filtering

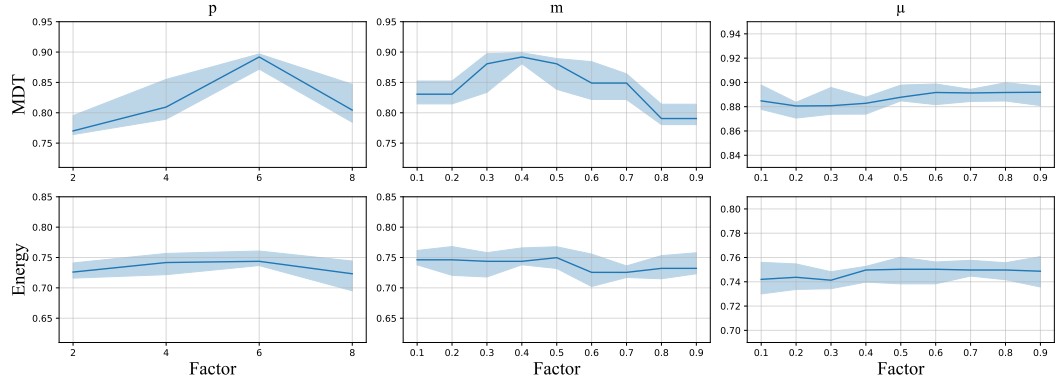

Figure 4: Results of the sensitivity analysis. The vertical coordinate shows the Aff-F score, with higher scores representing better performance. The dark line represents the mean of 5 experiments, and the light area represents the range.

is applied before alignment, causing potentially useful time-relevant information to be discarded prematurely.

**Parameter sensitivity.** We also study the parameter sensitivity of the MindTS. Figure 4 shows the performance under different patch sizes $p$, time series mask ratios $m$, and compression strengths $\mu$ in the Energy and MDT. As the experimental results show, model performance initially improves and then declines as the patch size increases. Note that a small patch size indicates a larger memory cost. In our experiments, the patch size is usually set to 6. Furthermore, we find that maintaining a mask ratio near 50% generally leads to better performance. As the mask ratio increases, reconstructing the original time series becomes more challenging, leading to poorer model performance. Besides, we further investigate the impact of the compression strength $\mu$. As shown in the results, the model maintains high performance across a broad range of $\mu$ values (0.1 to 0.9), suggesting that the content condenser is robust to varying compression levels. This stability indicates that MindTS effectively balances semantic preservation and redundancy reduction across different compression strengths.

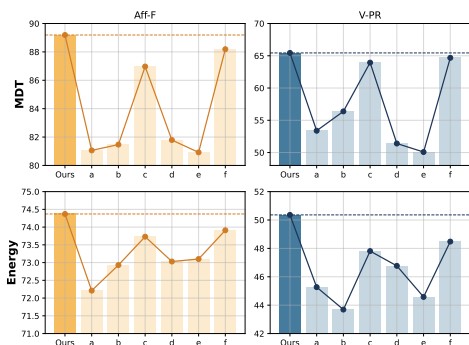

Figure 3: Ablation studies for MindTS, with the highest metrics highlighted in dark-colored bars.

**Visual analysis.** Figure 5 shows how MindTS works by visualizing different datasets. The first row shows the original data distribution along with the ground-truth anomaly positions, and the MindTS anomaly scores in the third row. It can be seen that MindTS can robustly detect anomalies. More detailed visualization cases can be found in Appendixes D, F.

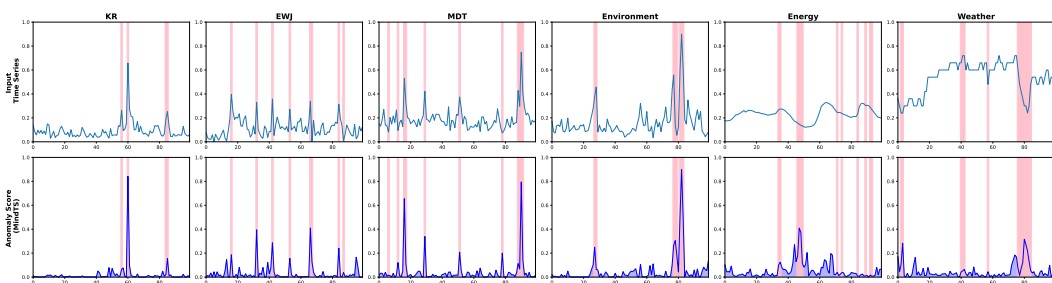

Figure 5: Visualization comparisons of anomaly scores from MindTS for all datasets.

**Comparison of different LLMs.** To more convincingly demonstrate that the performance improvements of our work stem primarily from architectural design rather than reliance on specific LLMs, we conducted experiments with different LLMs. DeepSeek is employed as the default LLM in our model throughout previous experiments. As shown in Table 3, our findings indicate that MindTS maintains stable performance across different LLMs and even achieves competitive results when using BERT. This suggests that the choice of LLMs does not exhibit a significant correlation with MindTS performance, and different LLMs can be flexibly adopted within MindTS.

Table 3: Comparison of different LLMs. Metrics include Aff-F, V-PR, and V-ROC for each dataset.

| Method | Weather | | | Energy | | | Environment | | | KR | | | EWJ | | | MDT | | |
|---|---|---|---|---|---|---|---|---|---|---|---|---|---|---|---|---|---|---|
| Metric | Aff-F | V-PR | V-ROC | Aff-F | V-PR | V-ROC | Aff-F | V-PR | V-ROC | Aff-F | V-PR | V-ROC | Aff-F | V-PR | V-ROC | Aff-F | V-PR | V-ROC |
| GPT2 | **82.71** | 57.81 | **82.97** | 73.38 | 43.78 | 68.40 | 84.95 | 52.83 | **93.96** | **91.21** | **55.88** | 90.15 | 81.39 | 48.97 | 80.96 | 85.56 | 60.24 | 80.81 |
| BERT | 80.48 | **57.84** | 82.67 | 72.99 | 43.07 | 66.65 | 84.75 | 54.25 | 92.14 | 87.05 | 53.75 | **91.61** | **84.63** | 43.95 | 78.85 | 82.30 | 53.68 | 76.37 |
| LLAMA | 81.34 | 56.93 | 82.39 | **75.64** | 49.21 | **74.58** | 84.83 | 52.99 | 92.86 | 90.81 | 53.24 | 85.17 | 80.17 | 45.76 | 82.33 | 85.48 | 59.24 | 81.90 |
| DeepSeek | 82.66 | 57.48 | 82.64 | 74.37 | **50.36** | 74.44 | **85.29** | **56.79** | 93.78 | 90.28 | 53.15 | 89.86 | 83.89 | **50.42** | **84.12** | **89.19** | **65.44** | **83.02** |

**Variant of content condenser.** To provide a more comprehensive analysis, we examine a content condenser variant that explicitly conditions its token retention probabilities on both text and unmasked time patches.

As shown in Table 4, our original design still outperforms the variant in most cases. This is because allowing the content condenser to access unmasked time patches introduces a potential shortcut. While the intention is to provide additional guidance, it makes the model focus on the time series modality. This variant tends to identify text that appears superficially aligned with known temporal patterns, rather than selecting text based on its actual semantic contribution to the reconstruction task, which is achieve by the cross-modal semantic complementarities. As a result, the ability of the condenser to filter redundant content may be reduced.

To assess the contribution of the smoothness term $\mathcal{L}_{SM}$, we conduct an ablation study. As shown in Table 5, removing $\mathcal{L}_{SM}$ leads to a performance drop. This suggests that the absence of the smoothness constraint enforced by $\mathcal{L}_{SM}$ may lead the model to generate incoherent and unstable compressed outputs.

Table 4: Performance comparison of content condenser variants conditioned with and without unmasked time patches. The best results are highlighted in bold.

| Method | Weather | | | Energy | | | Environment | | | KR | | | EWJ | | | MDT | | |
|---|---|---|---|---|---|---|---|---|---|---|---|---|---|---|---|---|---|---|
| Metric | Aff-F | V-PR | V-ROC | Aff-F | V-PR | V-ROC | Aff-F | V-PR | V-ROC | Aff-F | V-PR | V-ROC | Aff-F | V-PR | V-ROC | Aff-F | V-PR | V-ROC |
| Condenser Variant | 82.02 | 56.74 | 82.36 | 73.29 | 49.01 | 73.20 | 82.46 | 56.77 | **93.89** | 89.93 | 51.16 | **90.55** | 81.89 | 47.05 | 83.77 | 85.59 | 62.72 | 80.67 |
| MindTS | **82.66** | **57.48** | **82.64** | **74.37** | **50.36** | **74.44** | **85.29** | **56.79** | 93.78 | **90.28** | **53.15** | 89.86 | **83.89** | **50.42** | **84.12** | **89.19** | **65.44** | **83.02** |

Table 5: Ablation study on $\mathcal{L}_{SM}$ (all results in %, best results are highlighted in bold).

| Method | Weather | | | Energy | | | Environment | | | KR | | | EWJ | | | MDT | | |
|---|---|---|---|---|---|---|---|---|---|---|---|---|---|---|---|---|---|---|
| Metric | Aff-F | V-PR | V-ROC | Aff-F | V-PR | V-ROC | Aff-F | V-PR | V-ROC | Aff-F | V-PR | V-ROC | Aff-F | V-PR | V-ROC | Aff-F | V-PR | V-ROC |
| MindTS (w/o $\mathcal{L}_{SM}$) | 81.59 | 55.88 | 81.48 | 73.26 | 48.53 | 73.18 | 83.40 | 54.19 | 91.58 | 88.49 | 49.20 | 88.31 | 82.57 | 49.25 | 83.64 | 86.29 | 63.23 | 81.11 |
| MindTS | **82.66** | **57.48** | **82.64** | **74.37** | **50.36** | **74.44** | **85.29** | **56.79** | **93.78** | **90.28** | **53.15** | **89.86** | **83.89** | **50.42** | **84.12** | **89.19** | **65.44** | **83.02** |

## 5 CONCLUSION

In this work, we propose a highly capable multimodal time series anomaly detection model MindTS. The MindTS model is designed to address the limitations of existing unimodal approaches by effectively leveraging both time series data and textual information. Overall, it integrates text representations from both endogenous and exogenous views, enabling a fine-grained understanding of text semantics for precise time-text alignment. In addition, the content condenser filters out redundant information. The condensed text is further utilized for cross-modal reconstruction of the time series, optimizing cross-modal interaction. These components collectively empower MindTS with strong anomaly detection capabilities. Comprehensive experiments demonstrate that MindTS achieves competitive or superior performance.

## ETHICS STATEMENT

Our work exclusively uses publicly available benchmark datasets that contain no personally identifiable information. No human subjects are involved in this research.

## REPRODUCIBILITY STATEMENT

The performance of MindTS and the datasets used in our work are real, and all experimental results can be reproduced.

## ACKNOWLEDGEMENT

This work was partially supported by National Natural Science Foundation of China (62372179).

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

## THE USE OF LARGE LANGUAGE MODELS (LLMS)

In this work, we only adopt large language models in our methodology and endogenous text generation. Specifically, we employ large language models as the text encoder of MindTS to extract textual features. To generate endogenous text for the multimodal time series corpus, we provide raw time series to the models, encouraging them to produce descriptions of data characteristics. Note that we do not use large language models in writing.

## A    EXPERIMENTAL DETAILS

### A.1    DATASETS

The Table 6 provides a summary of the statistics for the publicly available real-world datasets (Liu et al., 2024b; Dong et al., 2024; Miao et al., 2024). To ensure broad coverage and initial relevance of exogenous text to time series, text sources are collected through web search and targeted crawling, combining widely sourced online content with domain-related reports. To ensure semantic relevance between exogenous text and time-series data, 2–3 domain-specific keywords are defined for each dataset and used for web search. For each keyword, the top-ranked search results are collected, and structured information such as timestamp, source, title, and content is extracted. For report-based sources (e.g., governmental or institutional reports), all available documents are parsed, and only the sections whose topics match the corresponding domain characteristics are retained. To prevent future ground-truth leakage, two safeguards are applied. First, all collected texts contain explicit timestamps, ensuring that no future documents are matched with past observations. Second, each text is further separated into historical factual statements and predictive descriptions, and only factual content is retained. This prevents any predicted future outcomes or implicitly revealed future values from leaking into the model. To comprehensively evaluate the performance of MindTS, we evaluate 6 real-world datasets that cover 4 domains. The anomaly ratio varies from 5.81% to 17.23%, the range of feature dimensions varies from 1 to 9, and the sequence length varies from 1622 to 15981. Exogenous texts are often collected from diverse sources such as official reports and news articles, which tend to focus on background context or general conditions. We refer to such text as background text, and the majority of the datasets we used also fall into this category. Therefore, our work focuses on this type of text. Temporal alignment is achieved through binary timestamps that mark the start and end dates of each text (Liu et al., 2024b). This provides a feasible and realistic alignment strategy. For example, in the energy dataset, exogenous texts are energy reports from the U.S. Energy Information Administration, which describe contextual factors such as market demand and economic cycles. Similar background-oriented alignment is common in industrial monitoring and other domains. Importantly, our model does not heavily rely on strict temporal alignment. MindTS can operate under window-level matching, making it applicable to practical scenarios where text is loosely or sparsely aligned with time series.

To investigate whether MindTS can be extended to broader multimodal time series tasks, we also evaluate its forecasting performance on three widely used benchmark datasets covering agriculture, climate, and social good Liu et al. (2024b). Detailed statistics are provided in Table 7.

Table 6: Statistics and descriptions of datasets used for multimodal time series anomaly detection. AR (%) denotes anomaly ratio.

| Dataset | Dim | AR (%) | Avg Total Length | Timespan (start - end) | Description |
|---|---|---|---|---|---|
| Weather | 4 | 17.10 | 12339 | 2012.7.17 - 2023.10.20 | Temperature and humidity statistics and reports collected from government websites. |
| Energy | 9 | 17.23 | 1622 | 1993.4.5 - 2024.4.29 | Gasoline price statistics and energy reports are collected from the U.S. Energy Information Administration. |
| Environment | 1 | 5.81 | 15981 | 1980.1.1 - 2023.9.30 | Air Quality Index data and related reports collected from the U.S. Environmental Protection Agency and NBC. |
| KR | 1 | 6.21 | 2655 | 2009.9.15 - 2020.4.1 | Financial datasets include numerical stock data from Yahoo |
| EWJ | 1 | 9.96 | 2658 | 2009.11.16 - 2020.6.9 | Finance and news information collected from various financial |
| MDT | 1 | 11.17 | 2732 | 2009.8.6 - 2020.6.12 | news websites such as NASDAQ, Bloomberg, and others. |

### A.2    BASELINES

We extensively compare MindTS against 19 baselines, including (1) LLM-based methods: LLMMixer (LMixer) (Kowsher et al., 2024), UniTime (UTime) (Liu et al., 2024c), GPT4TS (G4TS) (Zhou et al., 2023), CALF (Liu et al., 2025b); (2) Pre-trained methods: DADA (Shentu

Table 7: The statistics of evaluation datasets for the forecasting task.

| Dataset | Dim | Prediction Length | Avg Total Length | Timespan (start - end) | Description |
|---|---|---|---|---|---|
| Agriculture | 1 | {6, 8, 10, 12} | 496 | 1983-2024 | Retail Broiler Composite. |
| Climate | 5 | {6, 8, 10, 12} | 496 | 1983-2024 | Drought Level. |
| SocialGood | 1 | {6, 8, 10, 12} | 900 | 1950-2024 | Unemployment Rate. |

et al., 2025), Timer (Liu et al., 2024d), UniTS (Gao et al., 2024); (3) Deep learning-based methods: ModernTCN (Modern) (Luo & Wang, 2024), TimesNet (TsNet) (Wu et al., 2023), DCdetector (DC) (Yang et al., 2023c), Anomaly Transformer (A.T.) (Xu et al., 2021), PatchTST (Patch) (Nie et al., 2022), TranAD (Tuli et al., 2022), DualTF (Nam et al., 2024), iTransformer (iTrans) (Liu et al., 2023); (4) Non-learning methods: PCA (Shyu et al., 2003), IForest (IF) (Liu et al., 2008), LODA (Pevnỳ, 2016), HBOS (Goldstein & Dengel, 2012).

- LLMMixer (LMixer) (Kowsher et al., 2024): Incorporates multi-scale decomposition of time series data and leverages pre-trained LLMs to process both multi-scale signals and textual prompts, effectively utilizing the semantic knowledge of LLMs for comprehensive temporal analysis.

- UniTime (UTime) (Liu et al., 2024c): Focuses on prompt engineering by introducing learnable prompts, prompt pools, and domain-specific instructions to elicit domain-relevant temporal knowledge from large language models.

- GPT4TS (G4TS) (Zhou et al., 2023): Adopts selective fine-tuning of key LLM components such as positional encodings and layer normalization, enabling efficient adaptation to time series data while retaining most of the model's pre-trained capabilities.

- CALF (Liu et al., 2025b): Proposes a cross-modal fine-tuning framework that mitigates distributional discrepancies between the temporal prediction and the aligned textual source branches, enhancing alignment across modalities.

- DADA (Shentu et al., 2025): Develops a general-purpose anomaly detection model for time series, pre-trained on a wide range of domains and readily adaptable to various downstream tasks.

- Timer (Liu et al., 2024d): Unifies heterogeneous time series into a single sequence and performs predictive anomaly detection using a sequence modeling approach.

- UniTS (Gao et al., 2024): Transforms multiple tasks into a unified token-based representation using a prompt-based framework. For anomaly detection, it generates masked tokens and utilizes denoised outputs to identify anomalies.

- ModernTCN (Modern) (Luo & Wang, 2024): Adopts a purely convolutional architecture to decouple and model temporal, channel-wise, and variable-wise relationships in multivariate time series.

- TimesNet (TsNet) (Wu et al., 2023): Employs a modular structure to decompose complex temporal patterns into different frequency components and maps one-dimensional time series into a two-dimensional space to jointly model intra- and inter-period dynamics.

- DCdetector (DC) (Yang et al., 2023c): Uses contrastive learning from both patch-wise and point-wise perspectives to discriminate between normal and anomalous patterns.

- Anomaly Transformer (A.T.) (Xu et al., 2021): Based on the hypothesis that anomalies exhibit stronger associations with nearby time points, it uses a minimax strategy to amplify association differences and enhance anomaly discrimination.

- PatchTST (Patch) (Nie et al., 2022): Applies channel-independent patching to multivariate time series, improving the model's ability to capture localized temporal features.

- TranAD (Tuli et al., 2022): A Transformer-based Anomaly Detection Model that amplifies reconstruction error through adversarial training.

- DualTF (Nam et al., 2024): Employs a dual-domain architecture with nested sliding windows, where outer and inner windows handle time and frequency domains, respectively, aligning their anomaly scores to enhance detection.

- iTransformer (iTrans) (Liu et al., 2023): Embeds time information into variable tokens and applies attention mechanisms to model multivariate correlations.

- PCA (Shyu et al., 2003): Detects anomalies by measuring the deviation of samples in the principal component space, assuming anomalies lie far from the normal data distribution.

- IForest (IF) (Liu et al., 2008): Detects anomalies by explicitly isolating them through recursive partitioning rather than modeling normal behavior.

- LODA (Pevnỳ, 2016): Approximates joint distributions using multiple one-dimensional histograms to identify outliers.

- HBOS (Goldstein & Dengel, 2012): An unsupervised histogram-based anomaly detection method.

## A.3 METRICS

This subsection introduces the metrics used in this study, which are mainly categorized into two types (Qiu et al., 2025b). The first is label-based metrics, including Affiliated Precision (Aff-P), Affiliated Recall (Aff-R), and Affiliated F1-score (Aff-F) (Huet et al., 2022), Accuracy (Acc), Precision (P), Recall (R), F1-score (F1), Range Precision (R-P), Range Recall (R-R), and Range F1-score (R-F) (Tatbul et al., 2018). The second is score-based metrics, including the Area Under the Precision-Recall Curve (A-P) (Davis & Goadrich, 2006), the Area Under the Receiver Operating Characteristic Curve (A-R) (Fawcett, 2006), the Range Area Under the Precision-Recall Curve (R-A-P), the Range Area Under the Receiver Operating Characteristic Curve (R-A-R) (Paparrizos et al., 2022), the Volume Under the Precision-Recall Surface (V-PR), and the Volume Under the Receiver Operating Characteristic Surface (V-ROC) (Paparrizos et al., 2022). MindTS evaluates all metrics to assess each method's performance.

## A.4 IMPLEMENTATION DETAILS

We adhere to the evaluation protocol proposed in TFB (Qiu et al., 2024) during testing by disabling the "drop last" operation, ensuring a fair comparison across all models. We conduct experiments using Pytorch with NVIDIA Tesla-A800-80GB GPUs. We employ the Adam optimizer (Kingma & Ba, 2015) during training. All baselines are based on our runs, using the identical hardware. We employ official or open-source implementations published on GitHub and follow the configurations recommended in their papers. The initial batch size is 64, which can be halved (down to a minimum of 8) if an Out-Of-Memory (OOM) error occurs. We assign equal weights of 1 to optimization objectives. Experiments show that this sample configuration delivers stable and competitive performance, indicating that additional tuning is unnecessary.

# B LEMMA PROOFS

In this section, we explain the proof of Equation equation 8 in our paper.

*Proof.* The definition of mutual information is described as

$$I(\mathbf{Z}_{\text{text}}; \mathbf{Z}_{\text{con}}) = \sum_{\mathbf{Z}_{\text{text}}} \sum_{\mathbf{Z}_{\text{con}}} \mathbb{P}(\mathbf{Z}_{\text{text}}, \mathbf{Z}_{\text{con}}) \log \frac{\mathbb{P}(\mathbf{Z}_{\text{text}}, \mathbf{Z}_{\text{con}})}{\mathbb{P}(\mathbf{Z}_{\text{text}}) \mathbb{P}(\mathbf{Z}_{\text{con}})}$$
$$= \mathbb{E}_{\mathbf{Z}_{\text{text}}, \mathbf{z}_{\text{con}}} \left[ \log \frac{\mathbb{P}(\mathbf{Z}_{\text{text}}, \mathbf{Z}_{\text{con}})}{\mathbb{P}(\mathbf{Z}_{\text{text}}) \mathbb{P}(\mathbf{Z}_{\text{con}})} \right]. \tag{13}$$

Then, by introducing a variational approximation $\mathbb{G}(\mathbf{Z}_{\text{con}})$, we can further derive that

$$\mathbb{E}_{\mathbf{Z}_{\text{text}}, \mathbf{z}_{\text{con}}} \left[ \log \frac{\mathbb{P}(\mathbf{Z}_{\text{text}}, \mathbf{Z}_{\text{con}})}{\mathbb{P}(\mathbf{Z}_{\text{text}}) \mathbb{P}(\mathbf{Z}_{\text{con}})} \right]$$
$$= \mathbb{E}_{\mathbf{Z}_{\text{text}}, \mathbf{z}_{\text{con}}} \left[ \log \frac{\mathbb{P}(\mathbf{Z}_{\text{con}} | \mathbf{Z}_{\text{text}})}{\mathbb{P}(\mathbf{Z}_{\text{con}})} \right] \tag{14}$$
$$= \mathbb{E}_{\mathbf{Z}_{\text{text}}, \mathbf{z}_{\text{con}}} \left[ \log \frac{\mathbb{P}(\mathbf{Z}_{\text{con}} | \mathbf{Z}_{\text{text}})}{\mathbb{G}(\mathbf{Z}_{\text{con}})} + \log \frac{\mathbb{G}(\mathbf{Z}_{\text{con}})}{\mathbb{P}(\mathbf{Z}_{\text{con}})} \right].$$

Based on the KL-divergence, the second term of the above equation can be rewritten as

$$\mathbb{E}_{\mathbf{Z}_{\text{text}}, \mathbf{z}_{\text{con}}} \left[ \log \frac{\mathbb{G}(\mathbf{Z}_{\text{con}})}{\mathbb{P}(\mathbf{Z}_{\text{con}})} \right] = \mathbb{E}_{\mathbf{Z}_{\text{text}} | \mathbf{z}_{\text{con}}} \left[ \mathbb{P}(\mathbf{Z}_{\text{con}}) \log \frac{\mathbb{G}(\mathbf{Z}_{\text{con}})}{\mathbb{P}(\mathbf{Z}_{\text{con}})} \right]$$
$$= -\mathbb{E}_{\mathbf{Z}_{\text{text}} | \mathbf{z}_{\text{con}}} \left[ \mathbb{P}(\mathbf{Z}_{\text{con}}) \log \frac{\mathbb{P}(\mathbf{Z}_{\text{con}})}{\mathbb{G}(\mathbf{Z}_{\text{con}})} \right] \tag{15}$$
$$= -\mathbb{E}_{\mathbf{Z}_{\text{text}} | \mathbf{z}_{\text{con}}} \left[ \text{KL}(\mathbb{P}(\mathbf{Z}_{\text{con}}) || \mathbb{G}(\mathbf{Z}_{\text{con}})) \right].$$

Such that

$$I(\mathbf{Z}_{\text{text}}; \mathbf{Z}_{\text{con}}) = \mathbb{E}_{\mathbf{Z}_{\text{text}}} \left[ \text{KL}(\mathbb{P}(\mathbf{Z}_{\text{con}} | \mathbf{Z}_{\text{text}}) || \mathbb{G}(\mathbf{Z}_{\text{con}})) \right] - \mathbb{E}_{\mathbf{Z}_{\text{text}} | \mathbf{z}_{\text{con}}} \left[ \text{KL}(\mathbb{P}(\mathbf{Z}_{\text{con}}) || \mathbb{G}(\mathbf{Z}_{\text{con}})) \right]$$
$$\leq \mathbb{E}_{\mathbf{Z}_{\text{text}}} \left[ \text{KL}(\mathbb{P}(\mathbf{Z}_{\text{con}} | \mathbf{Z}_{\text{text}}) || \mathbb{G}(\mathbf{Z}_{\text{con}})) \right] \tag{16}$$

It should be noted that $\mathbb{G}(\mathbf{Z}_{\text{con}})$ is a variational approximation, such that the distribution $\mathbb{G}(\mathbf{Z}_{\text{con}})$ can approximate $\mathbb{P}(\mathbf{Z}_{\text{con}})$ in the process of optimization, that is $\mathbb{G}(\mathbf{Z}_{\text{con}}) = \mathbb{P}(\mathbf{Z}_{\text{con}})$. In this case, $I(\mathbf{Z}_{\text{text}}; \mathbf{Z}_{\text{con}}) = \mathbb{E}_{\mathbf{Z}_{\text{text}}} \left[ \text{KL}(\mathbb{P}(\mathbf{Z}_{\text{con}} | \mathbf{Z}_{\text{text}}) || \mathbb{G}(\mathbf{Z}_{\text{con}})) \right]$, and the upper bound presented in Lemma 1 is tight. The proof is completed. $\square$

## C  ADDITIONAL EXPERIMENTAL RESULTS

In this section, we present the performance of MindTS and 19 numerical-only unimodal methods on additional evaluation metrics. Specifically, Tables 8–13 present the comparative evaluation results across the following metrics: (AUC-ROC, R-AUC-ROC, VUS-ROC), (Accuracy), (AUC-PR, R-AUC-PR, VUS-PR), (Precision, Recall, F1-score), (Range-Recall, Range-Precision, Range-F1-score), and (Affiliated-Precision, Affiliated-Recall, Affiliated-F1-score), respectively.

Tables 14, 15, 16, 17, 18 and 19 present the extension of 19 numerical-only unimodal methods into multimodal forms using the MM-TSFLib framework (Liu et al., 2024b), and compare them with MindTS across a comprehensive set of 16 evaluation metrics.

Table 8: Average A-R (AUC-ROC), R-A-R (R-AUC-ROC) and V-ROC (VUS-ROC) accuracy for MindTS and all numerical-only unimodal methods. The best results are highlighted in bold, and the second-best results are underlined.

| Datasets | Weather | | | Energy | | | Environment | | | KR | | | EWJ | | | MDT | | |
|---|---|---|---|---|---|---|---|---|---|---|---|---|---|---|---|---|---|---|
| Metric | A-R | R-A-R | V-ROC | A-R | R-A-R | V-ROC | A-R | R-A-R | V-ROC | A-R | R-A-R | V-ROC | A-R | R-A-R | V-ROC | A-R | R-A-R | V-ROC |
| HBOS | 64.47 | 54.12 | 54.16 | 60.80 | 51.06 | 51.50 | 56.42 | 50.79 | 51.03 | 75.16 | 61.80 | 61.41 | 71.82 | 61.02 | 62.07 | 60.26 | 54.86 | 55.30 |
| LODA | 69.67 | 56.88 | 57.00 | 59.54 | 56.22 | 55.90 | 58.44 | 50.54 | 50.69 | 73.74 | 60.27 | 59.99 | 71.40 | 60.60 | 61.65 | 66.69 | 55.56 | 55.98 |
| IF | 67.81 | 56.69 | 56.45 | 60.32 | 52.64 | 53.61 | 52.37 | 45.96 | 46.20 | 74.45 | 61.10 | 60.70 | 69.20 | 57.94 | 59.24 | 63.92 | 53.52 | 54.02 |
| PCA | 67.17 | 57.80 | 57.38 | 61.14 | 52.64 | 53.07 | 48.60 | 35.71 | 37.08 | 63.58 | 51.01 | 47.51 | 54.35 | 43.78 | 45.26 | 54.51 | 41.90 | 44.09 |
| iTrans | 41.22 | 74.45 | 73.37 | 65.60 | 64.25 | 63.06 | 82.35 | 70.98 | 73.81 | 83.22 | 74.41 | 76.12 | 76.37 | 68.29 | 72.16 | 79.32 | 67.65 | 71.87 |
| DulTF | 64.49 | 28.95 | 57.84 | 49.90 | 38.00 | 38.36 | 46.64 | 31.84 | 6.30 | 68.77 | 55.45 | 54.51 | 74.12 | 60.49 | 64.31 | 75.13 | 60.50 | 63.38 |
| TranAD | **85.51** | 79.38 | 78.75 | 67.01 | 56.65 | 56.37 | 26.32 | 9.85 | 14.20 | 60.64 | 37.02 | 41.05 | 60.35 | 44.11 | 49.60 | 44.10 | 24.47 | 28.55 |
| Patch | 82.02 | 80.47 | 79.97 | 66.70 | 61.39 | 58.31 | 94.17 | 91.12 | 90.86 | 82.15 | 72.72 | 74.65 | 78.53 | 69.26 | 71.56 | 84.55 | 75.61 | 77.69 |
| A.T. | 47.11 | 43.11 | 45.02 | 38.68 | 31.52 | 31.56 | 61.88 | 51.42 | 51.98 | 51.25 | 40.18 | 41.97 | 43.81 | 27.50 | 31.75 | 56.44 | 41.41 | 44.53 |
| DC | 47.90 | 45.41 | 45.56 | 48.75 | 45.39 | 45.93 | 54.98 | 39.34 | 41.28 | 52.97 | 41.75 | 43.04 | 53.40 | 45.69 | 47.10 | 53.82 | 43.65 | 45.02 |
| TsNet | 81.10 | 83.11 | 82.30 | 71.36 | 61.56 | 59.47 | 91.84 | 87.56 | 87.97 | 85.88 | 78.29 | 79.00 | 82.39 | 74.22 | 75.76 | 86.67 | 77.01 | 79.56 |
| Modern | 80.66 | 82.36 | 81.14 | 70.80 | 65.34 | 65.05 | 93.53 | 90.25 | 89.78 | 93.39 | 89.77 | 89.78 | 87.82 | **83.72** | 83.88 | 88.77 | 81.59 | 82.30 |
| G4TS | 74.47 | 71.43 | 70.03 | 66.54 | 53.54 | 53.10 | 75.79 | 63.10 | 66.79 | 78.30 | 65.15 | 67.81 | 75.58 | 64.86 | 67.95 | 74.79 | 59.00 | 62.30 |
| CALF | 70.48 | 64.67 | 63.63 | 61.56 | 59.26 | 57.35 | 59.54 | 44.91 | 57.35 | 65.09 | 48.82 | 53.22 | 67.70 | 55.57 | 59.03 | 53.13 | 35.41 | 39.81 |
| UniTS | 81.22 | 75.55 | 75.08 | 63.38 | 52.12 | 51.15 | 95.19 | 92.55 | 92.03 | 80.95 | 71.29 | 73.93 | 79.87 | 71.32 | 73.91 | 73.19 | 56.72 | 58.67 |
| Timer | 80.86 | 73.73 | 73.22 | 60.54 | 46.82 | 46.03 | 95.36 | 92.37 | 92.10 | 66.72 | 74.61 | 75.99 | 76.15 | 64.71 | 75.99 | 75.65 | 58.78 | 60.28 |
| UTime | 81.09 | 79.32 | 78.45 | 64.17 | 51.01 | 49.97 | 95.13 | 92.16 | 91.77 | 82.36 | 71.49 | 73.55 | 77.71 | 67.15 | 64.49 | 75.59 | 58.98 | 61.00 |
| LMixer | 79.60 | 72.54 | 71.71 | 61.31 | 55.25 | 53.04 | 92.99 | 87.83 | 89.75 | 65.77 | 49.01 | 52.79 | 57.69 | 41.75 | 46.80 | 60.30 | 42.44 | 47.06 |
| DADA | 66.37 | 61.95 | 61.03 | 62.33 | 55.78 | 54.37 | 70.33 | 87.27 | 54.37 | 79.53 | 69.91 | 70.82 | 79.11 | 68.44 | 71.79 | 79.04 | 63.94 | 66.76 |
| MindTS | 84.06 | **93.80** | **82.64** | **81.26** | **75.51** | **74.44** | **96.33** | **94.04** | **93.78** | **93.51** | 89.60 | **89.86** | **87.95** | 83.19 | **84.12** | **90.46** | **83.15** | **83.02** |

Table 9: Average ACC (Accuracy) measures for MindTS and all numerical-only unimodal methods. The best results are highlighted in bold, and the second-best results are underlined.

| Datasets | Weather | Energy | Environment | KR | EWJ | MDT |
|---|---|---|---|---|---|---|
| Metric | ACC | ACC | ACC | ACC | ACC | ACC |
| HBOS | 84.60 | 59.69 | **94.87** | **95.86** | 87.03 | 90.48 |
| LODA | **87.60** | 52.92 | 92.03 | 95.85 | 86.28 | 90.48 |
| IF | 85.62 | 55.38 | 84.87 | 94.54 | 86.84 | 88.10 |
| PCA | 62.72 | 59.08 | 49.27 | 82.11 | 72.18 | 65.02 |
| iTrans | 67.02 | 53.85 | 81.46 | 77.97 | 68.70 | 77.84 |
| DulTF | 53.65 | 23.08 | 42.89 | 86.44 | 85.90 | 63.92 |
| TranAD | 64.67 | 67.08 | 29.20 | 90.21 | 64.66 | 41.58 |
| Patch | 77.76 | 49.85 | 90.06 | 84.37 | 85.53 | 83.88 |
| A.T. | 67.10 | 57.54 | 90.37 | 78.53 | 50.56 | 57.33 |
| DC | 72.85 | 71.08 | 54.45 | 72.32 | 82.71 | 74.36 |
| TsNet | 81.36 | 68.00 | 87.21 | 87.95 | 80.26 | 86.45 |
| Modern | 81.28 | 51.38 | 90.72 | 89.08 | 88.16 | 84.80 |
| G4TS | 65.19 | 39.38 | 82.37 | 74.01 | 79.51 | 83.70 |
| CALF | 63.01 | 68.31 | 66.18 | 87.19 | 76.70 | 71.25 |
| UniTS | 71.80 | 45.54 | 90.97 | 87.38 | 79.14 | 85.16 |
| Timer | 66.73 | 44.92 | 90.03 | 93.79 | 81.77 | 82.42 |
| UTime | 81.12 | 61.23 | 90.06 | 91.90 | 77.26 | 85.71 |
| LMixer | 65.76 | 45.23 | 89.97 | 76.27 | 71.43 | 65.57 |
| DADA | 58.39 | 59.38 | 94.19 | 92.47 | 85.71 | 87.18 |
| MindTS | 84.76 | **80.00** | 90.09 | 91.15 | **88.91** | **93.96** |

Table 10: Average A-P (AUC-PR), R-A-P (R-AUC-PR) and V-PR (VUS-PR) accuracy measures for MindTS and all numerical-only unimodal methods. The best results are highlighted in bold, and the second-best results are underlined.

| Datasets | Weather | | | Energy | | | Environment | | | KR | | | EWJ | | | MDT | | |
|---|---|---|---|---|---|---|---|---|---|---|---|---|---|---|---|---|---|---|
| Metric | A-P | R-A-P | V-PR | A-P | R-A-P | V-PR | A-P | R-A-P | V-PR | A-P | R-A-P | V-PR | A-P | R-A-P | V-PR | A-P | R-A-P | V-PR |
| HBOS | 31.16 | 46.37 | 46.58 | 21.55 | 42.14 | 42.57 | 16.97 | 49.84 | 50.30 | 41.09 | 51.69 | 52.06 | 25.24 | 38.48 | 41.19 | 28.66 | 43.16 | 44.77 |
| LODA | 41.22 | 54.75 | 55.03 | 20.75 | 48.94 | 48.63 | 9.98 | 18.19 | 18.66 | 40.14 | 51.31 | 51.82 | 24.16 | 37.46 | 40.08 | 29.81 | 43.18 | 44.63 |
| IF | 35.44 | 49.65 | 49.66 | 21.17 | 45.19 | 46.03 | 6.18 | 8.28 | 8.94 | 32.21 | 43.07 | 43.31 | 22.86 | 34.74 | 37.81 | 22.41 | 33.63 | 35.33 |
| PCA | 25.02 | 47.47 | 47.13 | 21.69 | 43.89 | 44.30 | 5.67 | 16.05 | 17.87 | 10.18 | 18.99 | 22.13 | 10.99 | 16.49 | 19.37 | 12.29 | 19.53 | 22.93 |
| iTrans | 41.22 | 42.93 | 42.56 | 35.63 | 35.71 | 35.82 | 34.90 | 23.09 | 24.87 | 36.05 | 24.95 | 27.37 | 34.42 | 24.99 | 28.98 | 46.91 | 32.94 | 36.36 |
| DulTF | 25.22 | 28.95 | 29.27 | 22.58 | 22.94 | 23.52 | 5.35 | 5.47 | 6.53 | 21.96 | 17.73 | 17.92 | 42.45 | 31.17 | 33.75 | 39.84 | 31.52 | 33.83 |
| TranAD | **60.90** | 52.04 | 52.08 | 36.38 | 33.17 | 33.80 | 7.09 | 4.49 | 4.91 | 53.23 | 28.04 | 28.42 | 27.85 | 15.20 | 17.80 | 25.69 | 13.11 | 14.33 |
| Patch | 53.39 | 49.81 | 50.03 | 34.25 | 35.25 | 34.41 | 58.92 | 45.65 | 45.78 | 53.60 | 35.32 | 36.18 | 47.91 | 33.37 | 36.08 | 54.11 | 39.70 | 41.67 |
| A.T. | 16.71 | 18.85 | 19.17 | 14.02 | 19.24 | 19.69 | 14.06 | 16.22 | 18.14 | 7.01 | 6.44 | 7.94 | 8.97 | 9.01 | 10.85 | 15.02 | 13.20 | 15.93 |
| DC | 17.08 | 18.06 | 18.33 | 17.69 | 21.77 | 22.57 | 6.48 | 6.55 | 7.69 | 8.10 | 7.04 | 8.49 | 10.88 | 12.52 | 15.37 | 11.59 | 13.30 | 15.72 |
| TsNet | 47.65 | 50.58 | 50.09 | 42.05 | 38.17 | 38.61 | 64.14 | 50.62 | 50.64 | 67.47 | **52.83** | 51.60 | 54.99 | 41.84 | 43.15 | 65.57 | 48.60 | 50.54 |
| Modern | 51.98 | 52.67 | 52.13 | 33.16 | 35.64 | 36.60 | 55.34 | 42.50 | 42.26 | 56.93 | 40.69 | 39.95 | 53.36 | 43.86 | 44.75 | 65.48 | 51.72 | 52.18 |
| G4TS | 44.12 | 41.37 | 41.30 | 33.75 | 31.10 | 31.68 | 35.37 | 22.14 | 23.94 | 56.78 | 37.53 | 38.23 | 46.75 | 32.83 | 35.63 | 60.40 | 42.48 | 44.81 |
| CALF | 37.38 | 34.98 | 35.07 | 32.32 | 32.61 | 33.49 | 9.19 | 7.49 | 8.96 | 25.26 | 13.25 | 16.04 | 22.74 | 17.53 | 20.66 | 16.25 | 12.54 | 15.15 |
| UniTS | 49.19 | 44.31 | 44.35 | 27.51 | 30.70 | 31.04 | 64.13 | 50.55 | 50.24 | 55.39 | 40.75 | 43.32 | 50.33 | 36.79 | 39.32 | 53.44 | 36.14 | 37.61 |
| Timer | 48.87 | 43.20 | 43.21 | 38.05 | 28.81 | 29.46 | 64.52 | 51.17 | 51.42 | 66.72 | 51.59 | 51.41 | 44.01 | 30.67 | 33.17 | 55.86 | 37.54 | 38.38 |
| UTime | 54.67 | 52.18 | 51.90 | 37.96 | 32.25 | 32.88 | 62.59 | 48.80 | 48.36 | 64.46 | 46.87 | — | 43.43 | 30.31 | 32.39 | 56.00 | 37.32 | 38.94 |
| LMixer | 49.71 | 43.40 | 43.47 | 32.85 | 30.59 | 30.35 | 64.49 | 49.95 | 52.94 | 28.19 | 13.25 | 15.13 | 18.81 | 12.36 | 15.21 | 19.86 | 15.30 | 19.10 |
| DADA | 29.80 | 29.86 | 30.00 | 37.81 | 33.47 | 34.18 | **70.33** | 55.96 | 56.20 | 63.55 | 46.95 | 45.90 | 55.24 | 41.61 | 43.36 | 63.03 | 44.63 | 46.81 |
| MindTS | 58.38 | **57.85** | **57.48** | **50.99** | **50.19** | **50.36** | 69.46 | **56.81** | **56.79** | **67.52** | 52.64 | **53.15** | **61.75** | **49.31** | **50.42** | **76.51** | **66.2** | **65.44** |

Table 11: Average P (Precision), R (Recall) and F1 (F1-score) accuracy measures for MindTS and all numerical-only unimodal methods. The best results are highlighted in bold, and the second-best results are underlined.

| Datasets | Weather | | | Energy | | | Environment | | | KR | | | EWJ | | | MDT | | |
|---|---|---|---|---|---|---|---|---|---|---|---|---|---|---|---|---|---|---|
| Metric | P | R | F1 | P | R | F1 | P | R | F1 | P | R | F1 | P | R | F1 | P | R | F1 |
| HBOS | 58.61 | 33.89 | 42.94 | 24.14 | 62.50 | 34.83 | **92.31** | 12.90 | 22.64 | 73.91 | 51.52 | 60.71 | 38.89 | 52.83 | 44.80 | 64.52 | 32.79 | 43.48 |
| LODA | 73.97 | 42.42 | 53.92 | 22.29 | 69.64 | 33.77 | 26.21 | 20.43 | 22.96 | **76.19** | 48.48 | 59.26 | 36.84 | 65.73 | 43.41 | 62.86 | 36.07 | 45.83 |
| IF | 62.09 | 40.76 | 49.21 | 23.03 | 64.29 | 34.39 | 8.15 | 15.59 | 10.70 | 56.67 | 51.52 | 53.97 | 37.31 | 47.17 | 41.67 | 45.45 | 32.79 | 38.10 |
| PCA | 27.81 | 73.93 | 40.41 | 24.16 | **96.43** | 35.12 | 5.51 | 47.85 | 9.88 | 13.18 | 32.07 | 18.68 | 13.89 | 40.98 | 20.75 | — | — | — |
| iTrans | 30.48 | 72.51 | 42.92 | 22.67 | 67.86 | 34.21 | 19.67 | 70.96 | 30.81 | 18.84 | 78.79 | 30.41 | 22.44 | 66.04 | 33.49 | 30.00 | 73.33 | 42.65 |
| DulTF | 17.27 | **99.53** | 29.43 | 17.76 | 46.43 | 30.00 | 5.43 | 53.76 | 9.87 | 5.82 | 81.82 | 10.87 | 36.90 | 58.49 | 45.26 | 31.50 | 65.57 | 42.55 |
| TranAD | 31.65 | 91.94 | 47.08 | 25.24 | 69.64 | 32.70 | 2.29 | 26.88 | 4.28 | 17.48 | 54.55 | 26.47 | 13.02 | 47.17 | 20.41 | 25.46 | 44.26 | 14.48 |
| Patch | 37.50 | 73.22 | 49.60 | 21.96 | 83.93 | 34.81 | 34.36 | 77.96 | 47.70 | 24.49 | 72.73 | 36.64 | 36.36 | 60.38 | 45.39 | 39.05 | 67.21 | 49.40 |
| A.T. | 13.89 | 17.78 | 15.59 | 9.02 | 17.86 | 12.50 | 23.48 | 29.03 | 26.28 | 10.30 | 30.30 | 11.90 | 8.55 | 43.40 | 14.39 | 14.00 | 45.90 | 25.46 |
| DC | 12.65 | 9.95 | 11.14 | 15.38 | 10.71 | 12.63 | 7.07 | 54.84 | 12.53 | 7.46 | 30.30 | 11.98 | 15.63 | 18.87 | 17.09 | 15.04 | 27.87 | 19.54 |
| TsNet | 46.40 | 58.06 | 51.58 | 21.40 | 82.14 | 33.95 | 29.24 | 84.41 | 43.43 | 47.17 | 75.76 | 58.14 | 37.14 | **73.58** | 49.37 | 43.69 | 73.77 | 54.88 |
| Modern | 46.48 | 62.56 | 53.33 | 29.63 | 71.43 | 41.88 | 35.66 | 74.19 | 48.16 | 33.77 | 78.79 | 47.27 | 44.05 | 69.81 | 54.01 | 40.68 | 78.69 | 53.63 |
| G4TS | 27.80 | 72.27 | 40.16 | 20.70 | 83.93 | 33.22 | 14.46 | 69.89 | 23.96 | 16.98 | **81.82** | **74.01** | 43.28 | 54.72 | 48.33 | 37.72 | 70.49 | 49.14 |
| CALF | 25.70 | 69.19 | 37.48 | 26.67 | 50.00 | 34.78 | 7.97 | 45.78 | 13.58 | 23.08 | 45.45 | 30.61 | 22.22 | 56.60 | 31.94 | 18.68 | 27.87 | 22.39 |
| UniTS | 35.88 | 82.46 | 50.00 | 20.20 | 73.21 | 31.66 | 35.96 | 83.33 | 50.24 | 30.23 | 79.79 | 30.23 | 26.95 | 71.70 | 39.18 | 44.19 | 62.30 | 51.70 |
| Timer | 31.79 | 86.02 | 46.42 | 20.53 | 69.64 | 31.71 | 38.13 | 89.78 | 53.53 | 45.00 | 81.82 | 58.04 | 30.36 | 64.15 | 41.21 | 36.00 | 73.77 | 48.39 |
| UTime | 45.99 | 59.72 | 51.96 | 18.75 | 69.64 | 29.55 | 35.27 | 84.94 | 49.80 | 41.94 | 78.79 | 54.74 | 26.71 | 73.58 | 39.20 | 45.12 | 60.66 | 51.75 |
| LMixer | 29.14 | 82.94 | 43.13 | 20.95 | 78.57 | 33.08 | 32.59 | 86.55 | 47.35 | 12.80 | 48.49 | 20.25 | 11.64 | 50.94 | 18.95 | 18.82 | 52.46 | 27.71 |
| DADA | 24.03 | 66.35 | 35.29 | 20.10 | 73.21 | 31.54 | 50.00 | 77.96 | **60.92** | 37.50 | 72.72 | 49.48 | 38.14 | 69.81 | 49.33 | 44.83 | 63.93 | 53.70 |
| MindTS | 54.77 | 62.56 | **58.41** | **45.16** | 75.00 | **56.38** | 36.21 | **92.47** | 52.04 | 39.40 | 78.79 | 52.53 | **46.34** | 71.70 | **56.30** | **69.44** | **81.97** | **75.19** |

Table 12: Average R-R (Range-Recall), R-P (Range-Precision) and R-F (Range-F1-score) accuracy measures for MindTS and all numerical-only unimodal methods. The best results are highlighted in bold, and the second-best results are underlined.

| Datasets | Weather | | | Energy | | | Environment | | | KR | | | EWJ | | | MDT | | |
|---|---|---|---|---|---|---|---|---|---|---|---|---|---|---|---|---|---|---|
| Metric | R-R | R-P | R-F | R-R | R-P | R-F | R-R | R-P | R-F | R-R | R-P | R-F | R-R | R-P | R-F | R-R | R-P | R-F |
| HBOS | 22.51 | 36.88 | 27.96 | 57.14 | 1.51 | 2.94 | 10.50 | **92.11** | 18.85 | 48.15 | 72.55 | 57.88 | 58.18 | **53.72** | 55.86 | 34.62 | **89.91** | 49.99 |
| LODA | 27.78 | **70.30** | 39.82 | 67.86 | 20.64 | 31.65 | 19.39 | 31.55 | 24.02 | 44.44 | **77.38** | 56.46 | 58.18 | 47.86 | 52.52 | 35.77 | 86.74 | 50.65 |
| IF | 28.08 | 50.24 | 36.02 | 64.29 | 5.13 | 9.50 | 15.27 | 10.16 | 12.20 | 48.15 | 53.03 | 50.47 | 51.36 | 50.52 | 50.94 | 33.85 | 63.78 | 44.22 |
| PCA | 61.71 | 11.98 | 20.07 | 60.71 | 1.42 | 2.78 | 44.68 | 5.45 | 9.71 | 34.09 | 15.58 | 21.38 | 37.31 | 16.47 | 22.86 | — | — | — |
| DulTF | **93.99** | 12.08 | 21.41 | **95.00** | 6.34 | 11.89 | 54.37 | 4.43 | 8.19 | 74.81 | 9.35 | 16.62 | 64.09 | 42.20 | 50.89 | 68.85 | 37.61 | 48.65 |
| iTrans | 87.59 | 23.28 | 36.79 | 73.10 | 19.49 | 30.78 | 70.35 | 19.32 | 30.31 | 81.48 | 15.01 | 25.35 | 74.55 | 20.41 | 32.05 | 74.62 | 27.44 | 40.13 |
| TranAD | 84.84 | 24.02 | 37.44 | 42.86 | 6.25 | 10.90 | 24.63 | 4.45 | 7.54 | 50.37 | 32.66 | 39.62 | 51.36 | 14.74 | 22.78 | 46.54 | 18.94 | 26.92 |
| Patch | 62.83 | 34.29 | 44.37 | 76.67 | 11.28 | 19.67 | 78.77 | 37.10 | 50.44 | 76.30 | 23.26 | 35.65 | 65.00 | 40.67 | 50.03 | 66.73 | 41.78 | 51.39 |
| A.T. | 22.82 | 14.72 | 17.92 | 22.86 | 10.72 | 14.60 | 27.99 | 24.77 | 26.28 | 31.85 | 5.99 | 10.08 | 45.91 | 5.53 | 9.87 | 45.00 | 10.34 | 16.81 |
| DC | 13.57 | 14.97 | 14.24 | 16.71 | 16.96 | 16.84 | 53.05 | 5.53 | 10.09 | 24.44 | 6.56 | 10.34 | 16.18 | 10.87 | 13.00 | 26.54 | 14.58 | 18.82 |
| TsNet | 69.90 | 41.45 | 52.04 | 74.05 | 13.45 | 22.76 | 84.35 | 26.93 | 40.82 | 80.00 | 53.92 | 64.42 | 79.55 | 38.08 | 51.51 | 71.54 | 44.49 | 54.86 |
| Modern | 72.13 | 39.46 | 51.01 | 75.43 | 21.64 | 33.63 | 76.17 | 38.27 | 50.94 | 81.23 | 41.04 | 54.53 | 77.73 | 49.72 | 60.64 | 79.04 | 42.01 | 54.86 |
| G4TS | 76.08 | 21.70 | 33.77 | 78.57 | 17.43 | 16.08 | 63.97 | 23.81 | 34.71 | 85.19 | 22.85 | 36.04 | 68.64 | 38.78 | 49.56 | 72.31 | 39.02 | 50.69 |
| CALF | 61.43 | 21.01 | 31.31 | 53.00 | 19.44 | 28.45 | 46.48 | 6.88 | 11.99 | 51.85 | 14.05 | 22.10 | 61.73 | 21.85 | 32.28 | 36.54 | 16.70 | 22.93 |
| UniTS | 79.39 | 30.15 | 43.70 | 65.00 | 10.50 | 18.09 | 84.11 | 37.00 | 51.39 | **85.93** | 30.37 | 44.88 | 80.91 | 27.32 | 40.85 | 66.15 | 38.93 | 49.01 |
| Timer | 79.51 | 25.20 | 38.27 | 60.71 | 13.12 | 21.57 | 88.27 | 39.86 | 54.92 | 83.70 | **55.56** | **66.78** | 70.91 | 36.96 | 48.59 | 73.08 | 43.00 | 54.14 |
| UTime | 66.97 | 40.38 | 50.38 | 59.76 | 12.61 | 20.83 | 83.31 | 37.00 | 51.25 | 83.70 | 47.81 | 60.86 | **80.91** | 34.29 | 48.16 | 52.89 | — | — |
| LMixer | 70.37 | 22.38 | 33.95 | 77.14 | 12.82 | 21.99 | 85.77 | 30.81 | 45.34 | 48.89 | 11.35 | 18.43 | 51.27 | 10.52 | 17.46 | 52.69 | 15.20 | 23.59 |
| DADA | 65.03 | 20.32 | 30.97 | 75.00 | 21.13 | 32.47 | 79.15 | 48.18 | **59.90** | 74.81 | 49.54 | 59.61 | 75.45 | 41.99 | 53.95 | 64.81 | 50.50 | 56.77 |
| MindTS | 66.32 | 45.71 | **54.12** | 71.19 | **34.09** | **46.10** | **92.43** | 35.96 | 51.77 | 83.95 | 42.98 | 56.85 | 80.91 | 49.01 | **61.04** | **82.88** | 70.28 | **76.06** |

Table 13: Average Aff-P (Affiliated-Precision), Aff-R (Affiliated-Recall) and Aff-F (Affiliated-F1score) accuracy measures for MindTS and all numerical-only unimodal methods. The best results are highlighted in bold, and the second-best results are underlined.

| Datasets | Weather | | | Energy | | | Environment | | | KR | | | EWJ | | | MDT | | |
|---|---|---|---|---|---|---|---|---|---|---|---|---|---|---|---|---|---|---|
| Metric | Aff-P | Aff-R | Aff-F | Aff-P | Aff-R | Aff-F | Aff-P | Aff-R | Aff-F | Aff-P | Aff-R | Aff-F | Aff-P | Aff-R | Aff-F | Aff-P | Aff-R | Aff-F |
| HBOS | 74.47 | 35.09 | 47.70 | 54.62 | 57.14 | 55.85 | **94.43** | 12.61 | 22.25 | 89.44 | 50.78 | 64.78 | **80.15** | 64.76 | 71.03 | 91.16 | 36.30 | 52.33 |
| LODA | **81.60** | 38.78 | 52.55 | 54.15 | 76.60 | 63.45 | 71.07 | 34.49 | 46.45 | **96.99** | 44.44 | 60.96 | 79.74 | 65.73 | 72.06 | **91.26** | 39.42 | 55.56 |
| IF | 76.25 | 41.88 | 76.25 | 54.50 | 71.96 | 62.03 | 51.31 | 42.52 | 46.51 | 86.20 | 58.05 | 69.38 | 79.18 | 58.89 | 67.55 | 87.66 | 38.75 | 53.74 |
| PCA | 59.14 | 71.93 | 64.91 | 54.88 | 60.71 | 57.65 | 46.75 | 68.64 | 55.62 | 61.94 | 44.44 | 51.75 | 53.93 | 48.48 | 51.06 | 56.98 | 52.51 | 54.66 |
| DulTF | 55.12 | **98.43** | 70.66 | **99.39** | 53.22 | 69.32 | 50.04 | 87.83 | 63.76 | 49.01 | 88.62 | 63.11 | 76.97 | 80.47 | 78.68 | 76.21 | 80.53 | 78.31 |
| iTrans | 64.98 | 96.95 | 77.81 | 57.03 | 93.37 | 70.81 | 62.84 | 91.26 | 74.43 | 69.84 | 92.23 | 79.49 | 66.91 | **94.27** | 78.27 | 69.46 | 90.65 | 78.66 |
| TranAD | 64.62 | 92.16 | 75.97 | 51.45 | 48.05 | 49.69 | 46.21 | 90.38 | 61.15 | 75.25 | 71.37 | 73.26 | 56.68 | 88.87 | 69.22 | 51.82 | 83.43 | 63.93 |
| Patch | 68.56 | 88.24 | 77.17 | 53.34 | 89.52 | 66.85 | 76.41 | 86.56 | 81.18 | 74.63 | 85.09 | 79.52 | 74.19 | 77.53 | 75.82 | 75.94 | 83.34 | 79.47 |
| A.T. | 51.98 | 46.73 | 49.22 | 50.88 | 37.89 | 43.44 | 72.95 | 50.59 | 59.75 | 64.90 | 78.34 | 70.99 | 48.69 | 73.89 | 58.70 | 61.41 | 71.50 | 66.07 |
| DC | 46.00 | 40.02 | 42.80 | 66.01 | 36.57 | 47.07 | 53.20 | 74.98 | 62.24 | 61.26 | 62.64 | 61.94 | 62.57 | 39.06 | 48.10 | 50.62 | 44.67 | 47.46 |
| TsNet | 72.72 | 90.35 | 80.58 | 53.07 | 87.26 | 66.00 | 70.05 | 94.35 | 80.41 | 86.06 | 85.51 | 85.79 | 75.06 | 89.92 | 81.82 | 76.93 | 83.50 | 80.08 |
| Modern | 72.80 | 91.45 | 81.06 | 60.89 | 84.47 | 70.76 | 77.12 | 85.44 | 81.07 | 79.60 | 89.87 | 84.42 | 78.25 | 85.33 | 81.64 | 75.60 | 86.79 | 80.81 |
| G4TS | 59.73 | 92.41 | 72.56 | 53.06 | 88.58 | 66.37 | 64.25 | 82.94 | 72.41 | 70.00 | 92.14 | 79.56 | 74.54 | 79.84 | 77.10 | 77.43 | 84.50 | 80.81 |
| CALF | 55.89 | 91.67 | 69.77 | 59.14 | 78.18 | 67.34 | 52.92 | 92.19 | 67.24 | 69.25 | 77.11 | 72.97 | 62.31 | 83.07 | 71.21 | 54.01 | 68.33 | 60.33 |
| UniTS | 64.75 | 92.48 | 76.17 | 53.58 | 78.53 | 63.70 | 76.77 | 90.47 | 83.06 | 76.77 | 88.68 | 82.24 | 69.18 | 88.38 | 77.61 | 75.82 | 75.08 | 75.45 |
| Timer | 63.70 | 92.57 | 75.46 | 52.94 | 69.75 | 60.20 | 78.78 | 92.41 | 85.05 | 91.52 | 87.67 | 89.55 | 72.85 | 84.08 | 78.06 | 78.02 | 79.00 | 78.51 |
| UTime | 69.40 | 85.12 | 76.46 | 52.76 | 75.12 | 61.98 | 76.30 | 87.94 | 81.71 | 88.61 | 88.54 | 88.58 | 68.09 | 91.89 | 78.22 | 78.90 | 73.84 | 76.28 |
| LMixer | 62.36 | 90.03 | 73.68 | 52.52 | 88.26 | 65.85 | 77.91 | 91.98 | 84.36 | 63.24 | 84.57 | 72.36 | 52.91 | 90.79 | 66.86 | 61.40 | 75.30 | 67.65 |
| DADA | 56.23 | 89.32 | 69.01 | 52.34 | 83.61 | 63.38 | 82.81 | 85.46 | 84.11 | 86.72 | 81.86 | 84.22 | 77.90 | 84.92 | 81.26 | 81.81 | 74.51 | 77.99 |
| MindTS | 76.88 | 89.37 | **82.66** | **71.50** | 77.47 | **74.37** | 77.38 | **95.01** | **85.29** | 85.91 | **95.24** | **90.28** | 77.99 | 90.74 | **83.89** | 90.80 | 87.64 | **89.19** |

Table 14: Average A-R (AUC-ROC), R-A-R (R-AUC-ROC) and V-ROC (VUS-ROC) accuracy for MindTS and baselines within the MM-TSFLib. The best results are highlighted in bold, and the second-best results are underlined.

| Datasets | Weather | | | Energy | | | Environment | | | KR | | | EWJ | | | MDT | | |
|---|---|---|---|---|---|---|---|---|---|---|---|---|---|---|---|---|---|---|
| Metric | A-R | R-A-R | V-ROC | A-R | R-A-R | V-ROC | A-R | R-A-R | V-ROC | A-R | R-A-R | V-ROC | A-R | R-A-R | V-ROC | A-R | R-A-R | V-ROC |
| iTrans* | 73.48 | 71.21 | 70.11 | 67.76 | 67.07 | 65.62 | 81.33 | 70.60 | 73.66 | 84.28 | 75.71 | 77.17 | 78.23 | 70.67 | 74.11 | 78.11 | 66.36 | 69.95 |
| TranAD* | **85.51** | 79.33 | 78.72 | 67.00 | 56.66 | 56.38 | 26.51 | 9.93 | 14.29 | 60.76 | 37.23 | 41.24 | 60.55 | 44.36 | 49.85 | 44.36 | 24.82 | 28.88 |
| Patch* | 82.18 | 80.59 | 80.08 | 66.89 | 61.57 | 58.47 | 94.17 | 91.14 | 90.87 | 82.28 | 72.94 | 74.80 | 78.68 | 69.48 | 72.21 | 84.55 | 75.65 | 77.72 |
| TsNet* | 81.28 | 83.22 | 82.06 | 71.53 | 61.97 | 59.80 | 91.91 | 87.70 | 88.14 | 85.92 | 78.32 | 78.54 | 82.52 | 74.40 | 75.91 | 82.05 | 68.57 | 73.57 |
| Modern* | 53.07 | 81.67 | 81.67 | 71.61 | 66.80 | 66.37 | 92.76 | 89.53 | 89.14 | **93.66** | **90.20** | 89.22 | **87.96** | **83.90** | 83.98 | **88.99** | 81.95 | 82.66 |
| G4TS* | 78.93 | 75.57 | 74.61 | 66.38 | 53.91 | 53.52 | 94.53 | 90.08 | 90.22 | 86.86 | 79.93 | 80.43 | 83.36 | 75.52 | 76.93 | 84.05 | 71.67 | 73.39 |
| CALF | 78.16 | 72.46 | 71.88 | 66.57 | 61.98 | 58.06 | 88.73 | 82.04 | 83.03 | 76.93 | 63.26 | 64.90 | 70.55 | 56.51 | 59.80 | 68.76 | 50.65 | 54.04 |
| UniTS* | 81.38 | 75.67 | 75.21 | 63.89 | 52.76 | 51.89 | 95.16 | 92.50 | 91.98 | 81.29 | 71.69 | 74.25 | 80.34 | 71.91 | 74.39 | 73.35 | 56.87 | 58.82 |
| Timer | 80.96 | 73.74 | 73.26 | 60.65 | 47.01 | 46.39 | 95.31 | 92.02 | 92.02 | 84.08 | 74.61 | 75.92 | 76.59 | 65.39 | 68.19 | 75.54 | 58.66 | 68.19 |
| UTime* | 79.71 | 74.28 | 73.71 | 60.32 | 51.81 | 49.85 | 90.12 | 83.40 | 84.20 | 78.45 | 65.34 | 66.96 | 76.07 | 65.99 | 67.57 | 64.95 | 45.55 | 49.16 |
| LMixer* | 82.77 | 75.73 | 75.29 | 62.94 | 49.93 | 49.06 | 95.25 | 92.04 | 91.74 | 84.02 | 74.58 | 75.21 | 77.63 | 67.12 | 69.37 | 76.83 | 60.21 | 61.72 |
| DADA* | 67.03 | 62.52 | 61.51 | 63.08 | 57.33 | 55.63 | 93.10 | 87.74 | 88.02 | 80.22 | 71.03 | 72.08 | 78.28 | 67.45 | 71.06 | 79.71 | 65.52 | 68.06 |
| MindTS | 84.06 | **83.80** | **82.64** | **81.26** | **75.51** | **74.44** | **96.33** | **94.04** | **93.78** | 93.51 | 89.60 | **89.86** | 87.95 | 83.19 | .84.12 | 90.46 | **83.15** | **83.02** |

Table 15: Average ACC (Accuracy) measures for MindTS and baselines within the MM-TSFLib. The best results are highlighted in bold, and the second-best results are underlined.

| Datasets | Weather | Energy | Environment | KR | EWJ | MDT |
|---|---|---|---|---|---|---|
| Metric | ACC | ACC | ACC | ACC | ACC | ACC |
| iTrans* | 76.09 | 63.69 | 87.31 | 70.62 | 74.44 | 73.44 |
| TranAD* | 64.67 | 70.15 | 44.73 | 87.95 | 64.47 | 65.38 |
| Patch* | 74.59 | 44.92 | 90.22 | 84.37 | 85.53 | 81.32 |
| TsNet* | 82.01 | 66.15 | 93.25 | 94.35 | 84.02 | 76.92 |
| Modern* | 80.23 | 62.46 | 90.50 | 89.45 | 87.22 | 84.62 |
| G4TS* | 69.89 | 41.85 | 93.00 | **95.10** | 85.34 | 91.21 |
| CALF* | 64.75 | 52.62 | 84.90 | 81.36 | 80.45 | 79.12 |
| UniTS* | 70.58 | 47.69 | 90.09 | 89.27 | 84.21 | 86.08 |
| Timer* | 67.50 | 45.54 | 91.43 | 93.60 | 81.02 | 85.16 |
| UTime* | 68.11 | 47.69 | 76.81 | 87.01 | 69.74 | 73.99 |
| LMixer* | 68.69 | 52.00 | 90.56 | 94.92 | 82.14 | 88.46 |
| DADA* | 61.02 | 45.54 | **94.22** | 92.47 | 86.84 | 87.18 |
| MindTS | **84.76** | **80.00** | 90.09 | 91.15 | **88.91** | **93.96** |

Table 16: Average A-P (AUC-PR), R-A-P (R-AUC-PR) and V-PR (VUS-PR) accuracy measures for MindTS and baselines within the MM-TSFLib. The best results are highlighted in bold, and the second-best results are underlined.

| Datasets | Weather | | | Energy | | | Environment | | | KR | | | EWJ | | | MDT | | |
|---|---|---|---|---|---|---|---|---|---|---|---|---|---|---|---|---|---|---|
| Metric | A-P | R-A-P | V-PR | A-P | R-A-P | V-PR | A-P | R-A-P | V-PR | A-P | R-A-P | V-PR | A-P | R-A-P | V-PR | A-P | R-A-P | V-PR |
| iTrans* | 38.56 | 40.43 | 40.30 | 35.68 | 36.17 | 36.21 | 35.35 | 23.91 | 25.85 | 37.89 | 25.61 | 28.12 | 33.63 | 25.74 | 29.79 | 44.10 | 30.76 | 33.88 |
| TranAD* | 60.91 | 52.04 | 52.09 | 36.34 | 33.13 | 33.74 | 7.08 | 4.51 | 4.93 | 53.27 | 28.07 | 28.47 | 27.89 | 15.25 | 17.87 | 26.06 | 13.31 | 14.55 |
| Patch* | 53.69 | 49.95 | 50.17 | 34.66 | 35.51 | 34.66 | 58.65 | 45.40 | 45.52 | 53.60 | 35.57 | 36.37 | 47.92 | 33.45 | 36.22 | 54.52 | 39.92 | 41.85 |
| TsNet* | 48.29 | 51.00 | 50.53 | 42.47 | 38.47 | 38.88 | 63.82 | 50.39 | 50.39 | 67.58 | 53.03 | 51.73 | 55.04 | 41.95 | 43.28 | 60.61 | 41.55 | 52.30 |
| Modern* | 53.07 | 54.17 | 53.42 | 33.98 | 36.64 | 37.44 | 54.31 | 41.53 | 41.36 | 57.71 | 41.72 | 40.86 | 54.48 | 44.64 | 45.41 | 65.84 | 52.31 | 45.88 |
| G4TS* | 49.31 | 45.79 | 45.83 | 33.55 | 31.23 | 31.83 | 69.72 | 56.82 | 56.65 | 72.14 | 58.99 | 57.93 | 57.69 | 42.31 | 43.93 | 68.63 | 51.69 | 52.65 |
| CALF* | 42.86 | 41.39 | 41.43 | 38.48 | 35.10 | 34.11 | 43.76 | 30.42 | 31.72 | 51.19 | 31.22 | 31.52 | 32.08 | 21.30 | 23.62 | 42.75 | 26.64 | 28.70 |
| UniTS* | 49.56 | 44.54 | 44.58 | 27.56 | 31.03 | 31.34 | 64.10 | 50.30 | 50.06 | 56.86 | 41.93 | 44.27 | 50.62 | 37.54 | 39.99 | 53.74 | 36.28 | 37.78 |
| Timer* | 49.24 | 43.32 | 43.36 | 37.69 | 28.91 | 29.57 | 64.31 | 50.97 | 51.20 | 66.57 | 51.81 | 51.56 | 44.23 | 30.88 | 33.36 | 55.56 | 37.36 | 38.23 |
| UTime* | 46.37 | 43.05 | 43.19 | 32.68 | 30.32 | 30.44 | 48.17 | 34.55 | 35.64 | 57.39 | 37.70 | 37.12 | 33.10 | 23.82 | 25.71 | 39.63 | 22.72 | 25.31 |
| LMixer* | 52.74 | 45.90 | 45.94 | 36.74 | 30.38 | 30.91 | 64.84 | 51.14 | 51.22 | 68.67 | 54.04 | 52.98 | 44.75 | 31.84 | 34.06 | 60.18 | 41.95 | 42.61 |
| DADA* | 30.22 | 30.28 | 30.42 | 38.30 | 33.77 | 34.38 | 70.37 | 55.99 | 56.20 | 63.28 | 46.69 | 45.68 | 55.06 | 41.34 | 43.18 | 63.19 | 45.13 | 47.22 |
| MindTS | 58.38 | 57.85 | 57.48 | 50.99 | 50.19 | 50.36 | 69.46 | 56.81 | 56.79 | 67.52 | 52.64 | 53.15 | 61.75 | 49.31 | 50.42 | 76.51 | 66.20 | 65.44 |

Table 17: Average P (Precision), R (Recall) and F1 (F1-score) accuracy measures for MindTS and baselines within the MM-TSFLib. The best results are highlighted in bold, and the second-best results are underlined.

| Datasets | Weather | | | Energy | | | Environment | | | KR | | | EWJ | | | MDT | | |
|---|---|---|---|---|---|---|---|---|---|---|---|---|---|---|---|---|---|---|
| Metric | P | R | F1 | P | R | F1 | P | R | F1 | P | R | F1 | P | R | F1 | P | R | F1 |
| iTrans* | 36.62 | 54.50 | 43.81 | 25.78 | 58.93 | 35.87 | 26.19 | 65.05 | 37.35 | 15.64 | 84.84 | 26.42 | 22.15 | 62.26 | 32.67 | 26.40 | 77.05 | 39.33 |
| TranAD* | 31.65 | 91.94 | 47.08 | 27.47 | 44.64 | 34.01 | 2.52 | 22.58 | 4.54 | 26.87 | 54.55 | 36.00 | 13.44 | 47.17 | 20.92 | 12.35 | 34.43 | 18.18 |
| Patch* | 37.70 | 74.41 | 50.04 | 21.40 | 82.14 | 33.95 | 34.84 | 78.49 | 48.26 | 24.49 | 72.73 | 36.64 | 36.36 | 60.38 | 45.39 | 33.60 | 68.85 | 45.16 |
| TsNet* | 47.77 | 55.92 | 51.53 | 29.55 | 69.64 | 41.49 | 44.79 | 69.35 | 54.43 | 53.19 | 75.76 | 62.50 | 35.45 | 73.58 | 47.85 | 29.56 | 77.05 | 42.73 |
| Modern* | 44.68 | 65.64 | 53.17 | 28.57 | 78.57 | 41.90 | 34.87 | 73.12 | 47.22 | 34.67 | 78.79 | 48.15 | 41.76 | 71.70 | 52.78 | 40.50 | 80.33 | 53.85 |
| G4TS* | 33.67 | 78.44 | 45.12 | 20.44 | 82.14 | 32.74 | 44.60 | 84.41 | 58.36 | 57.78 | 78.79 | 66.67 | 37.62 | 71.70 | 49.35 | 59.15 | 68.85 | 63.64 |
| CALF* | 30.14 | 80.57 | 43.87 | 23.66 | 78.57 | 36.36 | 25.04 | 80.11 | 38.16 | 20.54 | 69.70 | 31.72 | 27.43 | 58.49 | 37.35 | 28.80 | 59.02 | 38.71 |
| UniTS* | 34.95 | 83.65 | 49.30 | 21.50 | 76.79 | 33.59 | 35.41 | 85.48 | 50.08 | 34.21 | 78.79 | 47.71 | 35.24 | 69.81 | 46.84 | 41.94 | 63.93 | 50.65 |
| Timer* | 32.66 | 84.83 | 47.17 | 19.60 | 69.64 | 30.59 | 39.37 | 87.63 | 54.33 | 49.06 | 78.79 | 60.47 | 29.31 | 64.15 | 40.24 | 40.20 | 67.21 | 50.31 |
| UTime* | 32.64 | 81.28 | 46.57 | 20.00 | 67.86 | 30.89 | 18.97 | 91.40 | 31.42 | 27.50 | 66.67 | 38.94 | 21.88 | 79.25 | 34.29 | 23.18 | 57.38 | 33.02 |
| LMixer* | 34.64 | 85.78 | 49.35 | 21.59 | 67.86 | 32.76 | 36.88 | 87.63 | 51.91 | 56.82 | 75.76 | 64.95 | 30.91 | 64.15 | 41.72 | 48.81 | 67.21 | 56.55 |
| DADA* | 25.09 | 64.45 | 36.12 | 19.90 | 71.43 | 31.13 | 50.17 | 77.42 | 60.89 | 43.40 | 69.70 | 53.49 | 40.45 | 67.92 | 50.70 | 44.94 | 65.57 | 53.33 |
| MindTS | 54.77 | 62.56 | 58.41 | 45.16 | 75.00 | 56.38 | 36.21 | 92.47 | 52.04 | 39.40 | 79.09 | 52.53 | 46.34 | 71.70 | 56.30 | 69.44 | 81.97 | 75.19 |

Table 18: Average R-R (Range-Recall), R-P (Range-Precision) and R-F (Range-F1-score) accuracy measures for MindTS and baselines within the MM-TSFLib. The best results are highlighted in bold, and the second-best results are underlined.

| Datasets | Weather | | | Energy | | | Environment | | | KR | | | EWJ | | | MDT | | |
|---|---|---|---|---|---|---|---|---|---|---|---|---|---|---|---|---|---|---|
| Metric | R-R | R-P | R-F | R-R | R-P | R-F | R-R | R-P | R-F | R-R | R-P | R-F | R-R | R-P | R-F | R-R | R-P | R-F |
| iTrans* | 57.06 | 28.34 | 37.87 | 60.48 | 21.28 | 31.48 | 63.78 | 24.31 | 35.20 | 84.20 | 10.59 | 18.81 | 70.00 | 20.27 | 31.43 | 78.46 | 25.60 | 38.60 |
| TranAD* | 87.59 | 22.86 | 36.26 | 39.29 | 11.04 | 17.24 | 21.13 | 3.54 | 6.06 | 50.37 | 28.75 | 36.61 | 51.36 | 15.69 | 24.04 | 36.53 | 19.41 | 25.35 |
| Patch* | 76.28 | 26.40 | 39.23 | 73.10 | 13.12 | 22.25 | 79.48 | 38.28 | 51.67 | 76.30 | 23.26 | 35.65 | 65.00 | 39.94 | 49.48 | 67.50 | 35.81 | 46.80 |
| TsNet* | 68.30 | 44.44 | 53.85 | 57.86 | 28.17 | 37.89 | 71.63 | 44.48 | 54.88 | 81.50 | 53.88 | 67.82 | 79.55 | 38.61 | 51.99 | 75.38 | 26.87 | 39.62 |
| Modern* | 77.18 | 37.38 | 50.37 | 77.57 | 24.80 | 37.58 | 75.37 | 37.65 | 50.22 | 81.23 | 42.62 | 55.91 | 80.00 | 47.45 | 59.57 | 80.96 | 41.71 | 55.05 |
| G4TS* | 78.89 | 24.93 | 37.89 | 72.62 | 13.82 | 23.22 | 83.97 | 46.96 | 60.23 | 83.70 | 64.52 | 72.87 | 78.64 | 44.41 | 56.76 | 68.65 | 61.01 | 64.61 |
| CALF* | 73.30 | 19.52 | 30.83 | 72.14 | 13.79 | 23.16 | 79.95 | 24.60 | 37.02 | 72.59 | 19.70 | 30.35 | 64.09 | 29.55 | 40.45 | 59.81 | 29.60 | 39.60 |
| UniTS* | 80.79 | 27.05 | 40.53 | 70.00 | 18.24 | 28.94 | 85.96 | 36.04 | 50.78 | 85.93 | 34.94 | 49.58 | 78.64 | 37.93 | 51.18 | 66.15 | 41.09 | 50.69 |
| Timer* | 76.94 | 24.87 | 37.59 | 60.71 | 10.29 | 17.60 | 86.19 | 41.23 | 55.78 | 78.10 | 57.29 | 66.07 | 70.91 | 37.62 | 49.16 | 67.50 | 49.55 | 57.15 |
| UTime* | 74.30 | 19.17 | 30.48 | 63.10 | 16.05 | 25.59 | 90.21 | 17.39 | 29.16 | 71.11 | 28.95 | 41.15 | 86.36 | 26.74 | 40.84 | 57.88 | 22.11 | 31.99 |
| LMixer* | 78.67 | 26.09 | 39.18 | 57.14 | 14.18 | 22.73 | 85.77 | 37.99 | 52.66 | 82.22 | 62.37 | 70.93 | 70.91 | 36.33 | 48.05 | 66.54 | 53.16 | 59.10 |
| DADA* | 62.45 | 20.51 | 30.88 | 71.43 | 17.66 | 27.29 | 78.44 | 48.40 | 59.86 | 74.81 | 49.54 | 59.61 | 74.09 | 45.23 | 56.17 | 65.58 | 49.58 | 56.47 |
| MindTS | 66.32 | 45.71 | 54.12 | 71.19 | 34.09 | 46.10 | 92.43 | 35.96 | 51.77 | 83.95 | 42.98 | 56.85 | 80.91 | 49.01 | 61.04 | 82.88 | 70.28 | 76.06 |

Table 19: Average Aff-P (Affiliated-Precision), Aff-R (Affiliated-Recall) and Aff-F (Affiliated-F1score) accuracy measures for MindTS and baselines within the MM-TSFLib. The best results are highlighted in bold, and the second-best results are underlined.

| Datasets | Weather | | | Energy | | | Environment | | | KR | | | EWJ | | | MDT | | |
|---|---|---|---|---|---|---|---|---|---|---|---|---|---|---|---|---|---|---|
| Metric | Aff-P | Aff-R | Aff-F | Aff-P | Aff-R | Aff-F | Aff-P | Aff-R | Aff-F | Aff-P | Aff-R | Aff-F | Aff-P | Aff-R | Aff-F | Aff-P | Aff-R | Aff-F |
| iTrans* | 65.34 | 89.01 | 75.36 | 61.39 | 88.49 | 72.49 | 68.74 | 85.01 | 76.02 | 66.33 | 95.80 | 78.39 | 67.35 | 93.65 | 78.23 | 65.66 | 94.48 | 77.47 |
| TranAD* | 65.06 | 96.53 | 77.73 | 55.68 | 46.25 | 50.53 | 46.37 | 85.21 | 61.38 | 72.94 | 72.07 | 72.50 | 56.74 | 88.12 | 69.03 | 58.33 | 67.33 | 63.60 |
| Patch* | 65.00 | 95.59 | 77.05 | 52.79 | 89.04 | 66.28 | 76.82 | 87.27 | 81.71 | 74.63 | 85.09 | 79.52 | 74.07 | 77.53 | 76.49 | 73.28 | 85.38 | 78.87 |
| TsNet* | 72.96 | 88.77 | 80.09 | 59.98 | 70.49 | 66.71 | 78.96 | 81.50 | 80.21 | 86.98 | 84.73 | 85.84 | 75.23 | 89.92 | 81.92 | 70.26 | 94.56 | 80.62 |
| Modern* | 71.93 | 94.02 | 81.50 | 61.88 | 87.67 | 72.13 | 77.34 | 85.83 | 81.36 | 80.39 | 89.87 | 84.87 | 77.77 | 86.27 | 81.88 | 75.68 | 88.72 | 81.68 |
| G4TS* | 65.88 | 92.12 | 76.82 | 52.88 | 88.23 | 67.38 | 80.78 | 88.46 | 84.44 | 80.10 | 88.68 | 84.17 | 76.80 | 86.52 | 81.37 | 86.29 | 77.87 | 81.86 |
| CALF* | 60.87 | 94.23 | 73.96 | 56.97 | 85.13 | 68.26 | 70.34 | 92.85 | 80.04 | 67.34 | 83.77 | 75.02 | 67.43 | 83.71 | 74.69 | 69.62 | 70.81 | 70.80 |
| UniTS* | 64.53 | 92.92 | 76.38 | 55.64 | 79.36 | 65.42 | 76.52 | 91.71 | 83.43 | 78.10 | 88.68 | 83.06 | 73.47 | 83.22 | 78.04 | 78.40 | 75.08 | 76.70 |
| Timer* | 64.15 | 91.36 | 75.37 | 53.24 | 69.68 | 60.36 | 79.02 | 90.84 | 84.52 | 91.52 | 87.67 | 89.61 | 73.11 | 86.05 | 79.05 | 79.84 | 75.68 | 77.93 |
| UTime* | 61.34 | 93.03 | 73.93 | 54.42 | 81.87 | 65.38 | 63.38 | 97.22 | 76.73 | 71.38 | 84.49 | 77.38 | 60.36 | 92.98 | 73.20 | 68.40 | 76.74 | 72.33 |
| LMixer* | 65.68 | 82.77 | 76.30 | 54.97 | 69.68 | 61.46 | 78.10 | 90.01 | 83.76 | 92.60 | 87.59 | 90.03 | 71.76 | 85.98 | 78.31 | 84.87 | 72.70 | 78.31 |
| DADA* | 57.53 | 88.49 | 69.73 | 52.95 | 83.46 | 64.80 | 82.94 | 84.75 | 83.84 | 86.72 | 81.86 | 84.22 | 80.16 | 82.70 | 81.41 | 81.44 | 74.62 | 77.89 |
| MindTS | 76.88 | 89.37 | 82.66 | 71.50 | 77.47 | 74.37 | 77.38 | 95.01 | 85.29 | 85.91 | 95.24 | 90.28 | 77.99 | 90.74 | 83.89 | 90.80 | 87.64 | 89.19 |

## D    VISUALIZATION CASE STUDIES

To enable intuitive performance comparison, we conduct a comparative visualization of anomaly scores between MindTS and GPT4TS (the original model and within the MM-TSFLib), as shown in Figure 6. MindTS exhibits the most distinguishable anomaly scores compared to existing methods.

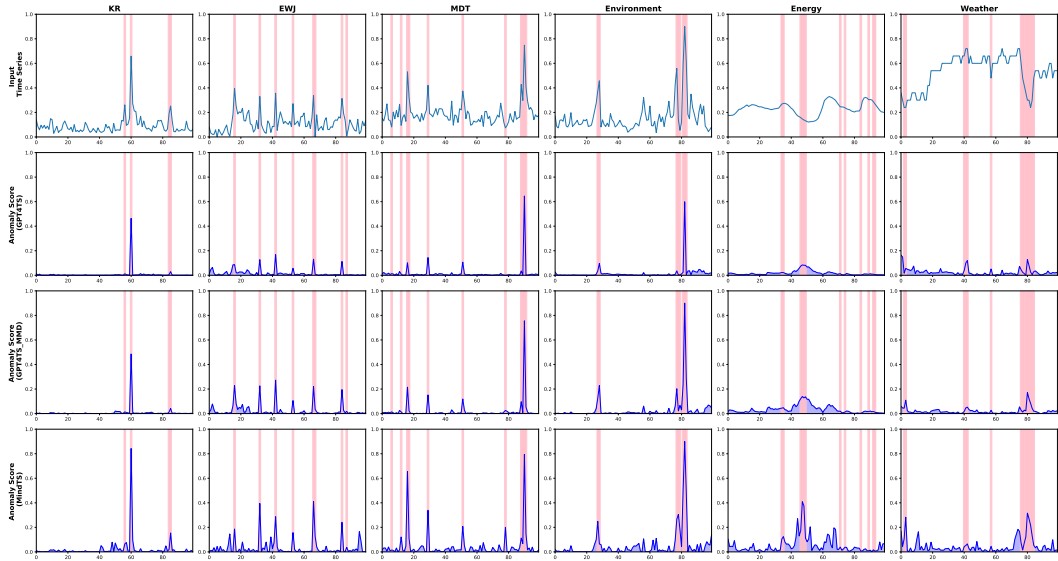

Figure 6: Visualization comparisons of anomaly scores between MindTS and GPT4TS for all datasets.

## E    ADDITIONAL MODEL ANALYSIS

### E.1    CROSS-VIEW TEXT FUSION ANALYSES

In this paper, we use the endogenous text $\mathbf{H}_{\text{text}}^{O}$ as the query and the exogenous text $\mathbf{H}_{\text{text}}^{C}$ as the key and value to obtain the fused text representation $\mathbf{Z}_{\text{text}}$ to enhance semantic consistency with the time series and extract the most relevant background information. We further conduct additional experiments comparing different attention strategies to demonstrate the effectiveness of this design: (a) MindTS (q/kv reverse), setting $\mathbf{H}_{\text{text}}^{C}$ as query and $\mathbf{H}_{\text{text}}^{O}$ as key/value; (b) MindTS (self-attention), using self-attention only; (c) MindTS (two-way), replace the one-way cross-attention with a two-way block where exogenous text also queries endogenous features; and (d) MindTS, setting $\mathbf{H}_{\text{text}}^{O}$ as query and $\mathbf{H}_{\text{text}}^{C}$ as key/value. As shown in Table 20, the configuration where $\mathbf{H}_{\text{text}}^{O}$ is used as the query and $\mathbf{H}_{\text{text}}^{C}$ as the key/value yields the best performance. In cross-view attention, using $\mathbf{H}_{\text{text}}^{O}$ as query and $\mathbf{H}_{\text{text}}^{C}$ as key/value allows the model to better extract supplementary information from exogenous text that is most relevant to each time patch. By contrast, relying solely on self-attention limits the model's ability to directly learn interactions between different text modalities. Although two-way cross-attention design possesses a certain level of representational capacity, it does not bring significant performance improvements on most datasets.

Table 20: Evaluation of cross-view text fusion variants.

| Datasets | Weather | | | Energy | | | Environment | | | KR | | | EWJ | | | MDT | | |
|---|---|---|---|---|---|---|---|---|---|---|---|---|---|---|---|---|---|---|
| Metric | Aff-F | V-PR | V-ROC | Aff-F | V-PR | V-ROC | Aff-F | V-PR | V-ROC | Aff-F | V-PR | V-ROC | Aff-F | V-PR | V-ROC | Aff-F | V-PR | V-ROC |
| MindTS (q/kv reverse) | 82.29 | 57.33 | 82.27 | 73.73 | 48.22 | 73.06 | 83.94 | 55.49 | 92.64 | 87.14 | 50.72 | 88.72 | 82.71 | 49.99 | **84.34** | 87.95 | 65.07 | 82.30 |
| MindTS (self-attention) | 81.69 | 56.69 | 82.46 | 72.47 | 48.17 | 71.18 | **85.79** | 56.54 | 93.41 | 87.27 | 52.48 | 90.31 | 82.96 | 46.28 | 83.59 | 83.82 | 57.67 | 77.80 |
| MindTS (two-way) | 79.43 | 57.17 | **82.70** | 72.57 | 44.64 | 70.42 | 84.47 | 53.17 | 93.17 | 87.39 | **54.54** | **91.77** | 77.76 | 46.92 | 78.28 | 81.12 | 57.62 | 75.23 |
| MindTS | **82.66** | **57.48** | 82.64 | **74.37** | **50.36** | **74.44** | 85.29 | **56.79** | **93.78** | **90.28** | 53.15 | 89.86 | **83.89** | **50.42** | 84.12 | **89.19** | **65.44** | **83.02** |

**Comparison with LLM condenser.** To further illustrate the effectiveness of the content condenser, We add comparative experiments among the following four model variants: (a) w/o text, input includes only time series (no text); (b) w/o filtering, input includes time series and text, where

the text is used without any redundancy filtering; (c) LLM-based compression, input includes time series and text, where the text is processed by LLM-based compression for redundancy filtering (the content condenser is removed); (d)content condenser, input includes time series and text, where the text is processed by our proposed content condenser for redundancy filtering.

As shown in Table 21, variant (d) achieves the best performance across all evaluation metrics. Variant (b) performs the worst, even lower than (a), indicating that unfiltered text introduces redundancy that degrades performance. This confirms the existence of text redundancy. Variant (c) with LLM-based compression to filter the text achieves better results than (a) and (b), demonstrating that compression helps alleviate redundancy to some extent. Most importantly, variant (d) significantly outperforms the LLM-based compression approach, highlighting the effectiveness of our proposed content condenser. Unlike LLMs, our module is explicitly optimized under the multimodal objective to preserve time-aligned semantics and suppress irrelevant textual content, thereby enhancing outlier detection performance. In contrast, LLM-based compression considers text-only semantics, which fails to capture time-aligned semantics.

Table 21: Ablation on text redundancy filtering strategies. The best results are highlighted in bold.

| Datasets | (a) w/o text | | | (b) w/o filtering | | | (c) LLM-based compression | | | (d) content condenser | | |
|---|---|---|---|---|---|---|---|---|---|---|---|---|
| Metric | Aff-F | V-PR | V-ROC | Aff-F | V-PR | V-ROC | Aff-F | V-PR | V-ROC | Aff-F | V-PR | V-ROC |
| KR | 84.11 | 45.11 | 79.43 | 80.52 | 37.82 | 74.38 | 87.32 | 47.95 | 88.52 | 90.28 | 53.15 | 89.86 |
| EWJ | 81.87 | 45.22 | 79.32 | 78.79 | 38.47 | 80.03 | 80.26 | 42.90 | 78.84 | 83.89 | 50.42 | 84.12 |
| MDT | 84.00 | 58.32 | 81.68 | 81.79 | 51.40 | 75.43 | 84.31 | 58.74 | 81.13 | 89.19 | 65.44 | 83.02 |

## E.2 MULTIMODAL ANALYSIS

In Table 22, we observe that compared to time series unimodal settings, incorporating time-text multimodal settings consistently yields better results. Notably, on some datasets (e.g., Energy and MDT), MindTS outperforms the baselines. As reported in Table 3 of the paper, these datasets are relatively small in size but have high anomaly ratios, making anomalies more densely distributed and easier to detect. For models with strong reconstruction capacity, this setting increases the risk of overfitting, as anomalies may also be reconstructed too well, thereby degrading detection performance. In contrast, simpler methods are less prone to reconstructing anomalies, which sometimes results in competitive outcomes. Nevertheless, across all datasets, MindTS consistently demonstrates superior performance over unimodal settings, effectively integrating multimodal information to enhance anomaly detection.

Table 22: The results of MindTS and MindTS(unimodel) across all datasets (all results in %, best results are highlighted in bold).

| Method | Weather | | | Energy | | | Environment | | | KR | | | EWJ | | | MDT | | |
|---|---|---|---|---|---|---|---|---|---|---|---|---|---|---|---|---|---|---|
| Metric | Aff-F | V-PR | V-ROC | Aff-F | V-PR | V-ROC | Aff-F | V-PR | V-ROC | Aff-F | V-PR | V-ROC | Aff-F | V-PR | V-ROC | Aff-F | V-PR | V-ROC |
| MindTS(unimodel) | 75.96 | 45.69 | 74.19 | 73.14 | 46.66 | 71.36 | 80.16 | 44.28 | 86.43 | 84.11 | 45.11 | 79.43 | 81.87 | 45.22 | 79.32 | 84.00 | 58.32 | 81.68 |
| DADA | 69.01 | 30.00 | 61.03 | 64.38 | 34.18 | 54.37 | 84.11 | 54.20 | 87.69 | 84.22 | 45.90 | 70.82 | 81.26 | 43.36 | 71.79 | 77.99 | 46.81 | 66.76 |
| ModernTCN | 81.06 | 52.13 | 81.14 | 70.76 | 36.60 | 65.05 | 81.07 | 42.26 | 89.78 | 84.42 | 39.95 | 88.87 | 81.57 | 44.75 | 83.88 | 80.81 | 52.18 | 82.30 |
| Timer | 75.46 | 43.21 | 73.22 | 60.20 | 29.46 | 46.03 | 84.19 | 51.42 | 92.10 | 89.55 | 51.41 | 75.99 | 78.06 | 33.17 | 67.72 | 78.51 | 38.38 | 60.28 |
| MindTS | 82.66 | 57.48 | 82.64 | 74.37 | 50.36 | 74.44 | 85.29 | 56.79 | 93.78 | 90.28 | 53.15 | 89.86 | 83.89 | 50.42 | 84.12 | 89.19 | 65.44 | 83.02 |

## E.3 INFERENCE TIME

In Table 23, we compare MindTS with other models across different datasets in terms of inference time and memory cost. Overall, MindTS achieves competitive inference time while maintaining superior detection performance. Regarding memory usage, the additional cost is moderate and remains well within the capacity of modern hardware, making MindTS practical for real-world deployment.

## E.4 EXOGENOUS TEXT QUALITY ANALYSIS

From Table 24 to Table 26, we compare different types of exogenous text quality variations to evaluate the robustness of our model: (a) noisy, by introducing random spelling errors within sentences; (b) irrelevant, by replacing the original sentences with unrelated text from different domains; and (c) incomplete, by removing portions of the text descriptions.

Table 23: Run times and memory costs on different datasets. Lower values represent better performance. The notation with $*$ denotes results obtained by extending the baselines using the recent time-series multimodal framework MM-TSFLib.

| Method | Inference Time (s) | | | Memory Cost (GB) | | |
|---|---|---|---|---|---|---|
| | MDT | KR | EWJ | MDT | KR | EWJ |
| MindTS | 0.2302 | 0.1977 | 0.4130 | 14.69 | 14.62 | 14.41 |
| ModernTCN$*$ | 0.1582 | 0.1383 | 0.3965 | 13.61 | 13.66 | 13.57 |
| GPT4TS$*$ | 0.2676 | 0.2425 | 0.4716 | 13.85 | 13.89 | 13.80 |
| LLMMixer$*$ | 0.2585 | 0.2104 | 0.4619 | 14.13 | 14.08 | 13.97 |
| UniTime$*$ | 0.2537 | 0.2462 | 0.4760 | 14.20 | 14.15 | 14.06 |

As shown in Table 24, under single-type settings, the performance of MindTS only slightly decreases compared to the clean setting. As shown in Table 25, when the noise strength is within a reasonable range (e.g., 0.2 and 0.4), the content condenser effectively filters out redundant text information, thereby mitigating its impact. As a result, MindTS still maintains robust performance. Under more challenging conditions, such as multiple text types combinations (Table 26) or high noise intensity (e.g., 0.8), the performance degradation becomes more noticeable. Nevertheless, the overall results remain within acceptable bounds. Additionally, we clarify that the exogenous texts in our datasets are collected from real-world sources (e.g., news, public reports), which inevitably contain redundant or partially irrelevant content. Nevertheless, MindTS consistently achieves strong results across multiple realistic datasets, demonstrating its robustness and adaptability to real-world text quality variation.

Table 24: Results under different types of exogenous text quality variations (all results in %, best results are highlighted in bold).

| Method | MindTS | | | MindTS (noisy) | | | MindTS (irrelevant) | | | MindTS (incomplete) | | |
|---|---|---|---|---|---|---|---|---|---|---|---|---|
| Metric | Aff-F | V-PR | V-ROC | Aff-F | V-PR | V-ROC | Aff-F | V-PR | V-ROC | Aff-F | V-PR | V-ROC |
| MDT | **89.19** | **65.44** | **83.02** | 87.45 | 62.12 | 81.02 | 86.54 | 60.10 | 80.17 | 87.85 | 61.95 | 80.59 |
| Energy | **74.37** | **50.36** | **74.44** | 72.53 | 47.42 | 72.16 | 72.21 | 46.52 | 71.29 | 73.13 | 47.62 | 73.50 |

Table 25: Results of the noisy method under different noise strengths (s).

| Dataset | MindTS | | | s = 0.2 | | | s = 0.4 | | | s = 0.6 | | | s = 0.8 | | |
|---|---|---|---|---|---|---|---|---|---|---|---|---|---|---|---|
| Metric | Aff-F | V-PR | V-ROC | Aff-F | V-PR | V-ROC | Aff-F | V-PR | V-ROC | Aff-F | V-PR | V-ROC | Aff-F | V-PR | V-ROC |
| MDT | **89.19** | **65.44** | **83.02** | 87.45 | 62.12 | 81.02 | 86.38 | 60.03 | 79.46 | 83.71 | 58.71 | 78.69 | 79.84 | 55.21 | 76.46 |
| Energy | **74.37** | **50.36** | **74.44** | 72.53 | 49.42 | 73.16 | 71.17 | 48.26 | 70.03 | 69.35 | 47.22 | 69.97 | 66.69 | 45.02 | 66.13 |

Table 26: Results under combined types of exogenous text quality variations.

| Method | MindTS | | | MindTS (noisy + irrelevant) | | | MindTS (noisy + incomplete) | | | MindTS (irrelevant + incomplete) | | | MindTS (three types) | | |
|---|---|---|---|---|---|---|---|---|---|---|---|---|---|---|---|
| Metric | Aff-F | V-PR | V-ROC | Aff-F | V-PR | V-ROC | Aff-F | V-PR | V-ROC | Aff-F | V-PR | V-ROC | Aff-F | V-PR | V-ROC |
| MDT | **89.19** | **65.44** | **83.02** | 84.91 | 57.29 | 78.23 | 85.28 | 55.60 | 78.34 | 84.81 | 58.34 | 77.68 | 81.64 | 52.17 | 75.28 |
| Energy | **74.37** | **50.36** | **74.44** | 70.33 | 44.71 | 70.18 | 71.89 | 45.86 | 69.96 | 69.74 | 45.26 | 70.51 | 66.05 | 42.08 | 66.88 |

## F ALIGNMENT VISUALIZATION ANALYSIS

In this section, we visualize the learned similarity matrix between time series and text representations before and after alignment (see Figure 7). Before alignment, the similarity distribution appears scattered, indicating weak semantic correspondence between modalities. After alignment, the similarity becomes more concentrated along the diagonal, revealing clear associations between relevant time series and text representations. This demonstrates that the proposed alignment module successfully establishes cross-modal consistency, thereby enhancing the model's ability to utilize textual information.

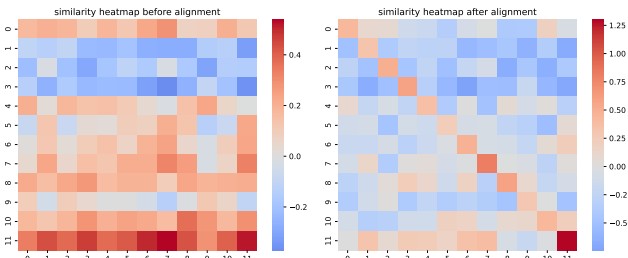

Figure 7: Visualization comparisons between before and after alignment. The x-axis denotes time representations of each patch, while the y-axis represents text representations of each patch.

In addition, we present qualitative visualizations that illustrate how alignment improves anomaly detection decisions. Specifically, we compare the anomaly score with and without alignment, as shown in Figure 8. The first row shows the original data with ground-truth anomaly positions, the second row displays the anomaly scores of the model with multimodal alignment, and the third row presents the anomaly scores of the model without multimodal alignment. When alignment is applied, the model exhibits more distinguishable anomaly scores around true abnormal regions. In contrast, the model without alignment fails to effectively increase the gap between the anomaly scores of normal points and anomalies, leading to many false positives.

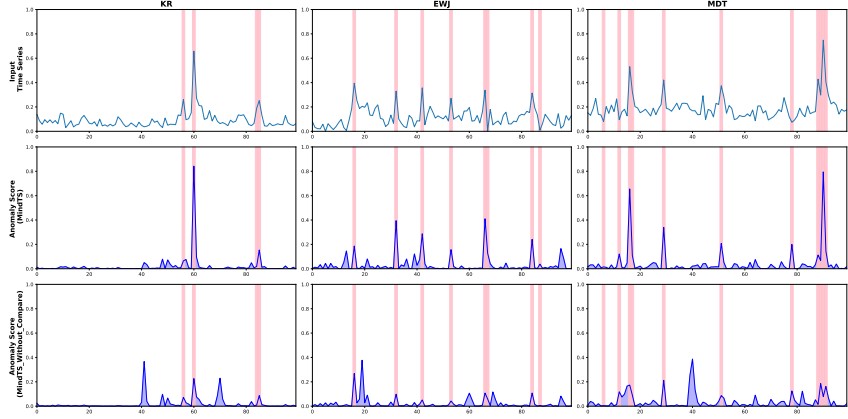

Figure 8: Visualization comparisons of the anomaly score between with and without alignment.

## G PROMPT DESIGN ANALYSIS

The endogenous prompt in MindTS is constructed using the template shown in Figure 9. Inspired by the analyses in TimeLLM Jin et al. (2023) and HiTime Tao et al. (2024), we explore how different prompt designs affect model performance as follows:

(1) Template variant (MindTS-T): prompts generated using a different template formulation, as shown in Figure 10;

(2) Statistical variant (MindTS-S): removing parts of the statistical descriptors: (a) keeping only dataset description, (b) removing min, max, median values, (c) removing trend information, (d) removing lag information;

(3) Temporal granularity variant (MindTS-TG): changing endogenous text generation from the per-patch level to the per-sample level.

As shown in Table 27, altering only the template formulation while keeping the content unchanged has almost no impact on performance. When a few statistical descriptors are removed, the performance decline is minor. However, when most of the statistics

Figure 9: Prompt example.

are omitted (MindTS-S(a)), performance drops more noticeably, as the generated endogenous text becomes too sparse to convey meaningful information. Changing the temporal granularity of endogenous prompt generation also leads to noticeable performance differences, primarily because coarse-grained endogenous text weakens MindTS's ability to achieve fine-grained alignment. These results together confirm MindTS's robustness to reasonable variations in prompt design. We clarify that the selected statistical descriptors represent fundamental characteristics of time series. Therefore, the performance improvement does not depend on carefully tuning the prompts but rather arises from the intrinsic capabilities of the proposed model.

Table 27: Ablation on endogenous prompt design across different variants.

| Dataset | Metric | MindTS | MindTS-S(a) | MindTS-S(b) | MindTS-S(c) | MindTS-S(d) | MindTS-T | MindTS-TG |
|---------|--------|--------|-------------|-------------|-------------|-------------|----------|-----------|
| **MDT** | Aff-F | 89.19 | 83.22 | 87.59 | 87.62 | 87.14 | 88.38 | 84.78 |
| | V-PR | 65.44 | 57.27 | 64.05 | 64.49 | 64.33 | 64.65 | 58.49 |
| | V-ROC | 83.02 | 78.72 | 82.75 | 82.13 | 82.90 | 83.19 | 80.75 |
| **Energy** | Aff-F | 74.37 | 66.65 | 73.74 | 74.03 | 73.85 | 74.21 | 69.31 |
| | V-PR | 50.36 | 44.29 | 48.77 | 49.25 | 49.81 | 49.34 | 45.32 |
| | V-ROC | 74.44 | 68.59 | 72.33 | 71.78 | 72.66 | 72.14 | 67.97 |

## H  Time Series Forecasting Experimental Results

Although MindTS is primarily designed for anomaly detection, its architectural components are inherently extensible. MindTS proposes a fine-grained time-text semantic alignment mechanism consisting of endogenous text generation, cross-view text fusion, and a multimodal alignment strategy to ensure that time series and text semantics are consistently matched. Since accurate alignment is crucial for multimodal tasks involving heterogeneous semantic spaces, the alignment mechanism in MindTS possesses extensibility. Moreover, MindTS incorporates

Figure 10: Different template formulation.

a content condenser to filter redundant textual information before cross-modal interaction. Redundant text is a common challenge in many multimodal applications. Together, these components make MindTS applicable to other multimodal time series applications.

To further evaluate extensibility, MindTS is adapted to time series forecasting and compared with forecasting-oriented baselines (Li et al., 2025b; Liu et al., 2024b). Beyond anomaly detection, time series forecasting (Tian et al., 2025; 2026; Fang et al., 2026; Mei et al., 2025; Pan et al., 2023; Feng et al.; Liu et al., 2025a; Li et al., 2025a; Cheng et al., 2023; Wu et al., 2024) is another critical task in temporal data analysis. As shown in Table 28, MindTS achieves competitive performance across multiple forecasting datasets, demonstrating that the core components generalize effectively beyond anomaly detection. These results confirm the extensibility of MindTS to other multimodal time series applications.

Table 28: Multimodel time series forecasting results with forecasting horizons $F \in \{6, 8, 10, 12\}$.

| Models | | MindTS | | Time-MMD | | TaTS | |
|---|---|---|---|---|---|---|---|
| Metrics | | mse | mae | mse | mae | mse | mae |
| Agriculture | 6 | 0.167 | 0.269 | 0.146 | 0.263 | 0.140 | 0.251 |
| | 8 | 0.195 | 0.283 | 0.189 | 0.310 | 0.187 | 0.282 |
| | 10 | 0.228 | 0.316 | 0.254 | 0.320 | 0.244 | 0.320 |
| | 12 | 0.258 | 0.343 | 0.338 | 0.369 | 0.290 | 0.350 |
| | AVG | **0.212** | 0.303 | 0.232 | 0.316 | 0.215 | **0.301** |
| Traffic | 6 | 0.157 | 0.225 | 0.162 | 0.242 | 0.174 | 0.239 |
| | 8 | 0.176 | 0.251 | 0.168 | 0.228 | 0.178 | 0.242 |
| | 10 | 0.167 | 0.213 | 0.178 | 0.237 | 0.185 | 0.243 |
| | 12 | 0.181 | 0.237 | 0.188 | 0.246 | 0.189 | 0.242 |
| | AVG | **0.170** | **0.232** | 0.174 | 0.239 | 0.179 | 0.238 |
| Economy | 6 | 0.172 | 0.331 | 0.199 | 0.350 | 0.196 | 0.350 |
| | 8 | 0.215 | 0.370 | 0.216 | 0.367 | 0.214 | 0.376 |
| | 10 | 0.215 | 0.363 | 0.224 | 0.373 | 0.223 | 0.367 |
| | 12 | 0.242 | 0.379 | 0.239 | 0.388 | 0.239 | 0.388 |
| | AVG | **0.211** | **0.361** | 0.219 | 0.370 | 0.215 | 0.368 |

# I    ANALYSIS OF ENDOGENOUS TEXT GENERATION

To assess the actual benefit of the endogenous text generation step, we conducted an ablation study in which the model directly uses $\mathbf{H}_{\text{time}}$ as the query to fuse exogenous text, without endogenous text generation. As shown in Table 29, the model incorporating endogenous text achieves clearly better performance. The endogenous text is derived directly from the time series, capturing temporal characteristics that align closely with local patterns. Consequently, fusing endogenous and exogenous texts enables the model to extract supplementary information from exogenous sources that is most relevant to each time patch.

In contrast, directly using $\mathbf{H}_{\text{time}}$ as the query to fuse exogenous text does not yield performance gains. This is likely because the time-series and text modalities inherently reside in different semantic spaces; interacting them directly without prior alignment results in insufficient information extraction due to modality discrepancies.

Table 29: Ablation study on $\mathbf{H}_{\text{time}}$ as the query to fuse exogenous text, without endogenous text generation (all results in %, best results are highlighted in bold).

| Method | Weather | | | Energy | | | Environment | | | KR | | | EWJ | | | MDT | | |
|---|---|---|---|---|---|---|---|---|---|---|---|---|---|---|---|---|---|---|
| Metric | Aff-F | V-PR | V-ROC | Aff-F | V-PR | V-ROC | Aff-F | V-PR | V-ROC | Aff-F | V-PR | V-ROC | Aff-F | V-PR | V-ROC | Aff-F | V-PR | V-ROC |
| MindTS ($\mathbf{H}_{\text{time}}$ as query) | 79.35 | 56.21 | 80.13 | 72.69 | 47.52 | 71.47 | 82.67 | 52.45 | 92.87 | 87.93 | 48.27 | 89.02 | 82.33 | 48.98 | 81.48 | 87.85 | 64.33 | 81.26 |
| MindTS | **82.66** | **57.48** | **82.64** | **74.37** | **50.36** | **74.44** | **85.29** | **56.79** | **93.78** | **90.28** | **53.15** | **89.86** | **83.89** | **50.42** | **84.12** | **89.19** | **65.44** | **83.02** |

# J    ANALYSIS OF MULTIMODAL ALIGNMENT AS A STANDALONE OBJECTIVE

In this section, we compare multimodal alignment trained as a standalone objective with the auxiliary alignment setting used in MindTS. As shown in Table 30, training alignment as a standalone objective does not lead to performance improvement. We clarify that although optimizing multimodal alignment alone may strengthen the alignment depth, it neglects the specific role of the learned representations in anomaly detection. In contrast, jointly optimizing alignment with other objectives allows multiple losses to guide and regularize each other, enabling the model to emphasize features that are both semantically aligned and task-relevant.

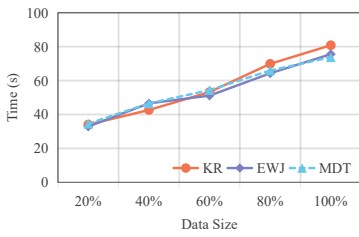

Figure 11: Scalability comparison under different data sizes.

Table 30: Multimodal alignment as a standalone objective (all results in %, best in Bold).

| Datasets | MindTS (auxiliary) | | | MindTS (standalone) | | |
|---|---|---|---|---|---|---|
| Metric | Aff-F | V-PR | V-ROC | Aff-F | V-PR | V-ROC |
| MDT | 89.19 | 65.44 | 83.02 | 80.33 | 57.18 | 76.35 |
| Energy | 74.37 | 50.36 | 74.44 | 70.28 | 44.89 | 69.61 |

## K  SCALABILITY STUDIES

We would like to clarify that due to the current difficulty in obtaining larger-scale multimodal datasets, we evaluate the scalability of MindTS by varying the proportion of training data (20%, 40%, 60%, 80%, 100%) on the available datasets. As shown in Figure 11, the running time increases approximately sub-linearly with the data size. As the data volume grows, total training time increases because more iterations are required to process larger datasets. Compared with baselines, MindTS has a certain advantage in training time (see Table 31). These observations collectively demonstrate that MindTS maintains reasonable computational scalability and remains practical for real-world multimodal time series applications.

Table 31: Training time (s) comparison with baselines.

| Datasets | MindTS | GPT4TS* | LLMMixer* | ModernTCN* | UniTime* |
|---|---|---|---|---|---|
| KR | 80.83 | 81.17 | 84.35 | 19.72 | 77.98 |
| EWJ | 75.56 | 73.66 | 70.09 | 18.65 | 70.23 |
| MDT | 73.76 | 74.49 | 70.01 | 18.61 | 68.53 |

