# OpenReview forum: "Towards Multimodal Time Series Anomaly Detection with Semantic Alignment and Condensed Interaction"
_ICLR.cc/2026/Conference — ICLR 2026 Poster_

### Official Review · Reviewer_2Z5d · 2025-10-20

**Soundness:** 3
**Presentation:** 3
**Contribution:** 3
**Rating:** 6
**Confidence:** 4

**Summary:**

This paper presents MindTS, a novel framework for multimodal time series anomaly detection that effectively integrates numerical and textual data. Traditional methods rely solely on unimodal numerical signals, neglecting the complementary semantic information in text. MindTS addresses two main challenges: achieving semantic alignment between heterogeneous modalities and filtering redundant textual content to enhance cross-modal interaction. To this end, the model introduces a Fine-grained Time–Text Semantic Alignment module, which fuses endogenous and exogenous text through multi-view integration to ensure consistent semantic representation, and a Content Condenser Reconstruction mechanism that removes redundant information and enables cross-modal reconstruction for improved interaction. Extensive experiments on six real-world multimodal datasets demonstrate that MindTS consistently achieves competitive or superior performance over existing methods, validating its capability to accurately capture semantic dependencies between modalities and detect anomalies more effectively than prior unimodal or naïve multimodal approaches.

**Strengths:**

1. The paper introduces a well-designed multimodal framework that effectively integrates both time-series and textual information for anomaly detection. Its dual-perspective modeling—using endogenous and exogenous texts—captures complementary semantics that unimodal models overlook. This fine-grained alignment significantly enhances contextual understanding, allowing the model to detect anomalies grounded in both temporal dynamics and semantic context.
2. The proposed Content Condenser Reconstruction module is an innovative mechanism for filtering redundant textual information while preserving meaningful semantics. By minimizing mutual information and enabling cross-modal reconstruction, it not only improves model interpretability but also strengthens the interaction between modalities. This leads to more compact, semantically relevant representations and improved robustness against noisy or irrelevant text inputs.
3. Extensive experiments on six real-world multimodal datasets validate the generality and effectiveness of MindTS. The model consistently achieves state-of-the-art or highly competitive results across multiple evaluation metrics, demonstrating its robustness, scalability, and strong cross-domain applicability. The paper’s comprehensive comparisons and ablation studies provide clear empirical evidence supporting the model’s advantages and technical soundness.

**Weaknesses:**

1. The paper lacks a detailed exploration of computational efficiency and scalability. While MindTS integrates multiple components, such as fine-grained semantic alignment and content condensation, the training cost and inference latency on large-scale or streaming time series remain unclear. Without such analysis, its practicality in real-time or resource-constrained applications may be limited.

2. Although MindTS achieves impressive performance, its interpretability could be improved. The model’s multiple fusion and reconstruction layers make it challenging to understand how specific textual cues influence anomaly decisions. More qualitative case studies or explainable AI techniques would help clarify how the semantic alignment contributes to detection outcomes.

3. The experiments, while comprehensive, are primarily limited to datasets with well-structured external text. The model’s robustness against noisy, incomplete, or domain-shifted text sources remains underexplored. This limits confidence in its generalization to real-world multimodal environments where textual information is often inconsistent or less reliable.

**Questions:**

See weakness

---

> ### Author Response · Authors · 2025-11-21
> **Response to W1: Computational efficiency and scalability.**
>
> We sincerely thank Reviewer 2Z5d for recognizing our work and for the constructive feedback. We address the concerns as follows.
>
> ### W1: Computational efficiency and scalability.
>
> Thank you for the insightful comment. We would like to clarify that due to the current difficulty in obtaining larger-scale multimodal datasets, we evaluate the scalability of MindTS by varying the proportion of training data (20%, 40%, 60%, 80%, 100%) on the available datasets. In the rebuttal, we summarize the results in **Rebuttal Tables K, L, and M**. In the revised paper ($\underline{\text{see Appendix K}}$), we further present the results using more intuitive visualizations.
>
> **Rebuttal Table K.** Scalability comparison under different data sizes (training time (s)). The running time increases approximately sub-linearly with the data size. As the data volume grows, total training time increases because more iterations are required to process larger datasets.
> | **Datasets** | **MindTS (20%)** | **MindTS (40%)** | **MindTS (60%)** | **MindTS (80%)** | **MindTS (100%)** |
> | :----------: | :------------: | :------------: | :------------: | :------------: | :------------: |
> |    **KR**    |     34.05     |     42.65     |     53.23     |     69.87    |     80.83    |
> |    **EWJ**   |     32.88     |     46.55     |     51.12     |     64.29   |     75.56   |
> |    **MDT**   |     34.59   |     46.54    |     54.38   |     65.95     |     73.76     |
>
>
> **Rebuttal Table L.** Training time (s) comparison with baselines. Compared with baselines, MindTS has a certain advantage in training time.
> | **Datasets** | **MindTS** | **GPT4TS*** | **LLMMixer*** | **ModernTCN*** | **UniTime*** |
> | :----------: | :--------: | :--------: | :----------: | :-----------: | :---------: |
> |    **KR**    |   80.83  |   81.17   |    84.35   |     19.72   |    77.98   |
> |    **EWJ**   |   75.56  |   73.66   |    70 .09   |     18.65    |    70.23   |
> |    **MDT**   |   73.76  |   74.49  |    70.01    |     18.61   |    68.53  |
>
>
> **Rebuttal Table M.** Memory costs (GB) on different datasets comparison with baselines. The additional memory usage introduced by MindTS is moderate and remains well within the capacity of modern hardware, making the method practical for real-world deployment.
> | **Datasets** | **MindTS** | **GPT4TS*** | **LLMMixer*** | **ModernTCN*** | **UniTime*** |
> | :----------: | :--------: | :--------: | :----------: | :-----------: | :---------: |
> |    **KR**    |   14.62   |   13.89    |    14.08     |     13.66    |    14.15   |
> |    **EWJ**   |   14.41   |   13.80   |    13.97    |     13.57    |    14.06   |
> |    **MDT**   |   14.69   |   13.85    |    14.13    |     13.61    |   14.20  |
>
> These observations collectively demonstrate that MindTS maintains reasonable computational scalability and remains practical for real-world multimodal time series applications.

---

> > ### Comment · Reviewer_2Z5d · 2025-11-27
> >
> > I think my concerns are well addressed by the author's additional experiments. I will increase my rating score.

---

> > > ### Author Response · Authors · 2025-11-28
> > >
> > > Thank you very much for raising the rating. We sincerely appreciate your constructive feedback.

---

> ### Author Response · Authors · 2025-11-21
> **Response to W2：More qualitative case studies or explainable AI techniques would help clarify how the semantic alignment contributes to detection outcomes.**
>
> ### W2：More qualitative case studies or explainable AI techniques would help clarify how the semantic alignment contributes to detection outcomes.
>
> In response, we visualize the similarity matrices between time series and text representations before and after alignment (see $\underline{\text{Figure 7 of the revised paper}}$). Prior to alignment, the similarity patterns are dispersed, reflecting weak semantic correspondence. After alignment, the similarity becomes more concentrated along the diagonal, revealing clear associations between relevant time series and text representations. This demonstrates that the alignment module enhances cross-modal consistency, enabling the model to better leverage textual information.
>
> Additionally, we have added qualitative visualizations in $\underline{\text{Appendix F (Figure 8) of the revised paper}}$, where we compare anomaly scores with and without alignment to illustrate alignment enhances the distinguishability of abnormal regions. **The first row** shows the original data with ground-truth anomaly positions, **the second row** displays the anomaly scores of the model with multimodal alignment, and **the third row** presents the anomaly scores of the model without multimodal alignment. When alignment is applied, the model exhibits more distinguishable anomaly scores around true abnormal regions. In contrast, the model without alignment fails to effectively increase the gap between the anomaly scores of normal points and anomalies, leading to many false positives.

---

> ### Author Response · Authors · 2025-11-21
> **Response to W3：The impact of exogenous text availability on MindTS performance.**
>
> ### W3：The impact of exogenous text availability on MindTS performance.
>
> Thank you for the suggestion. We would like to clarify that we provided comparative experiments under three types of exogenous text quality variations in Appendix E.7 (**Rebuttal Tables N, O, and P**).
>
> The three types of exogenous text quality variations are detailed as follows:
> (1) Noisy text, by injecting random spelling perturbations within sentences;
> (2) Irrelevant text, by replacing original content with unrelated sentences from other domains;
> (3) Incomplete text, by removing parts of the textual descriptions.
>
> Across these conditions, MindTS remains robust. The key findings are summarized below:
>
> **Rebuttal Table N.** Results under different types of exogenous text quality variations. Under single-type settings, the performance of MindTS decreases only slightly compared with the clean setting.
> | Method  |  MindTS | MindTS (Noisy)  |MindTS (Irrelevant) | MindTS (Incomplete) |
> | :--------- | :-------------------------------: | :-------------------: | :-------------------: | :-------------------: |
> | Metric |  Aff-F / V-PR / V-ROC |  Aff-F / V-PR / V-ROC |  Aff-F / V-PR / V-ROC |  Aff-F / V-PR / V-ROC |
> | **MDT** | **89.19** / **65.44** / **83.02** | 87.45 / 62.12 / 81.02 | 86.54 / 60.10 / 80.17 | 87.85 / 61.95 / 80.59 |
> | **Energy** | **74.37** / **50.36** / **74.44** | 72.53 / 47.42 / 72.16 | 72.21 / 46.52 / 71.29 | 73.13 / 47.62 / 73.50 |
>
>
> **Rebuttal Table O.** Results of the noisy method under different noise strengths. When the noise strength is within a reasonable range (e.g., 0.2 and 0.4), the content condenser effectively filters out redundant text information, thereby mitigating its impact. As a result, MindTS still maintains robust performance.
> | Dataset | MindTS | Noise strength with 0.2 | Noise strength with 0.4 | Noise strength with 0.6 | Noise strength with 0.8 |
> |:-------:|:--------------------------:|:---:|:---:|:---:|:---:|
> |  Metric |  Aff-F / V-PR / V-ROC | Aff-F / V-PR / V-ROC | Aff-F / V-PR / V-ROC | Aff-F / V-PR / V-ROC | Aff-F / V-PR / V-ROC |
> | MDT | **89.19** / **65.44** / **83.02** | 87.45 / 62.12 / 81.02 | 86.38 / 60.03 / 79.46 | 83.71 / 58.71 / 78.69 | 79.84 / 55.21 / 76.46 |
> | Energy| **74.37** / **50.36** / **74.44** | 72.53 / 49.42 / 73.16 | 71.17 / 48.26 / 70.03 | 69.35 / 47.22 / 69.97 | 66.69 / 45.02 / 66.13 |
>
>
> **Rebuttal Table P.** Results under combined types of exogenous text quality variations. Under more challenging conditions, such as multiple text types combinations or high noise intensity (e.g., 0.8), the performance degradation becomes more noticeable. Nevertheless, the overall results remain within acceptable bounds, demonstrating that MindTS is robust across both diverse noise types and varying intensity levels.
> | Method     | MindTS  | MindTS (Noisy + Irrelevant) | MindTS (Noisy + Incomplete) | MindTS (Irrelevant + Incomplete) | MindTS (Noisy + Irrelevant + Incomplete) |
> | :--------- | :-------------------------------: | :-------------------------: | :------------------------: | :--------------------------------: | :-------------------------------------------: |
> |  Metric | Aff-F / V-PR / V-ROC |Aff-F / V-PR / V-ROC |Aff-F / V-PR / V-ROC  | Aff-F / V-PR / V-ROC | Aff-F / V-PR / V-ROC  |
> | **MDT** | **89.19** / **65.44** / **83.02** |84.91 / 57.29 / 78.23|85.28 / 55.60 / 78.34   |84.81 / 58.34 / 77.68 |  81.64 / 52.17 / 75.28 |
> | **Energy** | **74.37** / **50.36** / **74.44** | 70.33 / 44.71 / 70.18|71.89 / 45.86 / 69.96   | 69.74 / 45.26 / 70.51| 66.05 / 42.08 / 66.88|

---

### Official Review · Reviewer_BUH8 · 2025-10-27

[review text omitted: it was posted to a different submission]

---

> ### Author Response · Authors · 2025-11-13
> **To Reviewer BUH8: Mismatch review**
>
> We thank reviewer BUH8 for the valuable and constructive comments.
>
> However, it appears that the current review content may not correspond to our submitted paper. We noticed another review on OpenReview that seems to align more closely with our work, and we would like to kindly confirm with the reviewer whether this is the correct one.
>
> ### Summary:
>
> This paper proposes MindTS, a multimodal time-series anomaly detection framework that leverages fine-grained alignment between time-series segments and textual signals from two sources: endogenous (template-guided descriptions generated per segment) and exogenous (external background texts). The method first aligns time–text features via cross-view attention and contrastive objectives, then performs a content condensation step (information-bottleneck-style masking) to reduce textual redundancy, and finally conducts cross-modal reconstruction with masked time-series to compute anomaly scores. Experiments over six real-world datasets and 17 baselines show consistently strong or SOTA performance, with ablations validating the necessity and ordering of the key modules.
>
> ### Soundness: 3: good
>
> ### Presentation: 3: good
>
> ### Contribution: 3: good
>
> ### Strengths:
>
> 1. The design of each modular ( alignment → condensation → cross-modal reconstruction ) is well-motivated and confirmed by ablation studies.
>
> 2. The empirical coverage is broad: 6 datasets, 17 baselines, plus multimodalized baselines, with consistent gains and careful sensitivity studies.
>
> 3. It provides insightful guidance on the coordinated use of endogenous and exogenous texts in alignment with time-series signals, and the proposed method for assessing and condensing informative content is compelling.
>
> 4. The exposition is well-structured and logically coherent, with rigorous reasoning throughout.
>
> ### Weaknesses:
>
> 1. Limited depth in optimizing multimodal alignment. Multimodal alignment somtimes trained as a standalone objective (e.g., BLIP2), whereas here it is treated as an auxiliary task, which may constrain the depth of alignment achieved. Although Figure 3 reports preliminary component ablations, a more systematic study would substantiate the alignment claims.
>
> 2. Insufficient ablation on endogenous prompt design. The construction of endogenous prompts appears to substantially affect performance in time-series contexts (cf. TimeLLM, HiTime). Some controlled ablations over prompt templates, statistical descriptors, and temporal granularity would clarify robustness and inform design choices.
>
> ### Questions:
>
> 1. It is better to train the multimodal alignment as a standalone objective (rather than only as an auxiliary loss) and report a dedicated set of results—ideally comparing standalone vs. auxiliary alignment settings.
>
> 2. It is better to provide a ablation study on the design of endogenous prompts (e.g., template variants, statistical descriptors, temporal granularity), to quantify their impact on the final performance.
>
> 3. The framework appears extensible to broader multimodal time-series tasks (e.g., classification and forecasting). It is better to evaluate on a few forecasting datasets, or more narrowly, on forecasting-based anomaly detection (early-warning) to assess transferability.
>
> ### Flag For Ethics Review: No ethics review needed.
>
> ### Rating: 6: marginally above the acceptance threshold. But would not mind if paper is rejected
>
> ### Confidence: 4: You are confident in your assessment, but not absolutely certain. It is unlikely, but not impossible, that you did not understand some parts of the submission or that you are unfamiliar with some pieces of related work.
>
> ### Code Of Conduct: Yes

---

> ### Author Response · Authors · 2025-11-21
> **Response to W1 & Q1: Discussion on training multimodal alignment as a standalone objective.**
>
> We sincerely thank Reviewer BUH8 for the recognition of our work and for providing constructive comments. Below, we address your concerns and questions individually:
>
> ### W1 & Q1: Discussion on training multimodal alignment as a standalone objective.
>
> As suggested, we compare multimodal alignment trained as a standalone objective with the auxiliary alignment setting used in MindTS. As shown in **Rebuttal Table H**, training alignment as a standalone objective does not lead to performance improvement. We clarify that although optimizing multimodal alignment alone may strengthen the alignment depth, it neglects the specific role of the learned representations in anomaly detection. In contrast, jointly optimizing alignment with other objectives allows multiple objectives to guide and regularize each other, enabling the model to emphasize features that are both semantically aligned and task-relevant. We have added the results in $\underline{\text{Table 30 of the revised paper}}$.
>
> **Rebuttal Table H.** Multimodal alignment as a standalone objective (all results in %, best in Bold).
>
> |   Dataset  |     Metric    |   MindTS (auxiliary)  | MindTS (standalone) |
> | :--------: | :-----------: | :-------: | :------------: |
> |   **MDT**  | Aff-F | **89.19** |      80.33     |
> |            |     V-PR    | **65.44** |      57.18     |
> |            |    V-ROC    | **83.02** |      76.35     |
> | **Energy** | Aff-F | **74.37** |      70.28     |
> |            |     V-PR    | **50.36** |      44.89     |
> |            |    V-ROC    | **74.44** |      69.61     |

---

> ### Author Response · Authors · 2025-11-21
> **Response to W2 & Q2: Ablation study on the design of endogenous prompts.**
>
> ### W2 & Q2: Ablation study on the design of endogenous prompts.
>
> As suggested, and inspired by the analyses in TimeLLM and HiTime, we have explored how different prompt designs affect model performance. The corresponding results and discussions have been added in $\underline{\text{Appendix G of the revised paper}}$.
>
> The endogenous prompt in MindTS is constructed using the following template:
>
> [start_prompt]
> \***
> [Dataset description]: <dataset_description>
> \***
> [Task description]: Reconstruct the <seq_len> steps given the previous <seq_len> steps.
> \***
> [Input statistics]:
> - Minimum value: <min_value>
> - Maximum value: <max_value>
> - Median value: <median_value>
> - Trend of input: <trend_description>
> - Top k lags: \<lags\>
>
> \***
> [end_prompt]
>
> We designed three groups of prompt variants to evaluate the robustness of MindTS under different design choices:
>
> (1) **Template variant (MindTS-T)**: prompts generated using a different template formulation, as follows:
>
> [Dataset]: The weather data includes temperature and humidity statistics as well as reports collected from government websites. [Task]: In this reconstruction task, the model is required to reconstruct the <seq_len> time steps. [Statistics]: Input value ranges from \<min\> to \<max\>, with a median of \<median\> and overall \<downward\> trend, and top k lags are \<lags\>.
>
> (2) **Statistical variant (MindTS-S)**: removing parts of the statistical descriptors:
>
> (a) keeping only dataset description;
> (b) removing min, max, median values;
> (c) removing trend information;
> (d) removing lag information;
>
> (3) **Temporal granularity variant (MindTS-TG)**: changing endogenous text generation from the per-patch level to the per-sample level.
>
> As shown in **Rebuttal Table I**, altering only the template formulation while keeping the content unchanged has almost no impact on performance. When a few statistical descriptors are removed, the performance decline is minor. However, when most of the statistics are omitted (MindTS-S(a)), performance drops more noticeably, as the generated endogenous text becomes too sparse to convey meaningful information. Changing the temporal granularity of endogenous prompt generation also leads to noticeable performance differences, primarily because coarse-grained endogenous text weakens MindTS’s ability to achieve fine-grained alignment. These results together confirm the robustness of MindTS to reasonable variations in prompt design.
>
> We further clarify that the selected statistical descriptors represent fundamental characteristics of time series. Therefore, the performance improvement does not depend on carefully tuning the prompts but instead arises from the intrinsic capability of the proposed model itself.
>
> **Rebuttal Table I.** The results of different designs of endogenous prompts.
>
> | Dataset    | Metric        |   MindTS  | MindTS-S (a) | MindTS-S (b) |  MindTS-S (c) | MindTS-S (d) |    MindTS-T   |  MindTS-TG |
> | :--------- | :------------ | :-------: | :--------: | :--------: | :---: | :---: | :---: | :-------: |
> | **MDT**    | Aff-F | 89.19 |    83.22   |   87.59 | 87.62 | 87.14 |   88.38   | 84.78 |
> |            | V-PR        | 65.44 |    57.27   |    64.05 | 64.49 | 64.33 |   64.65   | 58.49 |
> |            | V-ROC       | 83.02 |    78.72   |   82.75 | 82.13 |  82.90 | 83.19 | 80.75 |
> | **Energy** | Aff-F | 74.37 |    66.65   |  73.74 | 74.03 | 73.85 |   74.21   | 69.31 |
> |            | V-PR        | 50.36 |    44.29   | 48.77 | 49.25 | 49.81 |   49.34   | 45.32 |
> |            | V-ROC       | 74.44|    68.59   | 72.33 | 71.78 | 72.66 |   72.14   | 67.97 |

---

> ### Author Response · Authors · 2025-11-21
> **Response to Q3：The extensibility of MindTS to other multimodal time series tasks.**
>
> ### Q3：The extensibility of MindTS to other multimodal time series tasks.
>
> We need to clarify that MindTS is indeed designed with anomaly detection as the primary task, but its architectural components are inherently extensible. As suggested, we have revised the paper ($\underline{\text{see Appendix H}}$) and further conducted forecasting experiments to empirically validate its extensibility.
>
> MindTS proposes a fine-grained time-text semantic alignment mechanism consisting of endogenous text generation, cross-view text fusion, and a multimodal alignment strategy to ensure that time series and text semantics are consistently matched. Since accurate alignment is crucial for multimodal tasks involving heterogeneous semantic spaces, the alignment mechanism in MindTS possesses extensibility. Moreover, MindTS incorporates a content condenser to filter redundant textual information before cross-modal interaction. Redundant text is a common challenge in many multimodal applications. Together, these components make MindTS applicable to other multimodal time series applications. As shown in **Rebuttal Table J**, MindTS achieves competitive performance across multiple forecasting datasets, demonstrating that the core components generalize effectively beyond anomaly detection. These results confirm the extensibility of MindTS to other multimodal time series applications.
>
>
> **Rebuttal Table J.** Multimodel time series forecasting results with forecasting horizons $F \in \{6, 8, 10, 12\}$.
> |     Dataset     | Horizon | MindTS (mse/mae) | Time-MMD (mse/mae) | TaTS (mse/mae) |
> | :-------------: | :-----: | :--------------: | :----------------: | :------------: |
> | **Agriculture** |    6    | 0.167/0.269 |  0.146/0.263 |     0.140/0.251           |
> |                 |    8    | 0.195 /0.283   |  0.189/0.310     | 0.187/0.282      |
> |                 |    10   |   0.228/0.316 |  0.254/0.320    |  0.244/0.320   |
> |                 |    12   | 0.258 /0.343  | 0.338/0.369   | 0.290/0.350  |
> |        | **AVG**    | **0.212**/0.303 | 0.232/0.316 | 0.215/**0.301** |
> |   **Traffic**   |    6    |  0.157/0.225   |  0.162/0.242   | 0.174/0.239 |
> |                 |    8    | 0.176/0.251    | 0.168/0.228  | 0.178/0.242 |
> |                 |    10   | 0.167/0.213       | 0.178/0.237 | 0.185/0.243 |
> |                 |    12   | 0.181/0.237     | 0.188/0.246    | 0.189/0.242 |
> |        |**AVG**    |  **0.170/0.232** | 0.174/0.239 | 0.179/0.238 |
> | **Economy** |    6    |   0.172/0.331         |   0.199/0.350 | 0.196/0.350|
> |                 |    8    |   0.215 /0.370      | 0.216/0.367  | 0.214/0.376|
> |                 |    10   |  0.215/0.363      |  0.224/0.373   | 0.223/0.367|
> |                 |    12   |0.242/0.379    |  0.239/0.388    | 0.239/0.388|
> |     | **AVG**     | **0.211**/**0.361** | 0.219/0.370 | 0.215/0.368 |

---

### Official Review · Reviewer_v5hT · 2025-10-30

**Soundness:** 3
**Presentation:** 3
**Contribution:** 3
**Rating:** 6
**Confidence:** 4

**Summary:**

This paper introduces MindTS, a multimodal time-series anomaly detection framework that integrates numerical signals with textual information from both endogenous (LLM-generated from time segments) and exogenous (external documents or reports) sources. The model addresses two main challenges: achieving semantic alignment between heterogeneous modalities and filtering redundant textual content. It proposes a Fine-grained Time-Text Semantic Alignment module for cross-view fusion and alignment, and a Content Condenser Reconstruction mechanism that minimizes mutual information to distill informative text for cross-modal reconstruction.
Experiments on six real-world multimodal datasets show that MindTS consistently outperforms baselines, demonstrating the effectiveness and robustness of the proposed approach for multimodal anomaly detection.

**Strengths:**

1. The paper proposes a well-designed dual-view text alignment strategy that fuses endogenous and exogenous text to achieve precise semantic alignment between time and text modalities, effectively addressing heterogeneous data alignment issues.

2. The proposed Content Condenser, inspired by the Information Bottleneck principle, provides a principled solution to mitigate textual redundancy and noise in multimodal fusion, improving the efficiency of cross-modal interaction.

3. The cross-modal reconstruction task that reconstructs masked time series from condensed text is particularly innovative, encouraging the model to capture truly time-relevant textual information and enabling deeper cross-modal interaction.

4. The paper is clearly written and logically organized, with a coherent flow that makes the motivation well understood.

**Weaknesses:**

1. The paper claims to be multimodal, mentioning images and videos, but the proposed MindTS framework is actually limited to time-series–text fusion. Its core mechanisms are difficult to extend to other modalities. The authors should clarify this scope in the paper.
2. The paper lacks sufficient ablation on the design of endogenous prompts. Since the construction of these prompts (e.g., template formulation, choice of statistical descriptors, and temporal granularity) may strongly affect model performance, more experiments would help clarify their robustness and justify the design choices.

**Questions:**

1. More details on how the endogenous prompts are constructed and how sensitive the model is to their design would help strengthen the paper’s clarity and reproducibility.
2. Could the authors elaborate on the extensibility of MindTS? In particular, how might the proposed framework generalize to other multimodal time-series applications beyond anomaly detection?
3. Would it be possible to include an experiment that skips endogenous text generation and directly uses $H_{time}$ as the query for fusing exogenous text? This would clarify the actual benefit of the endogenous text generation step.

---

> ### Author Response · Authors · 2025-11-21
> **Response to W1: Clarifying that the framework is limited to time-series–text fusion.**
>
> We sincerely thank Reviewer v5hT for the recognition of our work and for providing constructive comments. Below, we address your concerns and questions individually:
>
> ### W1: Clarifying that the framework is limited to time-series–text fusion.
>
> As suggested, we have explicitly clarified in the revised paper ($\underline{\text{Section 1, lines 47–48}}$) that MindTS is designed specifically for time-series–text fusion, rather than a universal multimodal framework.

---

> ### Author Response · Authors · 2025-11-21
> **Response to W2 & Q1: Ablation study on the design of endogenous prompts.**
>
> ### W2 & Q1: Ablation study on the design of endogenous prompts.
>
> As suggested, we have examined how different prompt designs influence model performance. The corresponding experimental results and analyses have been added to the revised paper ($\underline{\text{see Appendix G}}$).
>
> In MindTS, the endogenous prompt is constructed using the following template:
>
> [Start prompt]
> \***
> [Dataset description]: <dataset_description>
> \***
> [Task description]: Reconstruct the <seq_len> steps given the previous <seq_len> steps.
> \***
> [Input statistics]:
> - Minimum value: <min_value>
> - Maximum value: <max_value>
> - Median value: <median_value>
> - Trend of input: <trend_description>
> - Top k lags: \<lags\>
>
> [End prompt]
>
> We designed three groups of prompt variants to evaluate the robustness of MindTS under different design choices:
>
> (1) **Template variant (MindTS-T)**: prompts generated using a different template formulation, as follows:
>
> [Dataset]: Gasoline prices display moderate weekly fluctuations driven by market dynamics and policy changes. [Task]: In this reconstruction task, the model is required to reconstruct the <seq_len> time steps. [Statistics]: Input value ranges from \<min\> to \<max\>, with a median of \<median\> and overall \<downward\> trend, and top k lags are \<lags\>.
>
> (2) **Statistical variant (MindTS-S)**: removing parts of the statistical descriptors:
>
> (a) keeping only dataset description;
> (b) removing min, max, median values;
> (c) removing trend information;
> (d) removing lag information;
>
> (3) **Temporal granularity variant (MindTS-TG)**: changing endogenous text generation from the per-patch level to the per-sample level.
>
> As shown in **Rebuttal Table E**, altering only the template formulation while keeping the content unchanged has almost no impact on performance. When a few statistical descriptors are removed, the performance decline is minor. However, when most of the statistics are omitted (MindTS-S(a)), performance drops more noticeably, as the generated endogenous text becomes too sparse to convey meaningful information. Changing the temporal granularity of endogenous prompt generation also leads to noticeable performance differences, primarily because coarse-grained endogenous text weakens MindTS’s ability to achieve fine-grained alignment. These results together confirm the robustness of MindTS to reasonable variations in prompt design.
>
> We further clarify that the performance improvement does not rely on carefully tuning the prompts, but instead stems from the intrinsic capability of the proposed model itself.
>
> **Rebuttal Table E.** The results of different designs of endogenous prompts.
>
> | Dataset    | Metric        |   MindTS  | MindTS-S (a) | MindTS-S (b) |  MindTS-S (c) | MindTS-S (d) |    MindTS-T   |  MindTS-TG |
> | :--------- | :------------ | :-------: | :--------: | :--------: | :---: | :---: | :---: | :-------: |
> | **MDT**    | Aff-F | 89.19 |    83.22   |   87.59 | 87.62 | 87.14 |   88.38   | 84.78 |
> |            | V-PR        | 65.44 |    57.27   |    64.05 | 64.49 | 64.33 |   64.65   | 58.49 |
> |            | V-ROC       | 83.02 |    78.72   |   82.75 | 82.13 |  82.90 | 83.19 | 80.75 |
> | **Energy** | Aff-F | 74.37 |    66.65   |  73.74 | 74.03 | 73.85 |   74.21   | 69.31 |
> |            | V-PR        | 50.36 |    44.29   | 48.77 | 49.25 | 49.81 |   49.34   | 45.32 |
> |            | V-ROC       | 74.44|    68.59   | 72.33 | 71.78 | 72.66 |   72.14   | 67.97 |

---

> ### Author Response · Authors · 2025-11-21
> **Response to Q2: The extensibility of MindTS to other multimodal time series applications.**
>
> ### Q2: The extensibility of MindTS to other multimodal time series applications.
>
> We clarify that the MindTS is primarily designed for anomaly detection, yet its architectural components are inherently extensible. We have revised the paper ($\underline{\text{see Appendix H}}$) and additionally conducted forecasting experiments to further demonstrate the extensibility of MindTS.
>
> MindTS proposes a fine-grained time-text semantic alignment mechanism consisting of endogenous text generation, cross-view text fusion, and a multimodal alignment strategy to ensure that time series and text semantics are consistently matched. Since accurate alignment is crucial for multimodal tasks involving heterogeneous semantic spaces, the alignment mechanism in MindTS possesses extensibility. Moreover, MindTS incorporates a content condenser to filter redundant textual information before cross-modal interaction. Redundant text is a common challenge in many multimodal applications. Together, these components make MindTS applicable to other multimodal time series applications. As shown in **Rebuttal Table F**, MindTS achieves competitive performance across multiple forecasting datasets, demonstrating that the core components generalize effectively beyond anomaly detection. These results confirm the extensibility of MindTS to other multimodal time series tasks.
>
>
> **Rebuttal Table F.** Multimodel time series forecasting results with forecasting horizons $F \in \{6, 8, 10, 12\}$.
> |     Dataset     | Horizon | MindTS (mse/mae) | Time-MMD (mse/mae) | TaTS (mse/mae) |
> | :-------------: | :-----: | :--------------: | :----------------: | :------------: |
> | **Agriculture** |    6    | 0.167/0.269 |  0.146/0.263 |     0.140/0.251           |
> |                 |    8    | 0.195 /0.283   |  0.189/0.310     | 0.187/0.282      |
> |                 |    10   |   0.228/0.316 |  0.254/0.320    |  0.244/0.320   |
> |                 |    12   | 0.258 /0.343  | 0.338/0.369   | 0.290/0.350  |
> |        | **AVG**    | **0.212**/0.303 | 0.232/0.316 | 0.215/**0.301** |
> |   **Traffic**   |    6    |  0.157/0.225   |  0.162/0.242   | 0.174/0.239 |
> |                 |    8    | 0.176/0.251    | 0.168/0.228  | 0.178/0.242 |
> |                 |    10   | 0.167/0.213       | 0.178/0.237 | 0.185/0.243 |
> |                 |    12   | 0.181/0.237     | 0.188/0.246    | 0.189/0.242 |
> |        |**AVG**    |  **0.170/0.232** | 0.174/0.239 | 0.179/0.238 |
> | **Economy** |    6    |   0.172/0.331         |   0.199/0.350 | 0.196/0.350|
> |                 |    8    |   0.215 /0.370      | 0.216/0.367  | 0.214/0.376|
> |                 |    10   |  0.215/0.363      |  0.224/0.373   | 0.223/0.367|
> |                 |    12   |0.242/0.379    |  0.239/0.388    | 0.239/0.388|
> |     | **AVG**     | **0.211**/**0.361** | 0.219/0.370 | 0.215/0.368 |

---

> ### Author Response · Authors · 2025-11-21
> **Response to Q3: Using $\mathbf{H}_{\text{time}}$ as the query for fusing exogenous text.**
>
> ### Q3: Using $\mathbf{H}_{\text{time}}$ as the query for fusing exogenous text.
>
> In response, we have conducted an ablation study where the model directly uses $\mathbf{H}_{\text{time}}$ as the query to fuse exogenous text, skipping the endogenous text generation step.
>
> As shown in **Rebuttal Table G**, the model incorporating endogenous text achieves clearly better performance. The endogenous text is derived directly from the time series, capturing temporal characteristics that align closely with local patterns. Consequently, fusing endogenous and exogenous texts enables the model to extract supplementary information from exogenous sources that is most relevant to each time patch.
>
> In contrast, directly using $\mathbf{H}_{\text{time}}$ as the query to fuse exogenous text does not yield performance gains. This is likely because the time series and text modalities inherently reside in different semantic spaces, interacting them directly without prior time-text alignment results in insufficient information extraction. We have added the analysis in $\underline{\text{Table 29 of the revised paper}}$.
>
> **Rebuttal Table G.** Ablation study on $\mathbf{H}_{\text{time}}$ as the query to fuse exogenous text and skips endogenous text generation (all results in %, best in **bold**).
>
> | Dataset         |     Metric    |   MindTS  | $\mathbf{H}_{\text{time}}$ as query |
> | :-------------- | :-----------: | :-------: | :---------: |
> | **KR**          | Aff-F | **90.28** |    87.93    |
> |                 |     V-PR    | **53.15** |    48.27    |
> |                 |    V-ROC    | **89.86** |    89.02    |
> | **EWJ**         | Aff-F | **83.89** |    82.33    |
> |                 |     V-PR    | **50.42** |    48.98    |
> |                 |    V-ROC    | **84.12** |    81.48    |
> | **MDT**         | Aff-F | **89.19** |    87.85    |
> |                 |     V-PR    | **65.44** |    64.33    |
> |                 |    V-ROC    | **83.02** |    81.26    |
> | **Environment** | Aff-F | **85.29** |    82.67    |
> |                 |     V-PR    | **56.79** |    52.45    |
> |                 |    V-ROC    | **93.78** |    92.87    |
> | **Weather**     | Aff-F | **82.66** |    79.35    |
> |                 |     V-PR    | **57.48** |    56.21    |
> |                 |    V-ROC    | **82.64** |    80.13    |
> | **Energy**      | Aff-F | **74.37** |    72.69    |
> |                 |     V-PR    | **50.36** |    47.52    |
> |                 |    V-ROC    | **74.44** |    71.47    |

---

> > ### Comment · Reviewer_v5hT · 2025-11-28
> >
> > The authors’ rebuttal and revisions have resolved my previous concerns. I will increase my overall score accordingly and have no further questions.

---

> > > ### Author Response · Authors · 2025-11-28
> > >
> > > Many thanks for your positive support. We sincerely appreciate your time in reviewing our paper and for providing such insightful and valuable comments.

---

### Official Review · Reviewer_zQng · 2025-11-02

**Soundness:** 3
**Presentation:** 3
**Contribution:** 2
**Rating:** 6
**Confidence:** 3

**Summary:**

This paper introduces model for multimodal time series anomaly detection that integrates both numerical and textual modalities. MindTS seeks to capture richer contextual and semantic information by aligning time series data with exogenous (external) and endogenous textual descriptions. The method consists of two key innovations: (1) Fine-grained Time-Text Semantic Alignment, which uses cross-view fusion and contrastive learning to ensure semantic consistency between time series and text; and (2) Content Condenser Reconstruction, which filters redundant textual information using mutual information minimization and reconstructs masked time series data to enhance cross-modal interaction. Experiments on six real-world datasets demonstrate that MindTS outperforms or matches SOTA methods across multiple evaluation metrics.

**Strengths:**

1. Addresses a novel and underexplored problem: multimodal time series anomaly detection.
2. Introduces a well-motivated novel dual-text approach  for semantic alignment.
3. Extensive experiments with multiple datasets and a strong set of 17+ baselines.
4. Comprehensive ablation and sensitivity analyses that validate the contribution of each component.

**Weaknesses:**

1. Dependence on the quality and availability of exogenous text may reduce generalizability to domains lacking rich textual context.
2. The discussion could benefit from more qualitative examples showing how textual alignment improves anomaly detection decisions.
3. Computational cost and inference time are not reported.

**Questions:**

1. How is exogenous text collected or preprocessed for each dataset, and what ensures its semantic relevance to time segments? How do you make sure the text doesn't leak future ground truth information?
2. How does MindTS perform when textual data are missing or random or very noisy?
3. Could the authors provide runtime or complexity comparisons with other multimodal baselines?

---

> ### Author Response · Authors · 2025-11-21
> **Response to W1 & Q2: The impact of exogenous text availability on MindTS performance.**
>
> Thank you for your valuable and constructive comments. We appreciate your recognition of the MindTS framework, its strong empirical performance across diverse datasets. Below, we address your concerns and questions individually:
>
> ### W1 & Q2: The impact of exogenous text availability on MindTS performance.
>
> Thank you for the suggestion. We would like to clarify that we provided comparative experiments under three types of exogenous text quality variations in Appendix E.7 (**Rebuttal Tables A, B, and C**).
>
> The three types of exogenous text quality variations are detailed as follows:
> (1) Noise, by introducing random spelling errors within sentences;
> (2) Irrelevance, by replacing the original sentences with unrelated text from different domains;
> (3) Incompleteness, by removing portions of the text descriptions.
>
> MindTS demonstrates robustness across different types and degrees of exogenous text variation, with the following key observations:
>
> **Rebuttal Table A.** Results under different types of exogenous text quality variations. Under single-type settings, the performance of MindTS decreases only slightly compared with the clean setting.
> | Method  |  MindTS | MindTS (Noisy)  |MindTS (Irrelevant) | MindTS (Incomplete) |
> | :--------- | :-------------------------------: | :-------------------: | :-------------------: | :-------------------: |
> | Metric |  Aff-F / V-PR / V-ROC |  Aff-F / V-PR / V-ROC |  Aff-F / V-PR / V-ROC |  Aff-F / V-PR / V-ROC |
> | **MDT** | **89.19** / **65.44** / **83.02** | 87.45 / 62.12 / 81.02 | 86.54 / 60.10 / 80.17 | 87.85 / 61.95 / 80.59 |
> | **Energy** | **74.37** / **50.36** / **74.44** | 72.53 / 47.42 / 72.16 | 72.21 / 46.52 / 71.29 | 73.13 / 47.62 / 73.50 |
>
>
> **Rebuttal Table B.** Results of the noisy method under different noise strengths. When the noise strength is within a reasonable range (e.g., 0.2 and 0.4), the content condenser effectively filters out redundant text information, thereby mitigating its impact. As a result, MindTS still maintains robust performance.
> | Dataset | MindTS | Noise strength with 0.2 | Noise strength with 0.4 | Noise strength with 0.6 | Noise strength with 0.8 |
> |:-------:|:--------------------------:|:---:|:---:|:---:|:---:|
> |  Metric |  Aff-F / V-PR / V-ROC | Aff-F / V-PR / V-ROC | Aff-F / V-PR / V-ROC | Aff-F / V-PR / V-ROC | Aff-F / V-PR / V-ROC |
> | **MDT** | **89.19** / **65.44** / **83.02** | 87.45 / 62.12 / 81.02 | 86.38 / 60.03 / 79.46 | 83.71 / 58.71 / 78.69 | 79.84 / 55.21 / 76.46 |
> | **Energy** | **74.37** / **50.36** / **74.44** | 72.53 / 49.42 / 73.16 | 71.17 / 48.26 / 70.03 | 69.35 / 47.22 / 69.97 | 66.69 / 45.02 / 66.13 |
>
>
> **Rebuttal Table C.** Results under combined types of exogenous text quality variations. Under more challenging conditions, such as multiple text types combinations or high noise intensity (e.g., 0.8), the performance degradation becomes more noticeable. Nevertheless, the overall results remain within acceptable bounds, demonstrating that MindTS is robust across both diverse noise types and varying intensity levels.
> | Method     | MindTS  | MindTS (Noisy + Irrelevant) | MindTS (Noisy + Incomplete) | MindTS (Irrelevant + Incomplete) | MindTS (Noisy + Irrelevant + Incomplete) |
> | :--------- | :-------------------------------: | :-------------------------: | :------------------------: | :--------------------------------: | :-------------------------------------------: |
> |  Metric | Aff-F / V-PR / V-ROC |Aff-F / V-PR / V-ROC |Aff-F / V-PR / V-ROC  | Aff-F / V-PR / V-ROC | Aff-F / V-PR / V-ROC  |
> | **MDT** | **89.19** / **65.44** / **83.02** |84.91 / 57.29 / 78.23|85.28 / 55.60 / 78.34   |84.81 / 58.34 / 77.68 |  81.64 / 52.17 / 75.28 |
> | **Energy** | **74.37** / **50.36** / **74.44** | 70.33 / 44.71 / 70.18|71.89 / 45.86 / 69.96   | 69.74 / 45.26 / 70.51| 66.05 / 42.08 / 66.88|

---

> ### Author Response · Authors · 2025-11-21
> **Response to W2: The discussion could benefit from more qualitative examples showing how textual alignment improves anomaly detection decisions.**
>
> ### W2: The discussion could benefit from more qualitative examples showing how textual alignment improves anomaly detection decisions.
>
> Following your valuable suggestion, we have visualized the learned similarity matrix between time series and text representations before and after alignment (see $\underline{\text{Figure 7 of the revised paper}}$). Before alignment, the similarity distribution appears scattered, indicating weak semantic correspondence between modalities. After alignment, the similarity becomes more concentrated along the diagonal, revealing clear associations between relevant time series and text representations. This demonstrates that the proposed alignment module successfully establishes cross-modal consistency, which in turn enhances the model’s ability to utilize textual information.
>
> In addition, we have added qualitative visualizations in $\underline{\text{Appendix F (Figure 8) of the revised paper}}$, where we compare anomaly scores with and without alignment to illustrate alignment enhances the distinguishability of abnormal regions. **The first row** shows the original data with ground-truth anomaly positions, **the second row** displays the anomaly scores of the model with multimodal alignment, and **the third row** presents the anomaly scores of the model without multimodal alignment. When alignment is applied, the model exhibits more distinguishable anomaly scores around true abnormal regions. In contrast, the model without alignment fails to effectively increase the gap between the anomaly scores of normal points and anomalies, leading to many false positives.

---

> ### Author Response · Authors · 2025-11-21
> **Response to W3 & Q3: Computational cost and inference time.**
>
> ### Response to W3 & Q3: Computational cost and inference time.
>
> We sincerely thank you for raising this insightful concern. We would like to clarify that we compared MindTS with other models across different datasets in terms of inference time and memory cost in Appendix E.6. As shown in **Rebuttal Table D**, MindTS achieves competitive inference time while maintaining superior detection performance. Regarding memory usage, the additional cost is moderate and remains well within the capacity of modern hardware, making MindTS practical for real-world deployment. We have clarified this point more explicitly in the revised paper.
>
> **Rebuttal Table D.** Run times on different datasets. Lower values represent the better performance. The notation with $*$ denotes results obtained by extending the baselines using the recent time-series multimodal framework MM-TSFLib.
>
> | Method | Inference Time (s) | Memory Cost (GB)  |
> |:-------------:|:---------------------------:|:---------------------------:|
> | Metric| MDT / KR / EWJ  | MDT / KR / EWJ |
> | MindTS | 0.2302 / 0.1977 / 0.4130 | 14.69 / 14.62 / 14.41 |
> | ModernTCN*| 0.1582 / 0.1383 / 0.3965 | 13.61 / 13.66 / 13.57|
> | GPT4TS* | 0.2676 / 0.2425 / 0.4716 | 13.85 / 13.89 / 13.80 |
> | LLMMixer* | 0.2585 / 0.2104 / 0.4619 | 14.13 / 14.08 / 13.97 |
> | UniTime* | 0.2537 / 0.2462 / 0.4760 | 14.20 / 14.15 / 14.06 |

---

> ### Author Response · Authors · 2025-11-21
> **Response to Q1: Collection and processing of exogenous text.**
>
> ### Q1: Collection and processing of exogenous text.
>
> Thank you for the constructive comment. All data collection and preprocessing strategies follow established practices in prior work [1]. We have revised the paper in Appendix A, and provide details below.
>
> (1) Given the broad coverage and initial relevance of exogenous text to time series, text sources are collected through web search and targeted crawling, combining widely sourced online content with domain-related reports.
>
> (2) To ensure semantic relevance between exogenous text and time series, 2–3 domain-specific keywords are defined for each dataset and used for web search. For each keyword, the top-ranked search results are collected, and structured information such as timestamp, source, title, and content is extracted. For report-based sources (e.g., governmental or institutional reports), all available documents are parsed, and only the sections whose topics match the corresponding domain characteristics are retained.
>
> (3) To prevent future ground-truth leakage, two safeguards are applied. First, all collected texts contain explicit timestamps, ensuring that no future documents are matched with past observations. Second, each text is further separated into historical factual statements and predictive descriptions, and only factual content is retained. This prevents any predicted future outcomes or implicitly revealed future values from leaking into the model.
>
> [1] Time-mmd: A new multi-domain multimodal dataset for time series analysis.

---

### Author Response · Authors · 2025-11-30
**Global Response and Summary of Revisions**

Dear all,

We are sorry to hear about the recent OpenReview bug issue, and we fully support the proposed remedy actions.

At the same time, we believe it is necessary to record the changes in our scores throughout the fruitful rebuttal phase up to Nov. 29. Our reasons are as follows:

1. **Appreciation for the review process.**
We sincerely thank all reviewers, the PC, SAC, the original AC, and the new AC for your time and effort in coordinating and evaluating our submission. The feedback from each reviewer has been invaluable throughout the entire process.

2. **Clarification regarding the misassigned review.**
The review submitted by Reviewer **BUH8** is not related to our paper. After checking active submissions in OpenReview, we identified another review (ID: 12086, title: “Multi-Scale Hypergraph Meets LLMs: Aligning Large Language Models for Time Series Analysis”, by Reviewer GFpP) that correctly corresponds to our submission. The correct initial rating for our paper is **6**, whereas the incorrect review gives a rating of 4. We have already responded to the correct review under the comments of Reviewer BUH8 and reported this issue to the AC at the beginning of the rebuttal.

3. **Score changes after rebuttal.**
**During the rebuttal, reviewers (Reviewer 2Z5d, v5hT) explicitly stated that their concerns have been well addressed and indicated increase rating score**.

**Table 1.** Initial Score -> After Rebuttal (Nov. 29).

| Reviewer ID  | Initial Score  | After Rebuttal Score  |
| :--------- | :-------------------------------: | :-------------------: |
| **zQng** |  6 |  6 (unchanged) |
| **v5hT** | **6** | **6 (comment indicates intention to increase score)** |
| **BUH8** | **4 (incorrect; correct score is 6)** | **4 (incorrect; correct score is 6)**|
| **2Z5d** | **6**  | **8** |

4. **Reviewers’ Opinions.**
The reviewers generally held positive opinions of our paper, as summarized in Table 2.

**Table 2.** Summary of Reviewers’ Opinions.
| Reviewer ID  | Aspect | Opinions  |
| :--------- | :-------------------------------: | :-------------------: |
| **zQng, v5hT, BUH8, 2Z5d** |  **Method**|  "**particularly innovative**", "**well-motivated**", "**technically novel**" |
| **zQng, BUH8, 2Z5d** | **Experiments** | "**extensive**",  "**comprehensive comparisons and ablation studies**"|
| **v5hT, BUH8** | **Writing** | "**clearly**", "**logically organized**", "**well-structured**"|

5. **Rebuttal summary.**
The reviewers raised insightful and constructive concerns. We made every effort to address all the concerns by providing sufficient evidence and requested results. Here is the summary of the major revisions, as shown in Table 3:

**Table 3.** Rebuttal summary.
| Reviewer ID  | Concerns | Revisions  |
| :--------- | :-------------------------------: | :-------------------: |
| **zQng, 2Z5d** |  **Impact of exogenous text quality.**|  Experiments for three text-quality variations (Appendix E.7), showing MindTS remains robust under different noise types and intensity levels. |
| **zQng, 2Z5d** | **Providing intuitive alignment examples.** | Added visual comparisons of anomaly scores with/without alignment, illustrating the benefit of the alignment module. |
| **zQng, 2Z5d** | **Efficiency analysis.** | Provided runtime and memory analysis, showing competitive training/inference time and moderate memory overhead. |
| **v5hT, BUH8** | **Design of endogenous prompts.** | Compared three prompt-design variants. Results show that performance improvement does not rely on carefully tuning the prompts. |
| **v5hT, BUH8** | **Ablation study.** | Added additional ablations (e.g., alternative query design, standalone alignment objective), further validating each module’s contribution. |
| **v5hT, BUH8** | **Extensibility to other time-series tasks.** | Extended MindTS to time series forecasting, demonstrating good adaptability. |


The valuable suggestions from reviewers are so helpful for us to revise the paper to a better version. All revisions have been included in the updated paper.

We hope that this summary will be helpful for the new AC in reducing their workload and making an informed final decision.

Best regards,

Authors

---

### Meta-Review · Area_Chair_2U8y · 2026-01-05

**Summary:**

The reviewers do not have major concerns on this paper. They all give the score of 6 to this paper. They only show some minor concerns such as computational cost.

**Reviewer Concerns:**

I think most of the concerns have been well addressed by the rebuttal.

**Reviewer Scores:**

Some reviewers have already increased their scores after rebuttal. Considering that the authors provided very detailed response and new experiments in rebuttal, I think the reviewers are not likely to change their scores from positive to negative.

---

### Decision · Program_Chairs · 2026-01-26

Accept (Poster)